# High microbiome and metabolome diversification in coexisting sponges with different bio-ecological traits
Valerio Mazzella [1,2], Antonio Dell'Anno [2,3] ✉, Néstor Etxebarría [4,5], Belén González-Gaya [4,5], Genoveffa Nuzzo [6], Angelo Fontana [6,7] & Laura Núñez-Pons [2,8] ✉

Marine Porifera host diverse microbial communities, which influence host metabolism and fitness. However, functional relationships between sponge microbiomes and metabolic signatures are poorly understood. We integrate microbiome characterization, metabolomics and microbial predicted functions of four coexisting Mediterranean sponges –*Petrosia ficiformis*, *Chondrosia reniformis*, *Crambe crambe* and *Chondrilla nucula*. Microscopy observations reveal anatomical differences in microbial densities. Microbiomes exhibit strong species-specific trends. *C. crambe* shares many rare amplicon sequence variants (ASV) with the surrounding seawater. This suggests important inputs of microbial diversity acquired by selective horizontal acquisition. Phylum Cyanobacteria is mainly represented in *C. nucula* and *C. crambe*. According to putative functions, the microbiome of *P. ficiformis* and *C. reniformis* are functionally heterotrophic, while *C. crambe* and *C. nucula* are autotrophic. The four species display distinct metabolic profiles at single compound level. However, at molecular class level they share a "core metabolome". Concurrently, we find global microbiome-metabolome association when considering all four sponge species. Within each species still, sets of microbe/metabolites are identified driving multi-omics congruence. Our findings suggest that diverse microbial players and metabolic profiles may promote niche diversification, but also, analogous phenotypic patterns of "symbiont evolutionary convergence" in sponge assemblages where holobionts co-exist in the same area.

Sponges are ubiquitous and functionally key components of marine communities, providing habitats for a variety of species, and coupling the benthic and pelagic ecosystems through intense seawater filtration and nutrient re-cycling[1,2]. These organisms have often been investigated as model holobiont systems to understand host-microbe and metabolic interactions, in the context of functional biological entities[3–5]. From this perspective, the present work focused on four widespread Mediterranean sponge holobionts (*Petrosia ficiformis*, *Chondrosia reniformis*, *Crambe crambe* and *Chondrilla nucula*) coexisting in high abundances across a study area of the Tyrrhenian Sea. These species were selected on the basis of distinctive bio-ecological potential range traits, involving reproductive strategies, microbial abundance, preferred habitat, general morphology, coloration, growth-type and allelochemistry (summarized in Table 1 and Fig. 1).

Sponges frequently host dense microbial assemblages as intracellular symbionts in specialized cells (bacteriocytes) and/or as free unicellular partners in the mesohyl, which in some cases account for up to 40-50% of the biomass[6,7]. Regarding the quantitative importance of the microbial

[1]Department of Integrative Marine Ecology (EMI), Stazione Zoologica Anton Dohrn, Ischia Marine Centre, 80077 Ischia, Naples, Italy. [2]NBFC, National Biodiversity Future Center, Piazza Marina 61, Palermo 90133, Italy. [3]Department of Life and Environmental Sciences, Polytechnic University of Marche, Via Brecce Bianche, 60131 Ancona, Italy. [4]Department of Analytical Chemistry, Faculty of Science and Technology, University of the Basque Country (UPV/EHU), Leioa, Basque Country, Spain. [5]Research Centre for Experimental Marine Biology and Biotechnology (PIE), University of the Basque Country (UPV/EHU), Plentzia, Basque Country, Spain. [6]Bio-Organic Chemistry Unit, Institute of Biomolecular Chemistry-CNR, Via Campi Flegrei 34, 80078 Pozzuoli, Italy. [7]Department of Biology, University of Naples Federico II, Via Cinthia–Bld. 7, 80126 Napoli, Italy. [8]Department of Integrative Marine Ecology (EMI), Stazione Zoologica Anton Dohrn, Villa Comunale, 80121 Naples, Italy. ✉e-mail: a.dellanno@univpm.it; laura.nunezpons@szn.it

**Table 1 | Summary of the main range of bio-ecological characteristics of the selected sponge species shown in Fig. 1 as living morphotypes**

| Sponge Species* | Reproduction | Microbiome | Preferred habitat | Allelochemistry | Morphology | Coluration |
|---|---|---|---|---|---|---|
| *Petrosia ficiformis*[18,33,54,58,96,168] | Oviparous | HMA | Sciophilous | Bioactive (secondary metabolites discovered) | Massive | Bleached or Pigmented |
| *Chondrosia reniformis*[58,104,169,170] | Oviparous | HMA | Sciophilous | Bioactive | Massive | Bleached or Pigmented |
| *Crambe crambe*[31,58,169,171,172] | Viviparous | LMA | Photophilous | Bioactive (secondary metabolites discovered) | Encrusting | Pigmented |
| *Chondrilla nucula*[21,22,33,58,173] | Oviparous | HMA | Photophilous | Bioactive | Massive | Bleached (very rare) or Pigmented |

*All samples in the same species are of the same phenotype: pigmented and massive for *P. ficiformis* and *C. reniformis*, pigmented and lobular for *C. nucula* and encrusting for *C. crambe*, and all from illuminated habitats and analogous conditions.

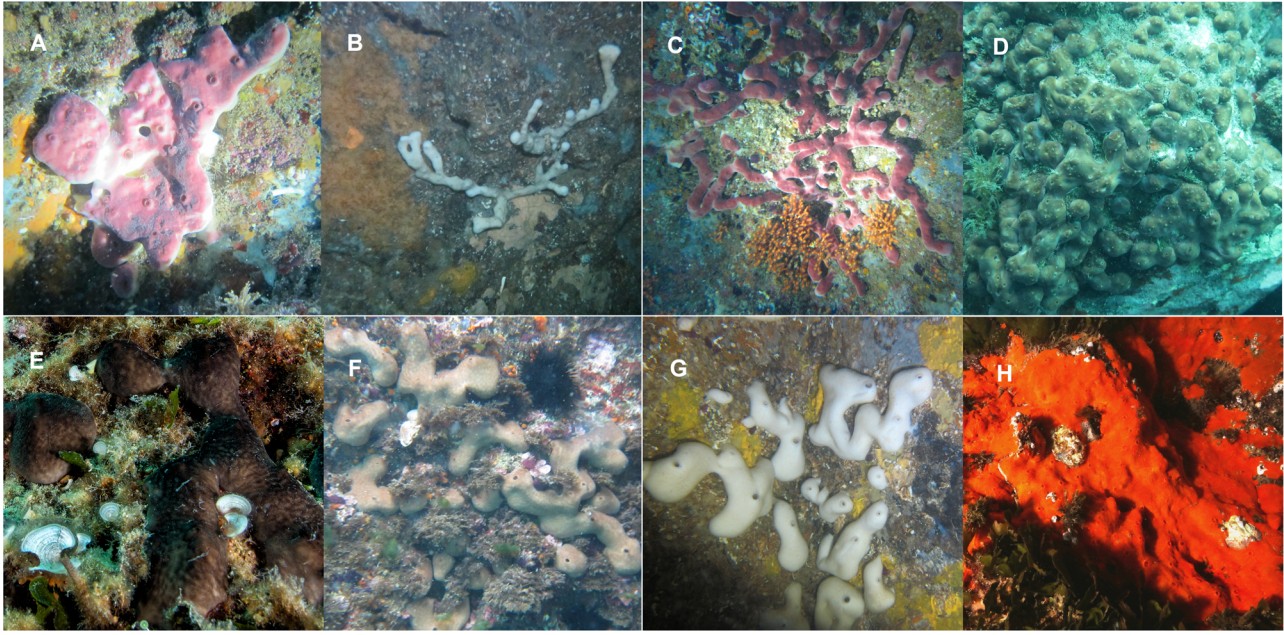

**Fig. 1 | Four sponge species considered in this study in their preferred habitat.** *Petrosia ficiformis*: **A** Classical massive form. **B** Reticular and bleached form (darkest habitats). **C** Reticular and colored form. **D** *Chondrilla nucula* in its preferred habitat (photic zone). *Chondrosia reniformis*: **E** Dark type, **F** Brownish type, **G** Bleached type (darkest habitats). **H** *Crambe crambe* patch in its preferred habitat (photic zone). The phenotypes collected to perform our study were represented in the pictures A, E, D, H for *P. ficiformis*, *C. reniformis*, *C. nucula* and *C. crambe* respectively.

compartment, sponges have been classified as "low microbial abundance"—LMA ($10^5$ to $10^6$ microbial cells/g tissue) and "high microbial abundance"—HMA (ca. $10^8$ to $10^{10}$ microbial cells/g tissue) species[6,8,9]. In general, HMA sponges –like *P. ficiformis*, *C. reniformis*, and *C. nucula*– compared with those LMA –such as *C. crambe*– display a denser mesohyl, lower pumping rates, and exhibit different mechanisms to process carbon and nitrogen compounds[10–12]. Notwithstanding this dichotomy, both groups of sponge holobionts display equivalent capabilities to colonize and flourish in the same range of habitat types[13,14].

The diversity, structure and composition of sponge microbiota communities have been widely investigated in the past decades[6,7]. Similar microbiomes have been frequently reported in the same host species from geographically distant locations[15], but still, there are cases (*e.g. P. ficiformis*) reporting microbial differences across different biogeographic regions[16–19]. Depending on the species and the environmental context, hosts can acquire microbial associates by two major mechanisms: via horizontal acquisition from the surrounding seawater (as majorly described for *P. ficiformis*), or by vertical transmission from parents to offspring in diversified reproductive strategies and modes (e.g., through brooding larvae as in *C. crambe*, or within oocytes in *C. nucula, C. reniformis*)[20–25]. Assemblages acquired

horizontally are in general more spatially and temporally variable (depending on time, location and environmental conditions), and can occur simultaneously involving similar taxa in diverse host species. Instead, inherited symbionts are more stable and are often species-specific[15,20,24]. All in all, it is broadly accepted that sponges combine different ratios of both microbial input types, relying on the ecological needs[20,24]; and that symbiont prokaryote communities reflect high degree of host species-specificity based on collections worldwide[6,7,15].

Microbial symbionts can supply nutrients and energy to their hosts, while processing waste products, and can as well produce secondary metabolites, useful to avoid predators and fouling, or to successfully compete for space[26]. Moreover, sponge microbiomes are involved in the C, N and S geochemical cycles, and in the nutrient re-cycling of the sponge loop, where DOM is filtered by sponges and subsequently released as POM in the form cellular detritus, thus becoming available for the neighboring trophic web levels[2,27]. Microhabitat acclimating aspects could be related to host-microbiome-metabolome interactions, and their joint potential. *Petrosia ficiformis* and *C. reniformis* for instance, flourish in large populations, either in illuminated areas as pigmented forms, or in shaded or even dark caves as bleached morphotypes often in reticulated growth (e.g., *P. ficiformis*)[28,29].

While *C. crambe* seem more abundant in the presence of some light or at most shade but not dark, and are colorful and charged of secondary metabolites to deter putative predators and competitors[30,31]. *Chondrilla nucula* still, are clearly photophilic and intolerant to shade-to-dark conditions[32,33]. There are still many gaps of knowledge on the metabolic functional role of diversified microbiomes in how these and other holobionts are adapted to similar or distinct environments[34–38]. Sponge microbiomes are known to benefit the holobiont via metabolic pathways of specific taxa[9,26,39]. In particular, cyanobacteria have been investigated for their involvement in the carbon cycle, and for supplying their hosts with >50% of energy requirements in certain cases –as suggested for *C. nucula*[32]. Nitrogen metabolism in sponge holobionts has been mainly attributed to ammonia-oxidizing archaea[9,40], while associated nitrogen fixation is considered to be a negligible source of nitrogen in Mediterranean sponges[41]. The exclusively sponge-associated genus Poribacteria, instead, has been found to be involved in nitrification and denitrification pathways[42]. In this regard, *C. reniformis* has been studied for its capability to oxidize exogenous ammonium, mediated by its associated microbial community[43]. Other sponge holobionts (e.g. *P. ficiformis*) undergo dark fixation processes that may involve anaplerotic carbon assimilation, which seem to be performed by Acidobacteria, Alphaproteobacteria and Chloroflexi[44]. Metabolic integration studies within several sponge holobionts reported carnitine derived from sponge cell debris, as a common microbial source of C, N and energy[45]; in concomitance with a clear involvement of microbial genes in the production of (primary and secondary) metabolites required for the host[46–48]. Predicted genomic functions of microbiomes linking metabolic signatures have been previously investigated[44,49–51], but it remains a poorly studied topic with still the requirement of rigorous experimental validations.

Generally, there is an equivocal belief that variance in host metabolome can be explained by microbiome modulation[52], but to what extent and which are the major drivers, and how this happens in marine holobionts is still poorly understood[52]. We found a wide gap in addressing microbiome-metabolome relatedness in Porifera, with regards the linkage between the microbial structure and taxonomy and the metabolic profiling phenotype. Therefore, networks of microbial taxa supplying access to metabolic pathways involved in phenotypes that afford host ecological success, are at the moment largely undiscovered, even if recent approaches are providing eager advances in the topic[48,49].

In this research we combined microscopy, metabarcoding and metabolomics approaches, along with predicted functions algorithms, to investigate the relationships between sponge microbiomes and metabolic signatures, in the context of host specificity, and considering several distinctive bio-ecological traits, characterizing the four holobiont systems selected. Multivariate correlation and distance-based methods were applied to characterize the strength of the dynamic relationship between the microbiome and the metabolome, and to identify key contributors to the overall concordance. To the best of our knowledge ours is one of the few studies analyzing microbiome-metabolomics relatedness in sponge holobionts, representing a novelty in the field of sponge ecology.

## Results

### Transmission electron microscopy observations

*Petrosia ficiformis* and *C. nucula* revealed numerous microbial cells with different morphologies, enclosed in host cells—"bacteriocytes" (Fig. S2A, B; G, H; Fig. S3B). Abundant cyanobacterial-like cells were remarkably present in *C. nucula*, whereas a lower number of cyanobacteria was observed in *P. ficiformis* (Fig. S2A, B; Fig. S3B–D). *C. crambe* and *C. reniformis* were characterized by the absence of bacteriocytes and by the presence of prokaryotic cells occupying the mesohyl, which were morphologically similar, in the sponge *C. crambe* (Fig. S2C–F).

### Microbiome diversity and community composition

Sequence data were rarefed to 7038 reads (Fig. S4) per sample resulting in 408204 reads distributed over 58 samples (sample details in Table S1).

Sequence reads ranged between a minimum of 7 and a maximum of 64215, distributed over 1181 ASVs. *Petrosia ficiformis* reported the highest number of ASVs (total = 425; mean = 174), followed by *C. crambe* (total = 343; mean = 95), *C. reniformis* (total = 225; mean = 106) and *C. nucula* (total = 121; mean = 79) (Fig. 2A, Fig. S6), and this pattern was also partially observed for the Shannon diversity index. Indeed *C. nucula* displayed in this case a higher alpha diversity (Fig. 2B). Microbial diversity in the seawater was higher with respect to any sponge investigated (Kruskal–Wallis, $p < 0.01$; Tables S2–S3). PCoA analysis based on Bray-Curtis dissimilarity (relative abundance) highlighted a clear separation in microbial composition among sponge species and with respect to the seawater (Fig. 2C, PERMANOVA, $p < 0.01$; Table S4). Similar results were highlighted also by nMDS analysis based on the Jaccard index (presence/absence), although with a certain overlap between *C. crambe* and the seawater (PERMANOVA, $p < 0.01$, Fig. 2D; Table S5).

Compositional RPCA (Robust Principal Component Analysis) based on Aitchison distance revealed four clusters of microbiomes significantly different among the four sponge species (Fig. 3A; PERMANOVA, $p < 0.01$, Table S6), and the twenty most significant ASVs driving sample grouping in the ordination space were identified (Fig. 3A). Log ratios from DEICODE feature-loadings grouped by lowest best taxonomy annotation level, and which best explained the separation of sample groupings along axis 2 in the RPCA space, were composed by 81 ASVs in the numerator and 50 in the denominator (Fig. 3C). In the numerator these differential taxa belonged to Subgroup 6 (Acidobacteria), SBR1031 (Chloroflexi), *Bdellovibrio* (Proteobacteria) and in the denominator to Rhodothermaceae, Cyanobiaceae, PAUC34f and EC94 (Betaproteobacteriales). *C. reniformis* displayed the highest log ratios, followed by *P. ficiformis, C. nucula* and *C. crambe* (Fig. 3B), and the differences in log ratios of the selected bacterial consortium were statistically significant for every comparison (Welch's Test, $p < 0.01$, Table S7).

A total of 27 microbial phyla were annotated across the four sponge species, distributed in 49 classes, 109 orders, 151 families and 225 genera (Fig. 4, Fig. S5, Table S8). Betaproteobacteriales, Flavobacteriales, SAR11 (Alphaproteobacteria), SAR202 clade (Chloroflexi) and Synechococcales were the most abundant bacterial orders across all samples. In *P. ficiformis* the microbiome was composed by 22 phyla, 40 classes, 68 orders, 81 families, and 99 genera. The five most abundant bacterial orders were Caldilineales (20%), SAR202 clade (Chloroflexi—15%), Gemmatimonadetes (8%), TK17 (Chloroflexi—5%) and Microtrichales (4%). Cyanobacteria and Archaea were only present in low percentages and accounted for 1.6 and 0.65% respectively.

*C. reniformis* harbored 18 phyla, 29 classes, 49 families, 54 orders and 61 genera. The most abundant bacterial orders were SAR202 clade (Chloroflexi—15%), Nitrosococcales (10%), Subgroup 6 (Acidobacteria—9%), SBR1031 (Chloroflexi—7%) and Myxococcales (7%). Archaea and Cyanobacteria accounted together <1%. The microbiome of *C. crambe* was composed by 18 phyla, 28 classes, 69 orders, 96 families and 133 genera. This species was dominated by an ASV belonging to the order Betaproteobacteriales, which accounted for ~73%. Cyanobacteria accounted for 10%, mostly belonging to the order Synechococcales. Finally, *C. nucula* were characterized by the presence of 16 prokaryotic phyla, 27 classes, 46 orders, 50 families and 55 genera. Synechococcales (Cyanobacteria) was the most abundant taxa associated with this species (18%). Other relevant taxa of *C. nucula* were Caldilineales (10%), Gammaproteobacteria (7%), TK17 (Chloroflexi, 7%) and Oceanospirillales (6%). Finally, Archaea accounted for <0.5%.

Seawater samples were characterized by the presence of 11 prokaryotic phyla, 17 classes, 57 orders, 79 families and 119 genera. The five most relevant ASVs belonged to Flavobacteriales (31%), SAR11 clade (Alphaproteobacteria—14%), Rhodobacterales (11%), Synechococcales (11%) and SAR86 clade (Gammaproteobacteria—8%). No archaea were detected in the seawater.

*C. nucula* displayed the highest relative abundance of cyanobacteria (7–26%), represented by genus *Synechococcus* (spongiarum group). *Synechococcus* (strain CC9902) was also dominant in *C. crambe* (4–19%), along

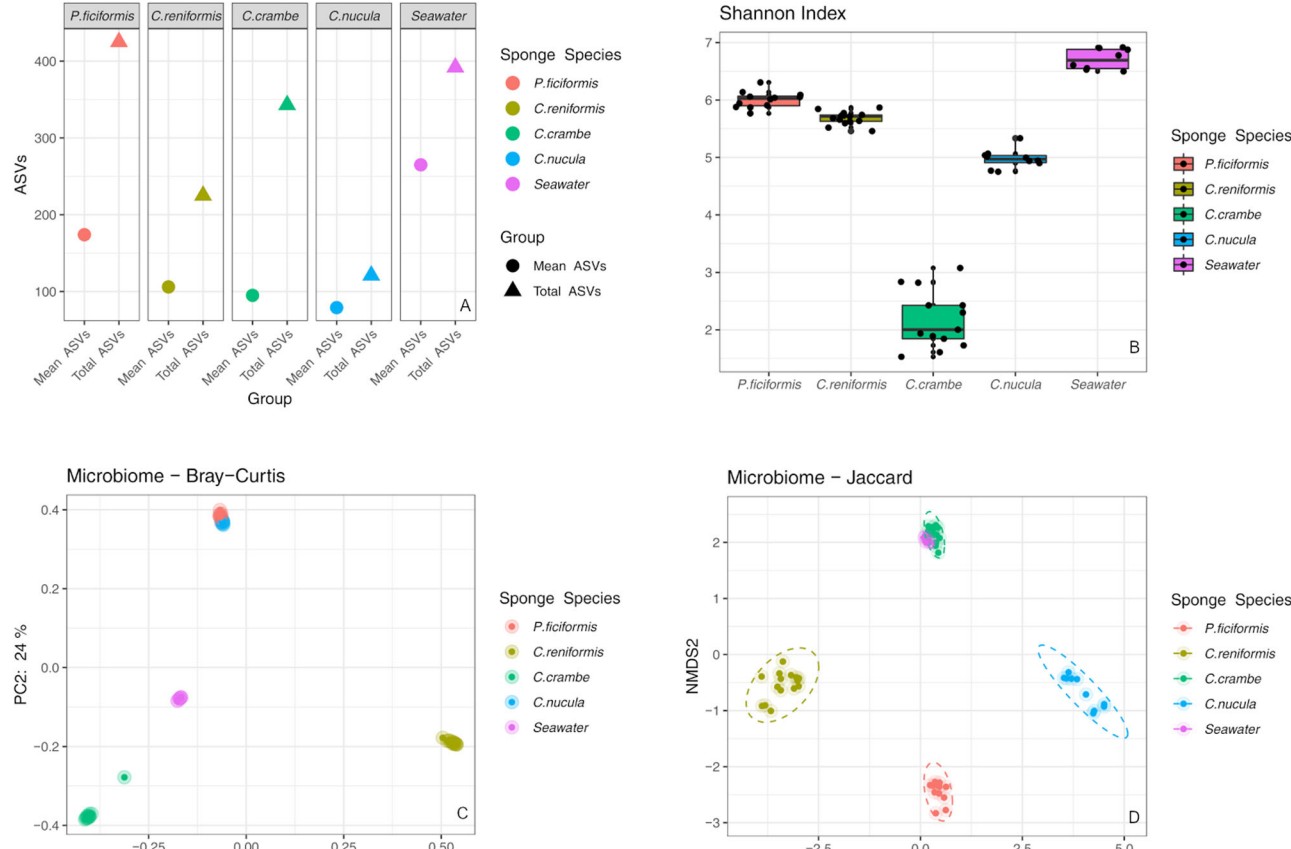

**Fig. 2 | Alpha and beta diversity plots of the microbiomes of the four sponge species and the surrounding seawater. A** Average number of ASVs per sample versus total ASVs across all samples. **B** Bacterial diversity (as Shannon index) across the different sponge species. PCoA and nMDS (2D stress = 0.085) of beta diversity ordinations based on Bray–Curtis (**C**) and Jaccard resemblances (**D**). Ellipses represent 95% confidence interval. Observed ASVs boxplots are reported in Fig. S6.

with genus *Cyanobium* (~1%). In *P. ficiformis* the most abundant cyanobacterial taxa belonged to family Cyanobiaceae (0.1–0.5%), followed by genera *Synechococcus* (strain CC9902), *Cyanobium* and *Prochlorococcus* (~0.1%). In *C. reniformis*, cyanobacteria accounted for a very low fraction of the total prokaryotic assemblages and were represented by the genus *Synechococcus* (strain CC9902) (~0.1%).

The four sponge species did not share any ASV (100% of the samples). *C. nucula* displayed the largest core microbiome in terms of ASV richness (55 ASVs), followed by *C. reniformis* (47), *P. ficiformis* (38), and *C. Crambe* (13; Fig. 5). *Crambe crambe* was the only host sharing core taxa with the seawater, sharing eight core ASVs. The size of the core microbiomes (percentage of the core microbiome with respect to the total) accounted for ~95% in *C. nucula*, ~81% in *C. crambe*, ~74% in *C. reniformis* and ~46%, *in P. ficiformis* (Fig. 5). The output of the network analysis indicated that each sponge species was characterized by the presence of several exclusive ASVs (Fig. 6A). In particular, *P. ficiformis* displayed the highest number of exclusive ASVs (337) followed by *C. reniformis* (179), *C. crambe* (146) and *C. nucula* with (99) (Fig. 6B). *C. crambe* shared the largest number of ASVs with seawater (175), followed by *P. ficiformis, C. reniformis and C. nucula* (64, 36 and 20 ASVs, respectively; Fig. 6B).

SIMPER analysis performed to estimate the contribution of each microbial taxon (%) to the dissimilarity between groups, confirmed high dissimilarity percentages (~98–99%) among microbiome composition of the four sponge species studied. Remarkable dissimilarities were also reported between all sponge microbiomes and a prokaryotic assemblages of the seawater. Instead, similarity of the microbiomes among individuals of the same sponge species was 76.4% for *C. nucula*, 75.8% for *C. crambe*, 68.5% for *C. reniformis*, 54.3% for *P. ficiformis*, and 61.8% for the seawater.

## Putative functions of prokaryotic assemblages
The ASV table of the full dataset was converted into putative functions table via cross-annotation against the Faprotax database[53] (https://pages.uoregon.edu/slouca/LoucaLab/archive/FAPROTAX/lib/php/index.php). The PCA ordination revealed three main clusters of putative functions patterns: one composed by *C. crambe* specimens, one by *C. nucula* and another one by *C. reniformis* and *P. ficiformis* (Fig. 6C; PERMANOVA, $p < 0.05$; Table S9).

The analysis of the putative functions revealed that the cluster composed by *P. ficiformis* and *C. reniformis* (Fig. 6D) was mostly related to functions of chemo-heterotrophy and aerobic chemo-heterotrophy (accounting together for 56 and 54%, respectively). Other relevant functions were nitrification-related processes, which accounted for ~32%, in *P. ficiformis* and ~20% in *C. reniformis*. *C. reniformis* was further characterized by predatory or exoparasitic related functions (16%) and fermentation (9%). The other two clusters formed by *C. crambe* and *C. nucula* were dominated by phototrophy, photo-autotrophy and cyanobacteria-related functions (>75%), with heterotrophy representing ~15% in both species. Nitrification-related functions accounted for ~4% in *C. nucula* and were negligible in *C. crambe* (ca. 1%). Other minor functions were exclusive of *C. crambe*.

## Metabolomics profiling
Metabolomics analysis carried out on 37 samples discriminated ~5000 chemical features grouped in 33 metabolite classes (further details in Table S12). Phosphatidylcholines (PC) were the most abundant organic compound class (Fig. 7A; Table S10), ranging from 33 to 49% in *P. ficiformis*, 63–66% in *C. reniformis*, 19–59% in *C. crambe* and 57–66% in *C. nucula*, followed by tryglicerides (TG) ranging from 18 to 38% in *P. ficiformis*, 6–9%

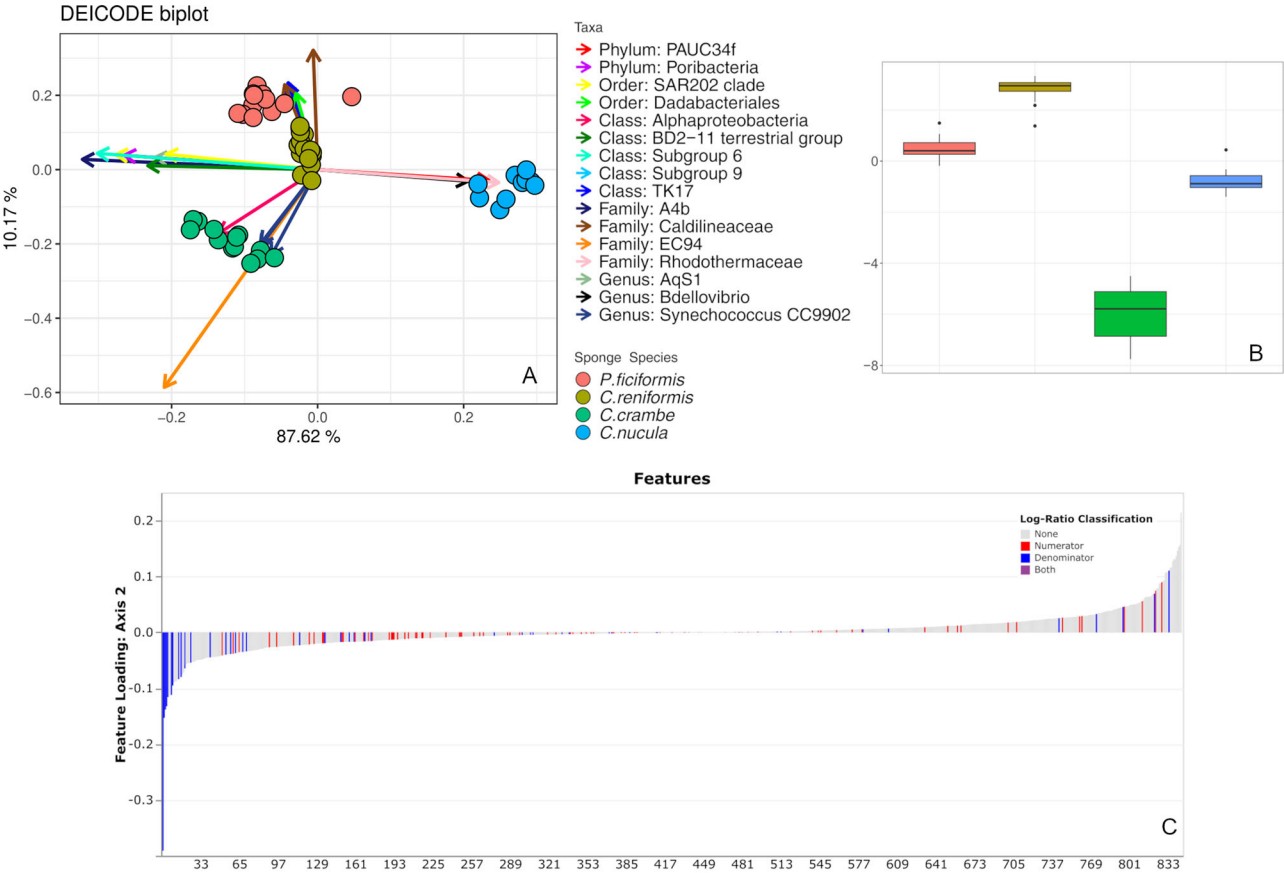

**Fig. 3 | Beta diversity insights based on Aitchison distance. A** Compositional RPCA biplot of beta diversity of bacterial communities associated with the four sponge species based on Aitchison distance. Sponge species are depicted by colors and points represent individuals. Twenty most relevant ASVs driving differences in the ordination space are illustrated by colored vectors labeled with the respective taxonomy, reported to the lowest best taxonomic annotation. **B** Box plots of log-ratios of the microbial balance generated by the selected bacterial consortium plotted across the four sponge species along axis 2 of the compositional RPCA. **C** Feature rankings, with the ASVs selected as numerator and denominator taxa to generate the microbial balance that maximized group separations by sponge species, along axis 2 of the compositional RPCA. *For other dimensional views, including the third ordination component PC3 of the RPCA plot (**A**) see Supplementary Fig. S7.

in *C. reniformis*, 0.6–7% in *C. nucula* and 0.05–55% in *C. crambe*. Phosphatidylethanolamines (PE) ranged from 7 to 16% in *P. ficiformis*, 8–11% in *C. reniformis*, 4–14% in *C. crambe* and 3–7% in *C. nucula*. Another relevant class of compounds were ceramides (Cer) which ranged from 2 to 8% in *P. ficiformis*, 2–4% in *C. reniformis*, 6–70% in *C. crambe* and 4–8% in *C. nucula*. Wax esters (WE) and methyl phosphatidylcholines (MePC) were detected only in the sponges *P. ficiformis* and *C. nucula*, ranging from 0.7 to 3% and 0.5 to 17%, respectively (Fig. 7A).

Network analysis highlighted major differences in the metabolomic profiles of the four sponges investigated (dissimilarity ~90%), with many chemical features being exclusive of each species, and a few being shared (Fig. 7B). Metabolic/chemical profiles based on molecular classes (functional biochemical affinity) were compared by presence/absence across species and visualized through network analysis (Fig. 7C). The four sponge species shared 14 molecular classes, including: triglycerides (TG), monoglycerides (MG), diglycerides (DG), phosphatidylcholines (PC), phosphatidylethanolamines (PE), phosphatidylserines (PS), phosphatidylglycerols (PG), coenzymes (Co), cholesterol esters (ChE), ceramides (Cer), ceramides phosphate (CerP), sitosterol esters (SiE), lysophosphatidylcholines (LPC), zymosterol ester (ZyE). Seven molecular classes (sulfatides ST, sphingosines SPH, wax esters WE, phosphatidylmethanols PMe, methyl phosphatidylcholines MePC, acyl carnitines AcCa, hexosylceramides Simple Glc Series) were shared only among *P. ficiformis*, *C. nucula* and *C. reniformis*, while three (hex1cer, hex2cer, phosphatidylinositols PI) were common to *P. ficiformis*, *C. reniformis* and *C. nucula*. Lysophosphatidylglycerols (LPG)

were only shared between *P. ficiformis* and *C. reniformis*. Stigmasterol esters (StE) were found in *P. ficiformis*, *C. reniformis* and *C. crambe*, while sphingomyelins (SM) were detected in *C. reniformis*, *C. crambe* and *C. nucula*. A few molecular classes were exclusive in each sponge species. *C. nucula* and *C. crambe* showed two exclusive molecular classes: monogalactosyl-diacylglycerol (MGDG) and acylGlcStigmasterol ester (AcHexStE) and phosphatidylinositols (PI) and lysophosphatidylinositol (LPI), respectively. *P. ficiformis* and *C. reniformis* were characterized by one exclusive molecular class each, campesterol ester (CmE) and lysophosphatidylethanolamines (LPE), respectively (Fig. 7C). Finally, differences in the overall metabolomics profiles among sponge species were highlighted by PCA (Fig. 7D) and PERMANOVA analyses ($p < 0.05$, Table S11).

**Procrustes distance based multi-omics concordance**

Strong global concordance between the microbiome and the metabolome was found when considering the overall sample set encompassing the four sponge hosts (Fig. 8), according to the symmetric Procrustes test ($p = 1.5$ e-5). Contrastingly, overall data structure association was not significant within any of the sponge species based on the respective paired PCAs (*C. nucula* – $p = 0.795$; *C. reniformis* – $p = 0.84$; *C. crambe* – $p = 0.78$; *P. ficiformis* – $p = 0.59$; Fig. 8).

**Spearman and Sparse canonical correlation (sCCA) analyses**

A total of 120 (95.2%) microbial genera and 33 (100%) metabolite classes revealed significant correlations in the full dataset (34.94% of total pairwise

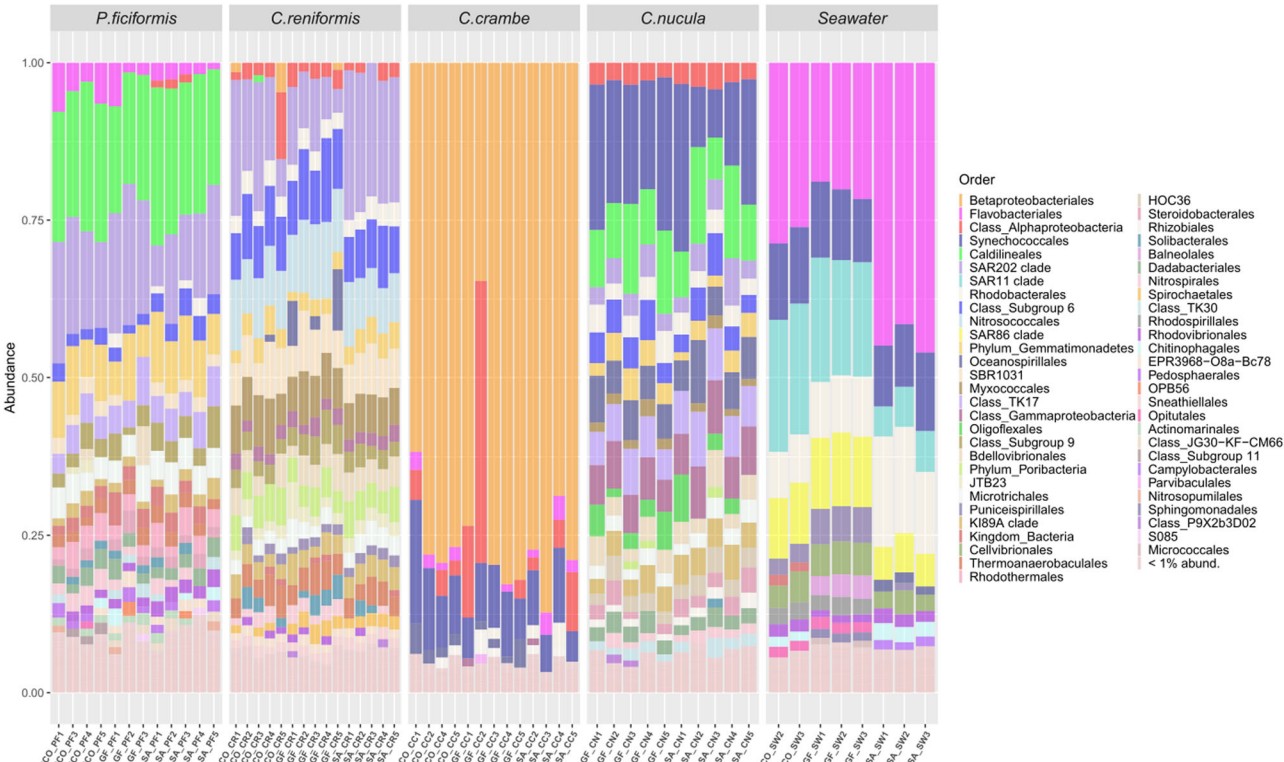

**Fig. 4 | Taxonomic composition (reported as relative abundance) of the microbiomes associated with different individuals of the four sponge species.** Taxa were represented at the best annotated taxonomic rank (i.e. order). Each label in the x axis represents a sample (details reported in the Table S1). *For whole bacterial community including rare background taxa see Fig. S5.

comparisons), including all four sponge species (Spearman, FDR controlled *q*-value < 0.05). This correlations pattern was strongly reduced when considering the subsets by the single sponge species. The number of significant correlations reported for represented microbial genera and metabolite classes were 12 (17%) and seven (21%) in *C. crambe*, seven (10%) and six (18%) in *P. ficiformis*, six (11%) and 12 (36%) in *C. nucula*, and four (8%) and six (18%) in *C. reniformis* (Spearman, FDR controlled *q*-value < 0.05). Overall multivariate correlation revealed significant results in the full dataset considering all four sponge species (permutation *p*-value = 0), and non-significance in the species subsets (*C. reniformis* permutation *p*-value = 0.995; *C. nucula* permutation *p*-value = 0.749; *P. ficiformis* permutation *p*-value = 0.131; *C. crambe* permutation *p*-value = 0.055). The canonical correlation was marginally higher in *P. ficiformis* (correlation: 0.989 [0.97–1]) than in *C. crambe* (correlation: 0.986 [0.955–1]), followed by *C. nucula* (correlation: 0.935 [0.925–0.998]), with non-significant appertaining differences due to overlapping confidence intervals. *Chondrosia reniformis* displayed the lowest correlation (0.816 [0.941–0.998]), and the degree of concordance was significantly lower than in *P. ficiformis* and *C. crambe*.

A suit of core microbial genera and metabolite classes leading the multivariate correlation were selected by sCCA (Fig. 9). *Petrosia ficiformis* revealed the largest set of informative microbial genera (12), of which only six were significantly correlated with three metabolite classes, being Sva0996 and TK17 correlated with ChE, NS3a and AEGEAN-169 with Co and SAR202 and Cryomorphaceae correlated with PMe. In *C. crambe*, out of the 11 genera selected by sCCA, only five revealed significant correlations with two metabolite classes. DG were correlated with *Romboutsia, Rubritalea, Blastopirellula* and OCS116; and Cer were correlated with *Romboutsia*, Cryomorphaceae and OCS116. All seven microbial genera selected by sCCA in *C. nucula* were significantly correlated with at least one metabolite class: Oligoflexaceae, TK17, *Synechococcus spongiarum*, UBA10353 and HOC36 were linked to DG; Oligoflexaceae, TK17, Synechococcus spongiarum and HOC36 were correlated with SiE; *Nitrospira*, Cytophagales and HOC36 were linked with ChE; and TK17 and *Synechococcus spongiarum* were

correlated with MePC. Finally, in *C. reniformis* only two of the seven selected microbial genera by sCCA revealed significant correlations with two metabolite classes, NS4 and Gammaproteobacteria were linked to PI and Clade la with ZyE.

## Discussion

Transmission electron microscopy observations reinforced previous findings that *P. ficiformis* and *C. nucula* host abundant and morphologically diverse microbes contained in bacteriocytes[54–57]. *Crambe crambe* was characterized by the presence of low abundant and dispersed microbes in the mesohyl, as previously reported[11,58–60], dominated by a single morphological type. In *C. reniformis*, our observations did not reveal the presence bacteriocytes. Previous studies instead, reported nurse cells charged with symbiotic bacteria during the development[20]. Molecular investigations highlighted that *C. crambe* hosts one or very few bacterial taxa, mostly (80–90%) belonging to phylum Proteobacteria[23,24,59,60]. Our results based on SILVA v.128 release 69, expand such taxonomic resolution, by identifying a dominant Betaproteobacteriales ASV in Family EC94 (~70–80%). Recently, predominant Gammaproteobacteria associating with several sponges including *C. crambe*, were proposed to take part of a new order named Ca. Tethybacterales, and in particular to the species Ca. *Beroebacter blanensis*, in *C. crambe* hosts from Spain[61]. Along this dominant symbiont, we found a moderate representation of cyanobacteria, and a high number of minor bacterial taxa. *C. crambe* specimens, despite hosting low microbial abundances, revealed values of ASVs richness comparable to those of HMA sponges, but with lower diversity. This outcome suggests that microbial densities and taxa richness are not always correlated[62], and highlights the relevance of the rare taxa to the overall diversity, especially in microbiota dominated by one or few symbionts (but see below)[63]. In LMA microbiomes dominated by one or few ASVs (as in *C. crambe*) dominant microbes are transmitted from adults to embryos with high fidelity thanks to brooding reproduction, as these symbionts are essential

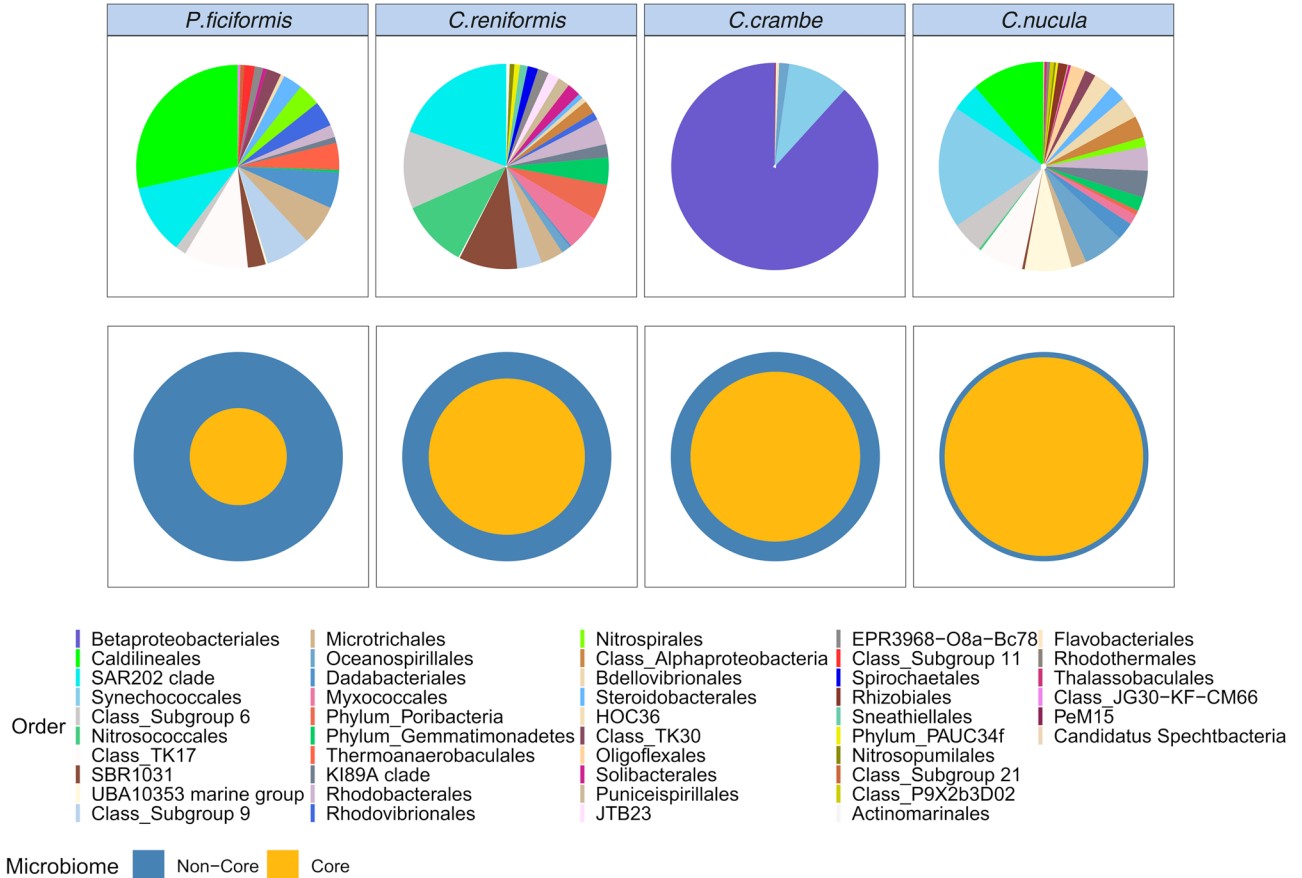

**Fig. 5 | Core microbiomes.** Upper pie charts show the core microbiome diversity (based on 100% shared AVSs within individuals of each sponge species). Lower pie charts show core microbiome size (%) of the four sponge species with respect to the total microbiome.

for fitness[25,64–66]. Instead, background taxa are mainly acquired horizontally from the water column[63,67]. These may include symbiont microbes retained with high fidelity, as the eight core taxa shared with the seawater recorded in our *C. crambe*, as well as other rare ASVs, some potentially representing transient microbes or food, accounting for up to ~175, in these hosts. The functional role of rare taxa is poorly understood, but they might be involved in host acclimatization processes. In this sense, background and/or newly introduced taxa may become operationally relevant, providing new metabolic pathways to face changing conditions, or replace lost functionalities during symbiont shuffling[68–70].

In general, LMA species have higher pumping rates than HMA sponges[71]. This entails filtering larger volumes of seawater and environmental microbes therein, enhancing the chances for transient microorganisms to enter and/or to establish[72–74]. However, there are cases of HMA sponges, like *P. ficiformis*, in which symbionts are largely acquired from the environment[25,54,75], along a likely minor fluctuant rate being transmitted within bacteriocytes to the oocytes[20,54]. This pattern is hypothesized to confer plasticity to colonize different biogeographical habitats[15,18,75]. On this line, Björk and collaborators[76] reported highly dynamic microbiomes in *P. ficiformis*, mostly dominated by temporally transient taxa. In our *P. ficiformis* specimens collected in the same day from several sampling sites, the prokaryotic assemblages displayed the highest intraspecific variability and the smallest core (microbiome) size. At the same time, this species harbored the richest and most diverse microbiome, dominated by phyla Chloroflexi and Gemmatimonadetes. Chloroflexi has been consistently found associated with *P. ficiformis* from diverse geographical locations and morphotypes, suggesting a co-evolutionary relationship between the sponge host and this microbial taxon[18]. Instead, Proteobacteria[77], and Gammaproteobacteria were predominant in other studies[18], again indicating the fluctuant microbial

communities even in the major phyla, in correlation with a diversity of growth forms (massive, reticular) described in these sponges. Within the genus Chloroflexi, Caldilineales, SAR202 clade and TK17 were the dominant orders in our sponges, being SAR202 clade commonly reported by other authors[15,18,75]. Older studies based on morphological approaches, described conspicuous presence of host specialized cells filled with vacuoles rich in cyanobacteria[32,78]. In our study, cyanobacteria (*i.e.* Synechococcales) displayed low relative abundances in *P. ficiformis* and were not a core taxon, which could indicate transitory partnerships. Low and variable presence of cyanobacteria have been similarly documented in recent surveys[18,24,77], suggesting minor tropic functional roles of this bacterial group, and potentially related to divergent growth morphologies, colouration or habitat preference[44,75,79] (see below).

Despite being widely investigated as a model organism, *C. reniformis* has received scarce attention regarding microbiome dynamics and composition[76,80,81]. Microbial assemblages in *C. reniformis* with pigmented morphotype investigated in the present study were mostly affiliated to the phyla Chloroflexi, Acidobacteria and Proteobacteria and exhibited rather low intraspecific variability. Such microbial steadiness could be linked to the uniformity in growth typologies under diversified habitat conditions (of light, depth, or other natural gradients). All the above may indicate that in *C. reniformis* there is a large proportion of bacteria transmitted in maternal bacteriocytes to the oocytes and related nursing cells[20,78].

The most prevailing phylum in *C. nucula* was Cyanobacteria (over 25% relative abundance), followed by Chloroflexi and Proteobacteria. The association of *C. nucula* with cyanobacteria was noticed in previous morphological studies, proposing them as key beneficial symbionts[82–84]. And on the basis of histological analyses, it has been demonstrated that cyanobacteria, different heterotrophic bacteria and yeasts can be acquired by vertical transmission[21,22]. In our study *C. nucula* displayed the highest intraspecific

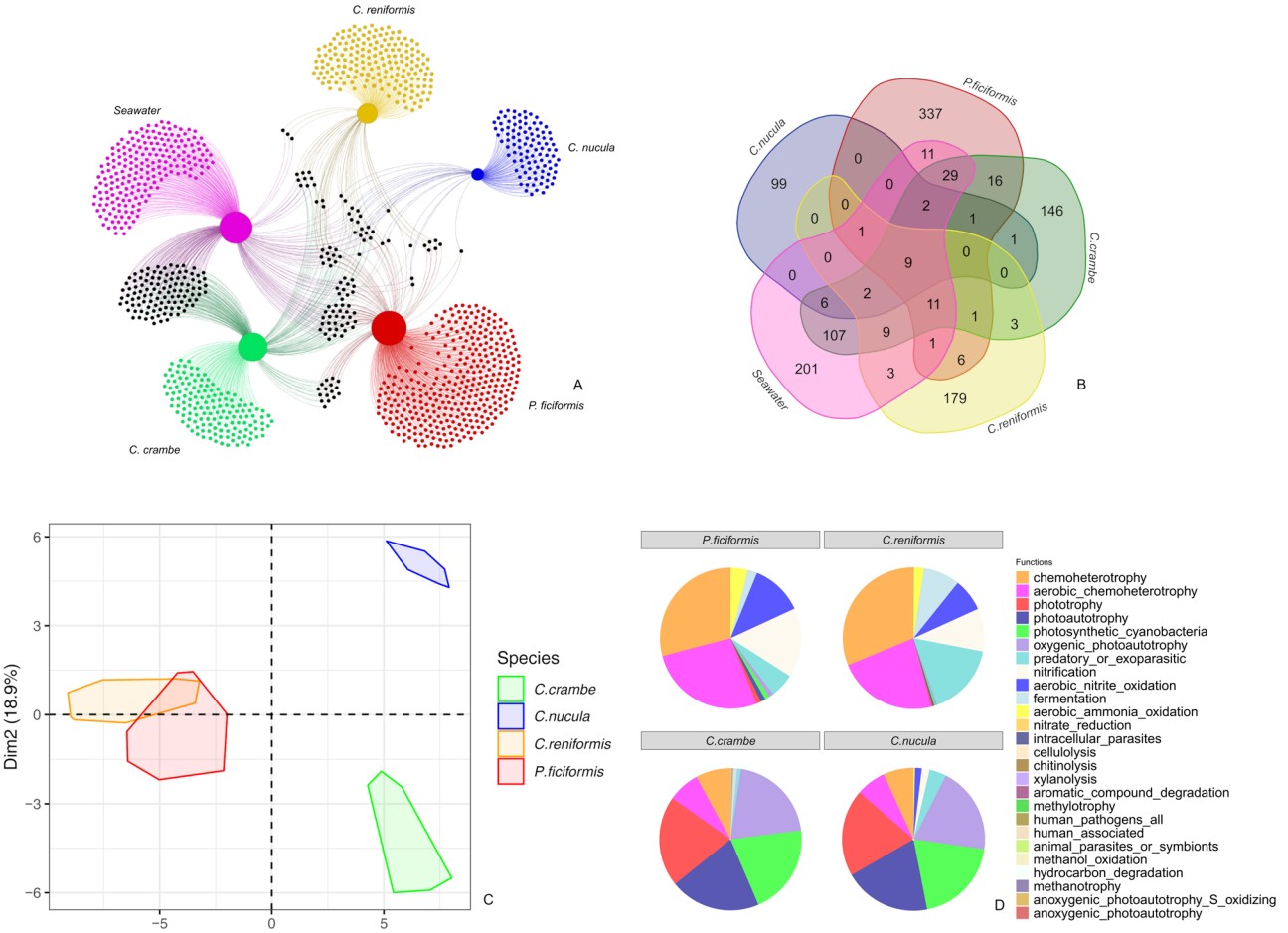

**Fig. 6 | Microbiome dynamics and functional profiling. A** Network showing bacterial ASVs shared between the four sponge species and the seawater. **B** Venn diagram showing the numbers of shared and exclusive ASVs among the four sponge species and the surrounding seawater. **C** PCA ordination of putative functions of microbiomes of each sponge species. Axes represent the first two components with the percentage of variation explained. **D** Putative functions of the microbiomes of the four sponge species.

similarity in microbiome composition, and the largest core microbiome in terms of both core ASV number and core size. A similar microbiome composition was reported using clone library sequencing on *C. nucula* collected in other locations of the Mediterranean[85] and on the congeneric *C. caribensis* from the Caribbean (previously identified as *C. nucula*). On this regard, both species are restricted to illuminated habitats and have identical lobulated "chicken liver" brownish to purple morphotypes[86], which may indicate a linkage between symbiosis and stable host morphology. All this information suggests that *C. nucula* harbors stable core bacterial assemblages across different biogeographic provinces, probably maintained via transgenerational transmission within oocytes.

The analysis of the microbiomes in the sponges investigated in the present study revealed a notable species-specific assemblage composition, which was also different from that of the surrounding seawater. Most of the prokaryotic taxa were exclusive of each sponge species and no shared ASVs were identified among the four species. Microbial assemblages chiefly modulated by host taxonomy have been previously reported in Porifera[15,34,39,87], even when sharing similar environmental conditions[17,18,46]. There are though cases of microbial shifts associated with sponge hosts living in different habitats[69], or in response to environmental factors[19]. Further studies, sampling diverse intraspecific phenotypes across diverse geographies and environmental scenarios should bring light on the phylosymbiosis, and level of within-species steadiness of microbial assemblages associated with our four target holobionts.

We identified a consortium of bacteria which were major drivers of the variability among sponge host microbiomes. In particular, taxa belonging to Subgroup 6 (Acidobacteria), SBR1031 (Chloroflexi), *Bdellovibrio* (Proteobacteria) were mainly associated to the HMA species (*C. reniformis*, *P. ficiformis* and *C. nucula*), whereas taxa belonging to Rhodothermaceae, Cyanobiaceae, PAUC34f and EC94 (Betaproteobacteriales) were mostly associated to *C. crambe*.

The bacterial taxon SBR1031 (Chloroflexi), mainly encountered in association with *C. reniformis* and *P. ficiformis*, belongs to the bacterial class Anaerolinae. It is involved in carbohydrate degradation and other degradation pathways of organic compounds (e.g. inositol[88];), thus allowing the utilization of different carbon sources, and likely linked to the capability of living an heterotrophic lifestyle in the absence of light. Subgroup 6 belongs to the phylum Acidobacteria, which is widespread in marine environments[89,90], and involved in the degradation of organic compounds and use of inorganic nutrients (i.e. sulfur and nitrogen), thus relevant for nutrient cycling[89,90]. The almost absence of this taxon in the *C. crambe* corroborates previous findings about the different sources of nitrogen compounds used by LMA sponges (mostly planktonic N) *vs.* HMA sponges (mostly nitrogenous waste)[13]. Within Proteobacteria, a dominant phylum in marine sponges[7,26], genus *Bdellovibrio* was correlated to HMA species. *Bdellovibrio*, due to their capability to prey on other bacterial taxa, can exert an important role on microbe-microbe interactions and community structure modulation in complex

**Fig. 7 | Metabolomics analyses. A** Metabolic profiles plotted by molecular classes in the four sponge species. Data were plotted by individual to show the intraspecific variability (for sample details see Table S1). **B** Network of chemical features associated with the four sponge species. **C** Network of presence/absence data of chemical classes associated with each sponge. **D** PCA ordination plot showing metabolomics feature data from the four different sponge species. Each dot represents a sample. Each arrow represents a class of metabolites.

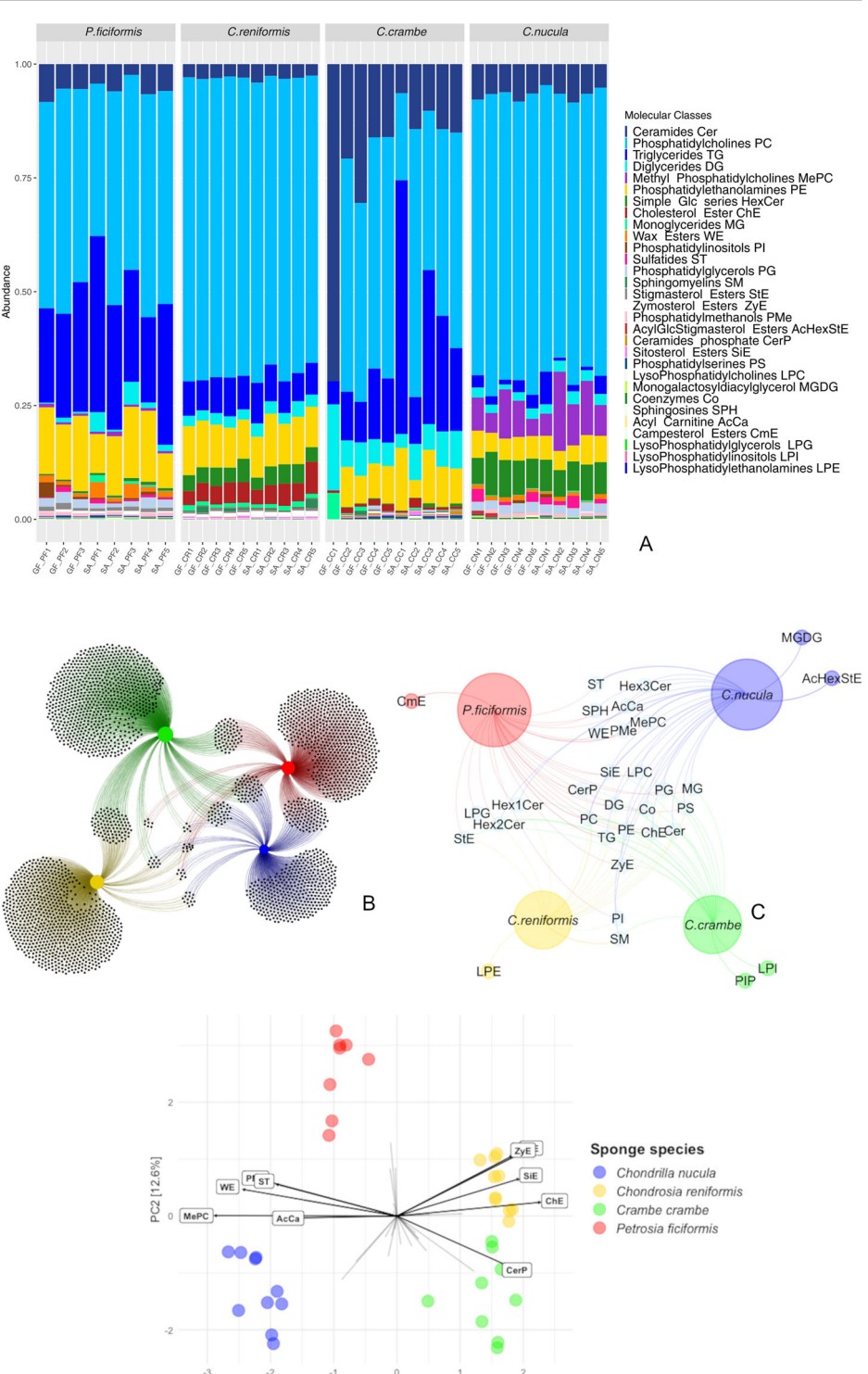

microbiota assemblages[91], thus potentially contributing to stabilize and explain differences among HMA microbiomes.

Among bacteria mostly associated with *C. crambe*, we found the EC94 group (Betaproteobacteriales), which has been previously reported as an important taxon in other LMA sponges from a variety of habitats of Antarctic ecosystems[92], and in corals[93]. The maintenance of EC94 (Ca. Tethybacterales[61]) group over evolutionary time, across generations and geographic locations, may suggest that these symbionts could play a major nutritional role in the heterotrophic metabolism of LMA sponges, despite the ecological significance of this taxon is still unknown[61,94].

Cumulative information on sponge microbiomes may allow soon to discriminate specific taxa driving co-divergence processes in Porifera, according to host taxonomies and/or habitat context.

Sponge microbiomes can be involved in a variety of processes including photosynthesis, methane oxidation, nitrification, nitrogen fixation, sulfate reduction, and dehalogenation[26]. Microbiomes of the sponges investigated showed putative functions mostly related to heterotrophic processes (e.g. chemoheterotrophy, aerobic heterotrophy) in *P. ficiformis* and *C. reniformis*, in accordance with the presence of different heterotrophic bacterial taxa (e.g. Chloroflexi), and the capability to be cave dweller sponges

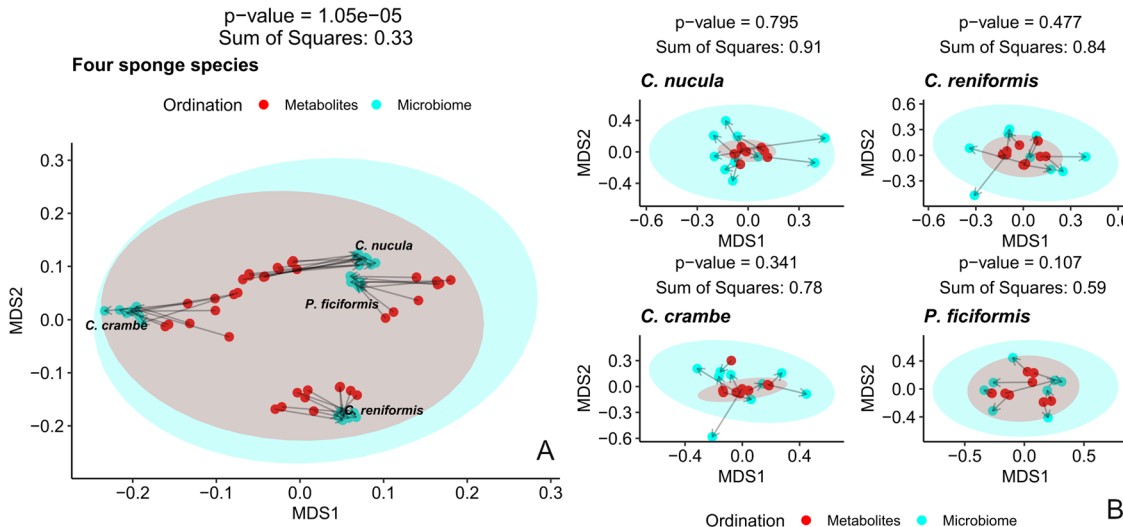

**Fig. 8 | Inter-omics Procrustes biplots comparing PCA ordinations of metabolite class profiles and microbiome genera based on centered log ratio (CLR) transformed data. A** Full data set including all four sponge species ($n = 37$) and **B** Sponge species panels: *Chondrilla nucula* ($n = 10$), *Chondrosia reniformis* ($n = 10$), *Crambe crambe* ($n = 9$), *Petrosia ficiformis* ($n = 8$). There was significant overall correlation between the microbiome and the metabolome when considering the full dataset comprising all four sponge species, however this association faded at the single species.

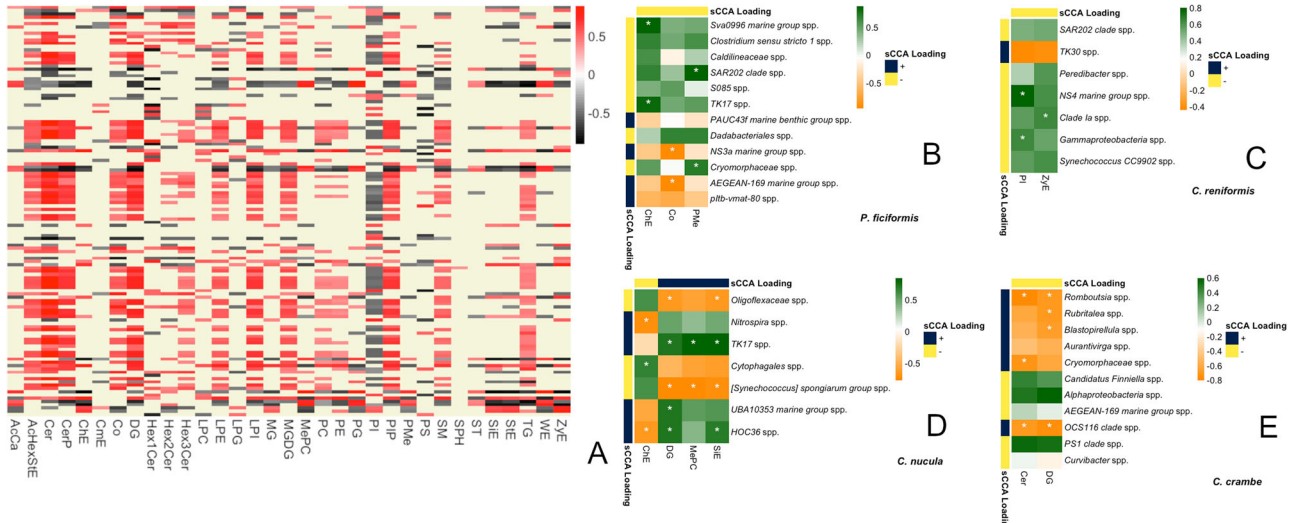

**Fig. 9 | Inter-omics Correlations. A** Heatmap depicting Total pairwise Spearman correlation of class metabolites and microbiome genera across the full dataset including all four sponge species (126 microbial genera, 33 metabolite classes, $n = 37$ sponge samples). **B-E** Panels display heatmaps restricted to microbial genera and metabolite classes selected by the sparse CCA procedure in the single sponge species, with significant Spearman correlations assigned by *: **B**: *P. ficiformis* ($n = 8$), **C**: *C. reniformis* ($n = 10$), **D**: *C. crambe* ($n = 9$), **E**: *C. nucula* ($n = 10$).

and live in dark conditions[44,79,95]. Conversely, the main putative functions of the microbiomes of *C. crambe* and *C. nucula* were related to autotrophic processes. These were mediated by the cyanobacterial component already documented[32,85,86,96], and identified in the present study, in correlation with the preference for illuminated habitats.

HMA and LMA sponges can display different functional traits, probably due to intrinsic factors, such as divergent pumping rates[50]. In this sense, the LMA species *C. crambe*, revealed a wide array of exclusive metabolic functions, which were probably related to the rare taxa acquired from the surrounding environment. Processes as nitrification and denitrification can be instead found in both sponge typologies[97]. In our study, nitrification related functions (nitrification, aerobic nitrite oxidation, ammonia oxidation) were more relevant in microbiomes of *P. ficiformis* and *C. reniformis* than in those of *C. nucula* and *C. crambe* (in this latter negligible), independently from the HMA and LMA dichotomy. Nonetheless, relevant

nitrification functional traits are normally due to a high presence of ammonia-oxidizing microbes in HMA hosts either heterotrophic or autotrophic[9,97-99]. Predatory or exoparasitic functions are related to predatory bacteria like Bdellovibrionales and Myxococcales, which were abundant in *C. reniformis*. These taxa feed on gram negative bacteria and can modulate microbial communities in sponges[100], likely contributing to the microbiome stability of this species[80]. Fermentation functions were preeminent in *C. reniformis*, likely due to the presence of members of the phylum Poribacteria[101]. Overall, these findings suggest the presence of a complex network of metabolic functions of microbiomes, whose relative importance changes in relation with the sponge species considered, and that may have different implications for the holobionts' fitness.

In the present study, we expand the current knowledge on compounds involved in the primary metabolism (i.e. lipids) of the sponge holobionts, being the majority of investigations focused on secondary

metabolites of biotechnological interests[102]. Previous findings carried out on *P. ficiformis* revealed the predominance of phospholipids (phosphatidyl ethanolamines PE, phosphatidylcholin PC, phosphatidylglycerols PG, phosphatidylserine PS and phosphatidylinositols PI), along with other constituents related to bacterial component (9-hexadecenoic, 15-methyl-9-hexadecenoic and l-octadecenoic acid)[103]. A single survey on *C. reniformis* highlighted the presence of 57 fatty acids, some of which with an odd number of carbon atoms putatively of bacterial origin[104], while for *C. nucula*, the only one published work revealed that 5,9,23-triacontatrienoic acid was the more relevant lipid compound[105]. In the present study, the four sponge species were characterized by the presence of a "core metabolome" (lipids shared among all samples) represented in order of relative importance by phosphatidylcholines (PC), triglycerides (TG), phosphatidylethanolamines (PE), ceramides (Cer) and diglycerides (DG). Phospholipids, such as phosphatidylcholine and phosphatidylethanolamine and others (e.g. phosphatidylserine, phosphatidylglycerin, phosphatidylinosite and diphosphatidylglycerin)[106], are key components of cellular membranes and their diversity in sponges has been proposed to contribute to the remarkable plasticity and acclimatization capability of this phylum[107]. Besides the shared lipids, we observed marked differences in metabolomics profiles of the different sponge holobionts, likely due to a combination of specific compounds characterizing the host and its microbiome. Indeed, in our samples, certain lipids were at times shared among some of the investigated holobionts or were exclusive of a single species. Phosphatidylinositols (PI), for instance, were found in all sponges except in *C. crambe*. High concentrations of phosphatidylinositols and phosphatidylglycerols are indicators of the presence of symbiotic bacteria[107]. Thus, the lack of phopsphatidylinositols signal in *C. crambe* could be related to the low microbial abundances characterizing this species (LMA). Lysophosphatidylglycerols are also integrant lipids of bacterial membranes[108], and were detected only in *P. ficiformis* and *C. reniformis*, suggesting the presence of particular large groups of bacteria shared between these two species (e.g. SAR202 clade). One characteristic compound type exclusive of *C. nucula* were the monogalactosyldiacylglycerols (MGDG), which are ubiquitous lipids in the thylakoids of cyanobacteria[109]. These findings agree with the intimate partnership established between *C. nucula* and these photosymbionts[32,85,86,96], and are further supported by our microbiome results, revealing cyanobacteria as the dominant phylum in this species. Another exclusive class of sterols were the campesterols (CmE), only found in *P. ficiformis* specimens, and considered specific markers of genus *Petrosia*[110,111]. Finally, sphingomyelins, which are sphingolipids of animal cell membranes related to the myelin sheath of nerve cells[112] and never found before in sponges[107], were detected in all the sponges' samples investigated with the exception of *P. ficiformis*.

Overall, findings reported in the present study indicate that some lipids, such as those constituents of cellular membranes (e.g. phosphatidylcholine and phosphatidylethanolamine) or involved in energy storage (e.g. triglycerides), are ubiquitously present in the investigated sponges independent of their taxonomy, and derived from the host cells and/or microbes providing consonant pathways. While a wide variety of other compounds appear to be species-dependent and related to specific microbial components.

Our correlations analysis based on multi-omic profiling via paired targeted sequencing of the 16 S rRNA gene and LC-MS metabolomics revealed high level of global congruence across all four species. Most microbes in the overall dataset were involved in demarcating metabolic signatures representing each of the host species. However, at the single species subsets inter-omic concordance was weak, with few microbial genera displaying relevant correlations with a reduced number of class metabolites. The vanishment in microbial–metabolic correlation signal within host species, may reflect a substantial functional redundancy, already reported in sponge microbiomes ([113–115]). On top of this, technical aspects in our approach could be masking microbiome-metabolome coupling cues. On the one hand, probable infra-generic intra strain diverseness might not be captured with taxonomical classification based on amplicon sequencing in place of whole genomes[116]; whereas on the other, the metabolite classes used might be too generic and hence poorly resolutive[117]. We are enthusiastic that both perspectives will improve as microbial, and particularly metabolomic annotations, will become more accessible in the reference databases.

Results recognized some degree of difference in microbiome-metabolome coupling across species. *Chondrosia reniformis* had significantly lower canonical correlation than *P. ficiformis* and *C. crambe*, and also than *C. nucula*, but against this last one the difference was not significant. This could indicate higher functional redundancy and metabolic stability in *C. reniformis'* promoted by a diverse and constant microbiome. This is in agreement with previous work reporting microbial stability from specimens collected in several natural habitats and from laboratory experimentations[80,81]. In contrast, *P. ficiformis* seems to rely on more fluctuant microbial communities modulating the metabolome, where large inputs of environmental symbionts of potentially different functionalities may narrow functional redundancy[48]. There is scarce previous information about the other two species on this topic. But, in the case of *C. crambe* the functional rare biosphere, mostly acquired by horizontal transmission, may as in *P. ficiformis* uphold more shifting metabolic signatures, yielding increased correlation signal.

Small clusters of core microbes and metabolites with non-zero loading coefficients in sCCA and significant pairwise Spearman ranks, were able to explain intraspecific interactions within each host. All the selected microbial taxa exhibiting significant metabolic coupling in *C. crambe* were negatively correlated with either Cer or DG. Cer and DG are structural lipids of the bilayered membrane of eukaryotic cells controlling its permeability, and acting as signaling molecules in immune responses[118,119]. Correlation with microbial groups suggest mechanisms of symbiont recognition[120]. Previous studies in anemones detected positive intensities of Cer and DG with the presence of intracellular symbionts[120]. Attending to our data and previous work[59], negative correlations found in *C. crambe* could be related to sparse microorganisms living in the mesohyl. In *C. nucula* negative associations with DG seem to co-occur also with extracellular symbionts. These include Oligoflexaceae, an extracellular partner taxon in corals[121], and *Synechoccocus spongiarium*[122], which had also negative association with MePC, phosphatidylcholine derivatives typical of eukaryote membranes, and reported in intracellular bacteria to avoid host immune responses[123]. Instead, HCO36, UBA10353 and TK17 microbes were positively related with DG, being TK17 also correlated with MePC. We cannot, though, ascribe these taxa as intracellular symbionts with the available information (see[88]). Interestingly, *Synechococcus spongiarium* (Cyanobacteria) and TK17 (Chloroflexi) revealed opposite correlations with the same class metabolites, including phytosterol-like SiE, which could indicate a functional interplay. Actually, fermentation coupling between Cyanobacteria and Chloroflexi as oxygenic and anoxygenic phototrophs respectively were demonstrated across hypersaline microbial mats[124]. Several cholesterol derivatives revealed microbial correlations, as such ChE with described antibacterial properties[125] interacting with several taxa in *C. nucula* and *P. ficiformis*, and ZyE, which was associated to Clade Ia in *C. reniformis*. Finally, cytosolic membrane related PI compounds exerted positive interactions with NS4 and selected Gammaproteobacteria microbes in *C. reniformis*. We can conclude that most metabolite classes captured in the correlation approaches were cell membrane affiliated components, presumably involved in host-symbiont recognition mechanisms. It is highly probable that future advances in microbial and metabolic analytics will retrieve other functional interactions. This will allow us to understand how sponge microbiomes participate in the multifactorial metabolic pathways, leading to intraspecific phenotypic diversification and species acclimatization to challenging conditions.

Our study performed on samples of a single phenotype appertaining to four different target species, allowed us to encounter trends in the microbiomes, metabolomes and/or their inter-omics interactions that seem to

correlate with the potentiality of range traits for each sponge. *Crambe crambe*, is a LMA species displaying a large core microbiome dominated by a single symbiont, of vertical brooding acquisition. The presence of a notable arsenal of secondary metabolites, and a moderate representation of cyanobacteria and other rare taxa of likely environmental origin may allow it to colonize habitats with wide-range of light exposure. *Petrosia ficiformis* has the most variable microbiome and displayed the smallest core microbiome. This variability in the microbiome (mainly acquired horizontally), together with the plasticity in the growth typologies is likely to contribute to the colonization of different habitats (e.g. marine caves, shaded habitats at different depths) and to the production of a diversified group of secondary metabolites.

*Chondrosia reniformis* is a HMA species with a stable microbiome. Its growth typology is massive, with a certain variability in the pigmentation, being independent from its associated microbial communities. Such pigmentation variability coupled with a large and diverse core microbiome, and the characteristic to reproduce asexually by budding, could be the key for this species to spread in different habitats (caves, coralligenous, shaded or dark habitats).

*Chondrilla nucula* belongs to the HMA sponge group and has a large and diversified core microbiome. The characteristic to vertically transmit most of its microbiome to the offspring might be crucial for this species to be ubiquitous in shallow environments. But may also delay colonization in new habitats with different conditions, to which parental symbionts might not be acclimatized. Additionally, its strict trophic relationship with cyanobacteria constraints this sponge to colonize mainly photophilous habitats, with rare cases of rachitic individuals, found in shaded conditions.

Correlations outcomes indicate an overall relatedness between microbial genera and metabolite classes in the full dataset comprising all four sponge species. But at the single species, this degree of coherence vanishes. This in part could be explained by the strong functional redundancies of sponge microbiomes sustaining homeostasis, in particular in such hosts, constantly exposed to multitude and dynamic microbial partnerships. But furthermore, the metabolite classes analyzed might not be illustrative of the intricate coupling between microbiomes and metabolomes, for being somewhat structural or inclusive, probably holding relatedness with many taxa concurrently. It is probable that more specific compound arrays may yield fitter correlations. Further, the use whole genomes instead of partial 16 S rRNA amplicons for accurate microbial taxonomic characterization may have resolved potential strain level variabilities.

Sponges co-existing within local community assemblages are likely to exhibit diverse microbiome networks and metabolic profiles that promote niche diversification, but also analogous phenotypic patterns of "symbiont evolutionary convergence", yielding holobionts with different bio-ecological traits. Our investigations supply new documentation in the interconnection of multi-omic approaches applied to marine holobionts, and propose their suitability to study phenotypes diversification in the face of changing environments and acclimatization processes.

## Methods
### Sponge collection and processing
All the samples in the same species collected for our study were of the same phenotype, this is: pigmented and massive for *P. ficiformis* and *C. reniformis*, pigmented and lobular for *C. nucula* and encrusting for *C. crambe*, and all from illuminated habitats, with analogous conditions as specified in Fig. 1.

Sponge pieces (~10 cm3) of different target species individuals *Petrosia ficiformis* (n = 12), *Crambe crambe* (n = 14), *Chondrilla nucula* (n = 10), and *Chondrosia reniformis* (n = 15) were collected along with ambient seawater (n = 9) in spring 2018 by scuba diving at 2–5 m depth around the island of Ischia, (Southern Tyrrhenian, Mediterranean Sea, Italy), where all species are highly abundant[126,127]. Sponges were recovered from three nearby sites with similar conditions: Castello Aragonese (CCO; 40°43'55.9"N–13°57'52.9"E), Grotta Mago external (GF; 40°42'41.6"N–13°57'51.4"E) and Sant'Anna

(SA; 40°43'36.5"N–13°57'43.4"E) (Fig. S1). The sampling sites were accessed with local authorization from the marine protected area: Area Marina Protetta Regno di Nettuno (Ischia).

Samples were subdivided and preserved as explained in the Supplementary Material and Methods (SM, section 1.0) for microscope, microbiome and metabolomic analyses.

### Transmission electron microscope (TEM) observations
After sample preparation (see SM, section 1.1), ultra-thin sections obtained using an ultra-microtome (Leica, UCT Ultracut Microtome) were observed by TEM (Zeiss, LEO 912 AB).

### Microbial profiling based on 16 S rRNA targeted gene sequencing
DNA from all four species (n = 51 total sponge samples) and from seawater filters (n = 9) obtained after filtration was extracted using the DNeasy PowerSoil Pro Kit (Qiagen), following manufacturer's instructions. For prokaryotic diversity analysis, the $V_3$-$V_4$ hypervariable region of the 16 S rRNA gene (*Escherichia coli* position: 341-805) was amplified using the primer pairs Bakt_341F 5′-CCTACGGGNGGCWGCAG-3′ and Bakt_805R 5′-GACTACHVGGGTATCTAATCC-3′[128]. These primers are specific for bacterial communities and may reveal lower capture of Archaea communities[128]. Sequencing was performed on an Illumina MiSeq sequencer with a Reagent Kit v3 (600-cycle) targeting ~400–450 bp amplification products.

Demultiplexed paired end sequence reads were imported into quantitative insights into microbial ecology pipeline (QIIME2) v.2019.10[129]. Sequences were filtered and denoised with DADA2[130] to obtain "Amplicon Sequence Variants" (ASVs) feature frequency table. Taxonomy was assigned using a pre-trained Naïve Bayes classifier (sklearn)[131] against the 16 S SILVA (99% identity) reference database v.128[132] (details in SM, sections 1.2–1.3).

### Metabolomics profiling based on untargeted LC-MS
For each sponge specimen (*P. ficiformis* n = 8, *C. reniformis* n = 10, *C. crambe* n = 9, *C. nucula* n = 10), ca. 500 mg of freeze-dried material was crushed and sonicated with 4.5 mL methanol (MeOH) for 30 s. Then, 15 mL of methyl tert-butyl ether (MTBE) were added and left 1 h at room temperature under shaking. Upon addition of 3.75 mL of chromatography water, the sample was mixed and then centrifuged to induce phase separation[133]. The upper organic phase was recovered, dried and kept in glass vials at -20 °C until use. Dry extracts were resuspended and injected in a Thermo Scientific Dionex UltiMate 3000 chromatograph coupled to a Thermo Scientific Q Exactive focus quadrupole-Orbitrap mass spectrometer (UHPLC-q-Orbitrap, Thermo Fisher Scientific, MA, USA) operated in full scan-data dependent MS2 (Full MS-ddMS2) discovery acquisition mode. Further details about UHPLC-q-Orbitrap configuration are reported in section 1.4 of the SM.

### Statistics and reproducibility
Sponge samples from single individuals were collected, along with ambient seawater, in the following number of replicates: *Petrosia ficiformis* (n = 12), *Crambe crambe* (n = 14), *Chondrilla nucula* (n = 10), and *Chondrosia reniformis* (n = 15), seawater (n = 9). Statistical tests (Kruskal-Wallis, Wilcoxon rank sum test, PERMANOVA, Welch's test; see below) were used to detect significant differences on alpha and beta diversity and were run in the R environment. Diversity comparisons based on binary and relative abundance transformed data and on center log transformed (CLR) data for differential abundance for compositional data were calculated applying QIIME2 plugins and R packages (for details see SM section 1.4). LipidSearch v4.0 (Thermo Fisher Scientific) and several R packages were used for metabolomics data analysis. Similarity Percentages (SIMPER) on network patterns were statistically evaluated with PRIMER[134] (section 1.5, SM).

## Microbial community analysis

Diversity analyses were run with core-metrics-phylogenetics command on rarefied data for alpha-diversity (within sample diversity) estimates using Observed ASVs and Shannon Indexes[135] and using Bray Curtis and Jaccard distance matrixes for beta-diversity[136,137]. Classical methods based on relative abundance (rarefaction, alpha and beta diversity, taxa barplots) can lead to misleading results, in which specific features individually carry no trustful information for the absolute abundance[138].

In order to address this potential flaw of compositional data, our analysis was complemented with an Aitchison distance based approach for diversity analysis, that can handle high levels of sparsity and zero values. The resulting distance matrix was plotted in ordination robust compositional principal component analysis (RPCA) biplots[139], following q2-DEICODE plugin (https://library.qiime2.org/plugins/deicode/19). Input data in DEICODE pipeline, works on non-rarefied tables in a non-supervised method (no co-variates included in the formula computing the model).

This tool has been previously used in order to analyze compositional datasets both in biomedical and ecological studies[140,141]. Data for this analysis were log transformed and centered to obtain center log-ratios (clr), a transformation widely applied for compositional data investigations[142].

Log ratios of the most important features driving the differences in the ordination space were visualized and plotted with Qurro interface to detect sample groupings[143] (https://github.com/biocore/qurro). Qurro calculates log-ratios between any chosen sets of numerator features and denominator features. This is achieved by computing, for each sample, the log-ratio between the total counts of the selected numerator features (numerator ASVs) and the selected denominator features (denominator ASVs)[143]. Certain consortia of features (microbial taxa) could be identified as major drivers (high ranked) of sample type grouping along a selected ordination axis of the RPCA biplot. The resulting selected numerator and denominator differential taxa were then used to generate/calculate new log-ratios of the most explanatory microbial balances separating sample types.

Kruskal-Wallis tests for overall tests and pairwise Wilcoxon rank sum tests for multilevel comparisons were applied for contrasting alpha-diversity indexes. Permutational multivariate analysis of variance (PERMANOVA) was used to detect significant differences on beta diversity based on resemblance matrices. Pairwise Adonis[144,145] (https://github.com/pmartinezarbizu/pairwiseAdonis) was used to perform multilevel comparisons. Welch's Tests were applied to test the statistical significance of log ratios based on groupings from the differential microbe consortia selected in Qurro. Core microbiome and core size were calculated in R (packages "Microbiome", "Phylosmith" and "Phyloseq") considering ASVs 100% present across groups[146–148].

Similarity percentage analysis (SIMPER)[149] was run on PRIMER 6 + [134] to calculate turnover diversity and visualize the similarity within and among sample groups. Exclusive and shared ASVs were counted using the CONTA.SE command in Microsoft Excel Software. Network analyses were computed and visualized on Gephi v0.9.2[150]. FAPROTAX v1.2.4 was used as a database for the predicted functions annotations[53], keeping in mind though, its limitations due to the taxonomy resolution[151]. R Studio (v1.2.5033) was used to perform statistical analyses and to generate plots with specific packages (Qiime2R, Phyloseq, Vegan, Ggplot2 R packages)[148,152,153].

## Metabolomics data analysis

LipidSearch v4.0 (Thermo Fisher Scientific) was used to process the UHPLC-q-Orbitrap data in non-targeted mode, to identify metabolomic features grouped as molecule class profiles. Class metabolites were represented in Principal component analysis (PCA) ordination from CLR transformed data in R (microbiome, MicroViz packages[146,154]). PERMANOVA was used to detect significant differences between the sponge species. Pairwise Adonis was used to perform multilevel comparisons (vegan R package)[144,145]. Networks of single compounds and molecular classes across sponge species were visualized and computed by Gephi v0.9.2[150]. Similarity percentage analysis (SIMPER) (Clarke, 1993) was run on PRIMER 6+

software[134]. Metabolomics data were deposited on the EMBL-EBI MetaboLights database[155].

## Microbiome and metabolome correlation patterns

For our correlations, we filtered out those ASVs in the microbiome data with lower 10% prevalence across all samples[156,157]. Metabolomics features were clustered into 33 metabolite classes. To deal with issues of data compositionality, both, the microbiome and metabolomics sets were centered log ratio transformed (CLR)[158], applying a previous pseudocount of min(relative abundance)/2 (microbiome R package)[146]. CLR transformation over performs with respect to other transformations in addressing spurious correlations, subcompositional incoherence and lack of isometry, while maintaining scale invariance and straightforward interpretation[159,160]. Microbial ASVs were agglomerated to the genus level[157], resulting in 126 genera across all four sponge species (full data set), and 54, 52, 71 and 77 genera for correlations subsets corresponding to *C. nucula* (n = 10), *C. reniformis* (n = 10), *C. crambe* (n = 9) and *P. ficiformis* (n = 8) respectively.

## Global concordance based on procrustes testing

Principal component analysis (PCA) is an unconstrained method that does not use distance matrix, with the additional benefit that the feature variables (microbes, metabolites) maintain their "loadings" for plotting in the ordination. PCA were performed directly on our CLR transformed microbial and metabolome variables using the rda function from the vegan package in R[152]. Paired microbiome and metabolomics PCAs, corresponding to the full data set (four species) and for each of the sponge species subsets, were compared based on the first two axes via symmetric Procrustes (protest function, vegan package[152]). Procrustes procedure measures the degree of concordance between two ordinations normalized to unit variance, by conducting a superimposition fit, in which coordinates are translated and rotated, maintaining the overall data structure. Significance is calculated according to the sum-of-squared differences (m2) against the permuted null (999 permutations). For ecology applications, procrustes has demonstrated to be more powerful than the Mantel test, while reducing type I error[161].

## Sparse canonical correlation (sCCA) analyses and pairwise spearman correlations

Pairwise Spearman correlations were conducted to obtain overall correlations patterns between metabolite classes and microbial genera in the overall data set, and further combined sparse canonical correlation analysis (sCCA) to identify closer clustered associations within each of the sponge species.

Pairwise Spearman correlations, with multiple hypothesis testing adjustment applying the Benjamini-Hochberg correction (false discovery rate, FDR, controlled q-values) method were calculated integrating psych::corr.test and the calculateCorForTwoMatrices function in R (packages psych and multiOmixViz[162,163]), setting as cutoff for significance pairwise correlations an FDR q-value of 0.05.

sCCA is a very useful tool to biologically integrate multi 'omics datasets, and select explanatory sets of features associations (e.g. microbe–metabolite). sCCA performs informative variable selection by calculating linear combinations of maximum correlation among the query data matrices, and bounding variable individual weights to generate sparsity. Based on L1-penalized matrix decomposition method of cross-product matrix akin to a LASSO regression problem[157,164,165], we calculated the overall multivariate associations given by the first canonical variate correlation coefficient. With this procedure, variables were selected based on their importance to the overall covariance between microbiome and metabolome datasets. CCA.permute (nperms = 50) was used for model fitting, with nboot = 1000 bootstrapped 95% confidence interval and nperm = 1000 permutation test setting, at the 0.05 significance level (PMA R package[164]). Function randomizeMatrix from R package picante[166] preserved data structure across the different permutations. Outcomes of Spearman

correlations and selected sCCA associations were combined in heatmaps for each species subset with pheatmap package in R[167].

## Reporting summary

Further information on research design is available in the Nature Portfolio Reporting Summary linked to this article.

## Data availability

Statistical outputs, Supplementary Figures. and tables can be found within the Supplementary Information file. Numerical source data for Fig. 5 and Fig. 8 can be found respectively within Supplementary Data 1 Table S8 and Supplementary Data 2 Table S10. Numerical source data for boxplots were provided in the Supplementary Data 3 file. Sequencing data for this study can be found under the Bioproject ID PRJNA851757 and can be accessed here: https://www.ncbi.nlm.nih.gov/bioproject/851757. The metagenomic reads for the 16 S $V_3$-$V_4$ region are available at NCBI under the accession numbers SAMN29249576-SAMN29249593, SAMN29249594-SAMN29249614, SAMN29249648-SAMN29249669. Metabolomics data have been deposited to the EMBL-EBI MetaboLights database (https://doi.org/10.1093/nar/gkz1019, PMID:31691833) with the identifier MTBLS5125. The complete dataset can be accessed here: https://www.ebi.ac.uk/metabolights/MTBLS5125.

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

## Acknowledgements
Thanks are due to Mr. Bruno Iacono for his help during the sampling phase of this study. We also thank the technicians from RIMAR microscopy group of the Stazione Zoologica: Franco Iamunno, Rita Graziano and Giampiero Lanzotti. Thanks are due to Dr. Maria Cristina Gambi for sharing important knowledge on the sampling area. We deeply thank Dr. Maria Costantini for her tips and for helping during the sampling and the laboratory activity. Authors deeply thank Mr. Pasquale Vassallo for providing underwater picture 1E. V. Mazzella has been supported by a PhD fellowship co-funded by the Stazione Zoologica Anton Dohrn and Polytechnic University of Marche (PhD course in Life and Environmental Sciences, XXXIII cycle). V. Mazzella was also supported by Assemble Plus transnational acces (Project n° 8437: 'MetMetMetAOA': Metabolic Meta-analysis in Metaorganisms adapted to Ocean Acidification) and EMBRIC (European Marine Biological Research Infrastructure Cluster) transnational access – (Project n° 5380, 'MicroMetOAc': Microbiome shifts and Metabolic production under Ocean Acidification). Graphical abstract was created with BioRender.com. Map in Fig. S1 was created using the Free and Open Source QGIS. Project was partially funded under the National Recovery and Resilience Plan (NRRP), Mission 4 Component 2 Investment 1.4 - Call for tender No. 3138 of 16 December 2021, rectified by Decree n.3175 of 18 December 2021 of Italian Ministry of University and Research funded by the European Union—NextGenerationEU: Award Number: Project code CN_00000033, Concession Decree No. 1034 of 17 June 2022 adopted by the Italian Ministry of University and Research, CUP C63C22000520001, Project title "National Biodiversity Future Center – NBFC.

## Author contributions
Conceptualization: L.N.P., A.D., V.M.; Methodology: V.M., N.E., B.G.G., G.N., L.N.P.; Investigation: V.M., A.D., N.E., B.G.G., G.N., A.F., L.N.P.; Data curation and visualization: V.M., L.N.P., N.E.; Original draft preparation: V.M., L.N.P.; Review and editing: V.M., A.D., N.E., B.G.G., G.N., A.F., L.N.P.; Supervision: L.N.P., A.D., N.E., A.F.; Project administration: L.N.P., A.D.; Funding acquisition: L.N.P., A.D. All authors have read and agreed to the final version of the manuscript.

## Competing interests
The authors declare no competing interests.
