## [Peer Review File · Communications Biology]

Reviewers' comments:

Reviewer #1 (Remarks to the Author):

General Comments:

Dr. Núñez-Pons and colleagues have presented their study investigating the functional relationships between microbiomes and metabolic signatures in coexisting Mediterranean sponges. The research uncovers distinct patterns among various sponge species concerning their microbiomes, growth typologies, and ecological adaptations. The findings suggest that diverse microbiome networks play a role in niche diversification and highlight analogous patterns of "symbiont evolutionary convergence" shaping their bio-ecological traits. Here are some specific suggestions to enhance the manuscript's clarity and impact:

1. In the introduction, the authors rightly identified the lack of integration between microbiome and metabolomics datasets as a gap in current knowledge. They effectively describe these datasets separately. However, I recommend conducting further analysis to establish connections between microbiome and metabolome profiles. It would be beneficial to include a table summarizing commonalities and differences in clustering patterns of microbiome diversity, microbiome functional predictions, and holobiont metabolome. Additionally, providing 2-3 examples linking predicted genomic functions of the microbiome with host metabolic signatures would strengthen the argument.
2. Since the metabolome is based on the holobiont, and the study compares both HMA and LMA sponges, it would be valuable to discuss how much of the differences in metabolic profiles are attributed to the microbiome versus the host. Does the microbiome contribute to divergence or convergence in the holobiont metabolome?
3. More details about the methods used in the DEICODE results should be provided. Specifically, clarify the meanings of log ratios, denominators, and numerators to aid reader understanding.
4. Ensure that figure panels have appropriate labels for clarity.

Specific Comments:

1. Line 159: Please explain why there are more DNA samples collected than sponge samples.
2. Line 281: This sentence requires clarification. Please provide additional information or context to make it more understandable.

Overall, the manuscript presents an intriguing study that describes microbiome characterization and metabolomics to explore functional relationships in Mediterranean sponges. Addressing the points outlined above would further enhance the clarity and potential of the study to engage a wider scientific audience.

Reviewer #2 (Remarks to the Author):

The work of Valerio Mazzella and coauthors, entitled "High microbiome and metabolome diversification in coexisting sponges with different bio-ecological traits" describes the microbiome and metabolome of 4 sponge species plus water in 3 locations of the Mediterranean Sea. It also includes identification of bacteriocytes and predicted functions. While the work is well performed and clearly written, there are several issues that need clarification.

My main concern is the choice of samples. From the introduction, the authors explain that some of the species present morphotypes (pigmented, bleached, lobular, reticular, etc) that could be related to host-microbiome-metabolome interactions (Line 109). Fig. 1 shows these morphotypes among the individuals considered in this study, however there is no table specifying what morphotype presented each sample. Location is also considered as a factor of variability, because there are illuminated areas and caves among the sampling site.

Any of these factors (morphotype and location) are considered and analysed separately for any of the species. All individuals are pulled together and treated as simply replicates throughout the study. If the goal was not to see differences in these morphotypes, I am not sure why the authors took a mix of samples instead of similar morphotypes.

I personally think that the differentiation of the morphotypes, together with the metabolome, would be the main interest of the work, since the sponge species themselves have been studied previously in many other locations, and it is not novel. These factors may also be behind the larger or smaller infraspecific differences of each species. For instance, we don't know whether the largest differences in *P. Petrosia* are related with their morphotypes. The authors should investigate a bit more whether differences among phenotypes or locations are worth mentioning, or whether it is ok to pull samples together.

Another two general points I don't personally agree with are: 1. the description of rare taxa in LMA sponges as part of the microbiome diversity. I believe it can just be food/passing bacteria (see specific comments below). 2. the presence of bacteriocytes in *C. reniformis*. In the microscopy images selected I definitely don't see that, and the references used to support that claim are in fact not correct (see comments below), so I would need a better image or proof to accept this finding.

In my opinion this work is interesting for the scientific community, but it could be greatly improved by checking and clarifying some of the issues I raised. Therefore I suggest acceptance after major changes are made.

Specific comments

Line 41-42. Could be contaminants from water in channels, food or passing bacteria.

Line 46. Change to: "According to putative functions, the microbiome of *P. ficiformis* and *C. reniformis* were mostly functionally heterotrophic."

Fig. 1 Legend: Also add "(photic zone)" for the preferred habitat of *c. crambe*.

Table S10. It is called Table S9 in the download area. Correct.

Line 88. *C. reniformis* is used as example of both via horizontal acquisition and vertical transmission. I would leave *P. ficiformis* as horizontal acquisition as this species does not present any vertical transmission, and the other 3 as example of VT (possible combined with HA of course).

Line 95. Why is "(endo)symbionts" specified here. Symbionts living outside host cells, do the same functions.

Line 108. This sentence should follow the previous with a comma (,). Eg. "While ..., others ..."

Line 144. It took me a while to understand the numbers in Line 159, since not all species were collected in all 3 locations. I think that should be clarified here.

Line 159. Why are not 15 samples for *P. ficiformis* and *C. Crambe*?? It should be explained whether the authors collected 5 individuals in each location or not. Or whether the extractions failed for some of them and why. But numbers are not clear at the moment (same for the water samples).

Fig. 3. I am not aware of *C. reniformis* presenting bacteriocytes, but definitely in those images I don't see any. I only see a host cell doing some phagocytosis, or cell extensions (same in Supp. Fig. S2)

Line 200. I would need a better prove of the bacteriocytes in *C. reniformis*.

Line 203. I strongly believe both *C. crambe* and *C. reniformis* present absence of bacteriocytes.

Line 214. I wouldn't say the pattern of reads is the same as the Shannon. *C. nucula* had low number of reads, but its diversity is higher, as expected for a HMA.

Line 223. I have not seen this deicode biplot before, and I don't understand which sponge species are defined by the arrows pointing to the left?? I also don't know where to find the most significant ASVs driving sample grouping, or the meaning of the Log-Ratio or numerator/denominator. Maybe the authors need to explain in more detail the meaning, procedure and outcome of this analysis.

Fig. 5. In this figure, low abundant groups could be removed, so there are not so many colours together. It is hard to distinguish anything.

Line 241. Archaea was definitely in low percentage since your primer choice does not amplify archaea. *C. nucula*, *C. reniformis*, and *C. Crambe* usually present low abundances (less than 4%) but *P. ficiformis* can harbour more than 20% relative abundance of archaea members. You could state this primer bias in methods, and don't worry about stressing here the low presence of archaea for each species.

Line 273. I got lost with this numbers. If *C. nucula* had 121 ASVs (line 213), and 55 were in the core (line 270), doesn't it make a core of 45%? Where is the 95% coming from?

Line 276. In the figure you can see 337 exclusive ASVs no 339, please check.

Line 344. References 10 and 52 does not mention any bacteriocytes, they are studies regarding HMA-LMA dichotomy. The 69 (Sara 1998) does mention "the follicular bacteriocytes enveloping the egg of *Chondrosia reniformis*" from the work of Levi and Levi 1976, but I think this is a bad translation of the original paper, which only mentions follicular cells around the oocytes. The follicular cells that form around the oocytes are not the same as the bacteriocytes containing bacteria in the mesohyl. Please, provide better references for this statement.

Line 352. Richness of *C. crambe* may have been similar, but diversity was clearly lower, which I believe is more important. *Crambe* probably have lots of the rare taxa contributing to the overall richness, but most likely, as LMA sponge, they are just passing bacteria from the seawater, not even symbionts.

Line 359. I agree that a large part of the microbiome can be acquired from seawater (horizontal), but the simple presence of seawater bacteria in a LMA sponge is not sufficient for this affirmation, mostly if it is only part of the rare microbiome. I think this would need additional information on how abundant were those ASVs in the water and in *C. Crambe*, whether they are part of the species core, etc. There should be some kind of enrichment from the sponge side to consider them HA symbiont and not just passing/food bacteria. See DOI:10.1038/s41598-018-33545-1.

Line 430. I definitely do not understand this analysis and plot. Rhodothermaceae and PAUC34f points to *C. nucula* in the plot. I don't understand how is it associated to *C. crambe*.

Line 443 and 448. Still extremely confused about all this. This two sentences: "Within Proteobacteria, a dominant phylum in marine sponges genus *Bdellovibrio* and order EC94 were correlated to HMA species." vs. "Among bacteria mostly associated with *C. crambe*, we found the EC94 group(Betaproteobacteriales)"....Is the EC94 the same in both sentences? How is it correlated to HMA and associated with *C. Crambe* (LMA) at the same time? Which one is it?

supplementary

Line 13. Correct: Supplementary figure S1 and figure S2

Line 47. What library construction was used?

Table S1. I would add info of the phenotypes of each sample.

Reviewer #3 (Remarks to the Author):

This study by Mazzella et al. investigates the microbiomes and metabolomes of four ecologically and biologically variable co-occurring sponge species, with the aim of uncovering functional relationships to help to explain niche/ phenotypic diversification among hosts.

Sponges are a valuable and intriguing model system for host-microbiome studies and while these species are well-studied in terms of their microbiomes using 16S methods, this study constitutes a novel experimental design with the additional use of metabolomics and the ecological and biological diversity of the species from the same geographic locales. The methods employed are technically sound and reproducible from the microbiome perspective (I do not have expertise in metabolomics so cannot comment to this) and the manuscript is well written and well presented. I enjoyed the discussion and the suggestions made to link microbiome/ metabolome features to the biology and ecology of the sponge holobionts – however I also have some reservations (comment 2, below) about the limitations of the evidence for these links and the conclusions drawn. Overall this study is very interesting and adds to the current knowledge of the functional role of sponge microbiomes within an ecological context.

Specific comments:

1) Intraspecific variation in growth forms, and bleached/ colored forms correlating to microhabitat are described in the introduction for some of the species. However, the authors have not described which of these were sampled or if this was consistent within species. This is vital to know as these characters would be expected to influence the variables measured in this study.

2) Related to above - a central premise and novelty for the study is around the bio-ecological differences between and within the species. However the samples appear to have only been collected from shallow (i.e. light) habitats, which limits exploration of the intraspecific differences described. Despite this, the authors do discuss their results in the context of these intraspecific differences – I feel that without having sampled multiple habitats and morphologies, and with only correlative evidence, some of the conclusions overreach in this regard and some further exploration of potential mechanisms in the discussion would be useful. E.g. the authors state regarding *P. ficiformis* that its

microbiome variability may enable colonisation of different habitats. But different habitats and depths were not sampled here. Elsewhere it is suggested that *P. ficiformis* maintains heterotrophic bacteria related to its ability to colonise dark/ shaded areas – how and why would this symbiont relationship be maintained in light-dwelling individuals sampled here, especially if horizontal transmission is thought to be the main mode of symbiont acquisition for *P. ficiformis*, and therefore symbiont switching could allow phenotypic plasticity in this regard?

3) The metabolome results offers interesting findings, but the authors seem to attribute this only to the microbiome in some areas of the text, but the sponge itself could potentially be producing some of these compounds?

4) Predicted function for microbiome – this analysis is based on 16S taxonomy at often quite broad taxonomic levels (e.g. order). Taxonomy does not correlate well to function in many cases and so many sponge symbionts have not been cultured or had their genomes fully sequenced - this needs to be acknowledged in the text as a caveat.

5) The introduction gives detailed background information on the study system and context, but does not highlight clearly the knowledge gap addressed and the novelty of the study.

6) In the methods, I would like to see more information about relationships that were investigated in the final section on statistics, even if the statistical tests themselves cannot be described due to space limitations. I understand the journal style is to keep the methods section short, so perhaps it's not possible though.

7) Results -I prefer to see the exact p values in the results (except when $p < 0.001$) – currently they are just given as < 0.05 when significant.

Line by line comments:

Line 67 – Consider substituting 'resumed' for 'summarised'

Line 77 – Consider substituting 'either group' for 'both groups'

Line 89 – Please specify in what way they are more variable (intraspecifically, spatially, temporally?)

Line 98 – Many readers may be unfamiliar with the sponge loop so this should be defined/ described briefly here

Line 99 – 'others' instead of 'other'

Line 102-110 – It was initially difficult to see where this was going until the final sentence. ("All these microhabitat acclimating aspects could be related to host- microbiome-metabolome interactions."). I feel introducing this idea prior to describing the study species microhabitats would be better for reader navigation.

Line 127 – Should be 'supplying'?

Table 1 – It is not clear why there is both 'Morphology' and 'Growth type' columns – some information is shared between these.

Line 164-165 – Please give references for primer sequences

Line 177 – Species names are not in italics

Line 209 – This part is a little unclear – does it mean 351900 reads across the sponge samples and 56304 reads across the seawater samples? I don't think it's necessary to include this as we already have the number of reads per sample given which is more informative than the cumulative total across sample types.

Line 211 – Not clear what 'feature' means here

Line 212-213 – Standard error would be nice to see here for the mean ASV numbers

Line 360-361 – "But they might be involved in host acclimatization processes, via symbiont shuffling – becoming functionally relevant". It would be good to add a little more explanation for clarity.

Line 382, line 420 – Suggest to replace "researches" with "studies"

Line 542 - "may allow it to colonize habitats with certain light exposure"- this is a little vague, please be specific on light exposure.

Figure 2 – Scale bars needed for two left side maps. The lower-left map does not seem to correlate to the box highlighted in the upper left map? Labelling A B and C might help with orienting the reader

Figure 4 - It would be good to see error bars for the per sample mean ASVs. I am unsure why inertia ellipses have been used for the Jaccard NMDS but not the Bray-Curtis NMDS. The inertia ellipses and what they show needs mentioning in the legend. The figures themselves are not labelled A B C D.

Figures 4, 5, 6, 7, 8, 9- Species names need to be italicised, and underscores removed from variable names.

Napoli, 24th November 2023

Dear Reviewers,

Our manuscript entitled: “**High microbiome and metabolome diversification in coexisting sponges with different bio-ecological traits**” submitted to the journal Communications Biology (Ms. Ref. No.: COMMSBIO-23-2829) has now been revised attending to the raised inquiries by the three reviewers. We hope to have addressed satisfactorily all the comments and inquiries.

Please, find our detailed responses (in black), underneath each reviewer’s suggestions (in blue). You may locate the edits in the CLEAN MS “Article File” by searching the indicated line numbers as “L XXX – XXX”. Reviewers could optionally compare the edited changes in a REVISED MARKED MS (“Revised Manuscript with Track Changes”), sent separately as a supplementary file or upon request (depending on journal submission system policies).

Reviewer #1: Host-microbiome interactions

Reviewer #1 (Remarks to the Author):

General Comments:

Dr. Núñez-Pons and colleagues have presented their study investigating the functional relationships between microbiomes and metabolic signatures in coexisting Mediterranean sponges. The research uncovers distinct patterns among various sponge species concerning their microbiomes, growth typologies, and ecological adaptations. The findings suggest that diverse microbiome networks play a role in niche diversification and highlight analogous patterns of "symbiont evolutionary convergence" shaping their bio-ecological traits. Here are some specific suggestions to enhance the manuscript's clarity and impact:

Authors much appreciate the suggestions provided to improve our manuscript (MS), they were extremely useful and constructive!

1. In the introduction, the authors rightly identified the lack of integration between microbiome and metabolomics datasets as a gap in current knowledge. They effectively describe these datasets separately. However, I recommend conducting further analysis to establish connections between microbiome and metabolome profiles. It would be beneficial to include a table summarizing commonalities and differences in clustering patterns of microbiome diversity, microbiome functional predictions, and holobiont metabolome. Additionally, providing 2-3 examples linking predicted genomic functions of the microbiome with host metabolic signatures would strengthen the argument.

Thank you for the valuable comments and recommendations. Indeed, as the reviewer has pointed, there is still a remarkable gap of available information in all regarding Porifera metabolic profiling, and the proved interconnections of metabolomes dynamics with the symbiotic microbial players and functions, according to the state-of-the-art literature. We appreciate the suggestions provided, since, despite of the actual lack of knowledge in the topic, and challenge to empirically proof holobiont metabolic and microbial integration, new correlation analyses conducted (as suggested by the reviewer) have revealed some interesting interconnections patterns. Thanks to the correlation analyses we were able to observe some trends, and formulate interpretations, which now open avenues for more guided future testing and verification with specific approaches. We added new sections in relation to these new multi-omics correlation

analyses in the manuscripts from the abstract to the discussion, and two more figures to the main text concerning the inter-omics analyses performed to clarify the reviewer's concerns. Please, see

Abstract: L: 51 – 58.

Introduction: L: 139 – 144

Materials and methods: L: 281 – 332

Results: L: 482 – 523

Discussion: L: 731 – 793

Conclusion: L: 822 – 832

As inspired by the reviewer, we have produced a Graphical abstract (in place of the proposed table) resuming all the outcomes interconnecting the 3 major approaches (microbiome community characterizations, predicted functions, and metabolome profiling) and their similarities in clustering the 4 sponge host species studied, and the most highlighting results for each host in each analysis. We strongly think this synthetic figure, together with the correlations analyses, all suggested by the reviewer have much enriched the interpretation and presentation of our work. Again thank you. Please, see Graphical Abstract.

We added a few examples linking microbial predicted functions with holobiont metabolic signatures in the Introduction section, as suggested. Please, see L: 134 – 136.

2. Since the metabolome is based on the holobiont, and the study compares both HMA and LMA sponges, it would be valuable to discuss how much of the differences in metabolic profiles are attributed to the microbiome versus the host. Does the microbiome contribute to divergence or convergence in the holobiont metabolome?

These questions are very interesting and a bit intricate to respond with certainty with the available information.

On what regards the first comment question: With our data we cannot accurately determine what is the exact contribution of the microbiome and its respective host within each holobiont metabolic profiling. On the one hand, because our procedures did not examine the metabolome separating host and microbial cells, and on the other, because, even if applying the mentioned approach (which is highly challenging, e.g., how to perform intracellular symbionts separation...), there are still compounds that are fruit from metabolic exchanges between host and symbiont cells, or are part of shared pathways, generating confounding production origins. On what regards, HMA and LMA classification, this organization delimits sponge holobionts by the microbial density, and may entail other accompanying morphological features (e.g., massive vs encrusting growth, low vs high pumping rates), but the HMA and LMA dichotomy may group together sponges that are very different in many functional aspects. From our experiences, the host species carries the strongest signal, to predict differences in microbiomes, metabolomes, and acclimatization processes. This said, in the MS text we often point this limitation on the possibility to disentangle the degree of the differences in metabolic profiles attributed to the microbiome *versus* the host, no matter the HMA and LMA condition (See for instance L: 822 – 826).

As for the second question: We can conclude that the microbiome contributes to divergence of sponge metabolomes, when considering interspecific datasets (including host sponge of diverse species), but more likely to metabolome convergence in intraspecific subsets. From the correlations analyses with our dataset we found an overall microbiome-metabolome coherence when considering all four sponge species based on Procrustes analyses and Spearman and Sparse canonical correlation analyses. This means that most microbes (microbial genera) were associated to a large number of metabolites (class metabolites). Indeed, for each sponge species specific microbes reflexed host-specific metabolic signature: in the full dataset sponge species could be discriminated by a microbiome and a correlated metabolome. In lieu, this global concordance lost significance at the single species, where small sets of microbes were correlated to few metabolite

classes, which we interpret as probable microbial metabolic functional redundancy within each host. This functional redundancy was seemingly more remarked in *Chondrosia reniformis*. Also, predicted functions advocate towards a seeming convergence, by clustering our hosts together. In the literature, we find diverse work reporting the existence of important metabolic redundancy in sponge microbiomes to attain overall homeostatic fitness, hence contribution to metabolome convergence (<https://www.nature.com/articles/s43705-022-00196-3>). Under changing environmental scenarios, we also propose some degree of metabolome divergence, when assisting holobionts in colonizing other niches, for niche diversification. All these points were argued in the text, e.g.: L: 833 – 836.

3. More details about the methods used in the DEICODE results should be provided. Specifically, clarify the meanings of log ratios, denominators, and numerators to aid reader understanding.

We understand the reviewer's point of view and we agree that this analysis is a bit complicated to understand due to its novelty, and not as intuitive terms. DEICODE (<https://library.qiime2.org/plugins/deicode/19>) is the name plugin integrated in QIIME2 or as a stand alone plugin, used to compare beta diversity compositions between sample groupings based on Aitchison distances. This approach is highly recommended for microbiome and metabolomics studies, as it takes into account the compositionality of data, and high levels of sparsity and zero values, common of these types of omics data. The Aitchison distance is calculated as an Euclidean distance on Centered Log Ratio (CLR) transformed data, and the result of this CLR transformation transforms the absolute counts or peak areas data into log ratios, which represent well how big differences between features (microbes/metabolites) are. The resulting Aitchison distance matrix is then visualized in ordination plots of the kind: robust compositional principal component analysis (RPCA) biplots, where features (microbes) loadings (log ratios) are stored and can be plotted as vectors. Feature loadings were visualized on the visualization tool plugin Qurro (Fedarko et al., 2019; <https://github.com/biocore/qurro>), where certain consortia of features (microbial taxa) could be identified as major drivers (high ranked) of sample type grouping along the selected ordination axis of the RPCA biplot. The resulting selected numerator and denominator differential taxa were then used to generate/calculate new log-ratios of the most explanatory microbial balances separating sample types. These microbial balances' log ratios based were then tested statistically with Welch's *t*-tests for dissimilarity and significance between sample groupings ($p < 0.05$).

In the precedent version of our MS, we remained more general in the Methods & Materials' section, and included more details about the procedures and bioinformatics pipelines in the Supplementary file. Since we need to comply with a text length compromise (be explanatory and avoid wordiness), while addressing the reviewer inquiry, we have added some more details about DEICODE beta diversity in M&M in the main text section, and we have made still more explanatory descriptions on all the methods in the Supplementary file. We hope to fulfill all reviewers points this way, please considering this our MS is not intended as a Methods paper contribution. See (L: 221 – 242).

“Data for this analysis were log transformed and centered to obtain centered log-ratios (clr), a transformation widely applied for compositional data investigations, which transforms raw absolute counts data into log ratios”

and

“Qurro calculates new log-ratios of microbial balances between any chosen sets of numerator features and denominator features. This is achieved by computing, for each sample *S*, the log-ratio between the total counts of the selected numerator features (numerator ASVs) and the selected denominator features (denominator ASVs)”

We have to point out that DEICODE is the name of a QIIME2 plugin that does a PCA based on aitchison distance in an unsupervised mode (no covariates included in the formula). Data for this analysis are centered and log-transformed to obtain centered log ratios (clr).

We highlighted that this tool was previously used adding some bibliography and the following sentence in the supplementary section:

“This tool has been previously used in order to analyze compositional datasets both in biomedical and ecological studies (L: 228 – 229).

We hope this is more clear to follow now. For further information see also some bibliography here:

<https://doi.org/10.1101/gr.265645.120>

<https://doi.org/10.1093/nargab/lqaa023>

<https://doi.org/10.1128/msystems.00016-19>

<https://doi.org/10.1016/j.scitotenv.2023.164040>

4. Ensure that figure panels have appropriate labels for clarity.

We corrected the figure labels accordingly, where necessary, see **Figs 1- 11**.

Specific Comments:

1. Line 159: Please explain why there are more DNA samples collected than sponge samples.

Yes, we agree, this part was not very clear with replicates and actual DNA samples. We could not collect the same number of replicates for all species, so as pointed, we clarified this observation in the main text as follows: in **L 160 – 167** >>> “Sponge pieces (~10 cm³) of the target species *Petrosia ficiformis* (n = 12), *Crambe crambe* (n = 14), *Chondrilla nucula* (n = 10), and *Chondrosia reniformis* (n = 15) were collected along with ambient seawater (n = 9) in spring 2018 by scuba diving at 2-5 m depth around the island of Ischia, (Southern Tyrrhenian, Mediterranean Sea, Italy), where all species are highly abundant^{61,62}. Sponges were recovered from three nearby sites with similar conditions: Castello Aragonese (CCO; 40°43'55.9"N - 13°57'52.9"E), Grotta Mago external (GF; 40°42'41.6"N - 13°57'51.4"E) and Sant'Anna (SA; 40°43'36.5"N - 13°57'43.4"E) (Fig. 2).”; also in **L 177 – 179** >>> “DNA from all four species (n = 51 total sponge samples) and from seawater filters (n = 9) was extracted using the DNeasy PowerSoil Pro Kit (Qiagen).”; and further in the **Supplementary file: section “1.3. Quality control and filtering of raw sequence data” L 57 – 59**.

“One sample belonging to the sponge species *C. crambe*, one belonging to the sponge *P. ficiformis*, and one seawater sample were dropped after rarefaction, due to sequencing depth not passing the established threshold of minimal reads.”

2. Line 281: This sentence requires clarification. Please provide additional information or context to make it more understandable.

The sentence was edited accordingly as follows: “SIMPER analysis, performed to estimate the contribution of each microbial taxon (%) to the dissimilarity between groups, confirmed high dissimilarity percentages (~98-99 %) among microbiome compositions of the four sponge species studied. Remarkable dissimilarities were also reported between all sponge microbiomes and prokaryotic assemblages of the seawater” (**L: 413 – 419**).

Overall, the manuscript presents an intriguing study that describes microbiome characterization and metabolomics to explore functional relationships in Mediterranean sponges. Addressing the points outlined above would further enhance the clarity and potential of the study to engage a

wider

scientific

audience.

Again, we are delightful about the revision and recommendations given, and hope we have addressed the points raised adequately.

Reviewer #2: (Remarks to the Author):

The work of Valerio Mazzella and coauthors, entitled "High microbiome and metabolome diversification in coexisting sponges with different bio-ecological traits" describes the microbiome and metabolome of 4 sponge species plus water in 3 locations of the Mediterranean Sea. It also includes identification of bacteriocytes and predicted functions. While the work is well performed and clearly written, there are several issues that need clarification.

Thank you for considering our work well performed and clearly written, in the following lines we will respond to the issues outlined, with the will to resolve them satisfactorily.

My main concern is the choice of samples. From the introduction, the authors explain that some of the species present morphotypes (pigmented, bleached, lobular, reticular, etc) that could be related to host-microbiome-metabolome interactions (Line 109). Fig. 1 shows these morphotypes among the individuals considered in this study, however there is no table specifying what morphotype presented each sample. Location is also considered as a factor of variability, because there are illuminated areas and caves among the sampling site.

Any of these factors (morphotype and location) are considered and analysed separately for any of the species. All individuals are pulled together and treated as simply replicates throughout the study. If the goal was not to see differences in these morphotypes, I am not sure why the authors took a mix of samples instead of similar morphotypes.

I personally think that the differentiation of the morphotypes, together with the metabolome, would be the main interest of the work, since the sponge species themselves have been studied previously in many other locations, and it is not novel. These factors may also be behind the larger or smaller infraspecific differences of each species. For instance, we don't know whether the largest differences in *P. Petrosia* are related with their morphotypes. The authors should investigate a bit more whether differences among phenotypes or locations are worth mentioning, or whether it is ok to pull samples together.

We apologize if the text was not clear enough in the previous version of the MS in these aspects mentioned, the concerns of the reviewer have helped us to put more care in explaining/addressing these parts in the new revised version. In the introduction it is exposed how the chosen species have a range of potential intraspecific phenotypic traits related to possible morphologies, growth forms, microbial densities, reproduction typology, predominant symbiont transmission modes, colouration, feeding strategies, habitat preferences, allelochemistry known etc... that make them different from one another. Our intention here was to introduce all those possible phenotypes described in nature, and try to find linking patterns potentialities with the microbiome compositions and the metabolome profiling across the four species, and further make correlations with these multi-omics datasets. Therefore, our scope in this paper is not to seek for intraspecific phenotypic changes, either in morphology or habitat in correlation to microbial and metabolic dynamics, as this is being object of upcoming contributions. Therefore, in our experimental design, we only include a single type of phenotype per species, this means that all samples in the same species are of the same phenotype and location: pigmented and massive for *P. ficiformis* and *C. reniformis*, and similarly pigmented and lobular for *C. nucula* and encrusting for *C. crambe*, and all from illuminated habitats, with similar conditions in general. We clarified this in the text as follows: In the Introduction: "These species were selected on the basis of distinctive

bio-ecological potential range traits, involving reproductive strategies, microbial abundance, preferred habitat, general morphology, coloration, growth-type and allelochemistry (summarised in Table 1 and Fig. 1)” (L 65 – 68). Also in (L 139 – 142) “to investigate the relationships between sponge microbiomes and metabolic signatures, in the context of host specificity, and considering several distinctive bio-ecological trait ranges, characterizing the four holobiont systems selected.” (L 147), Table 1 legend “**Table 1.** Summary of the main range of bio-ecological characteristics of the selected sponge species shown in Fig. 1 as living morphotypes.”. We added a note under the table clarifying the phenotypes of study: (L 149) “*All samples in the same species are of the same phenotype: pigmented and massive for *P. ficiformis* and *C. reniformis*, pigmented and lobular for *C. nucula* and encrusting for *C. crambe*, and all from illuminated habitats and analogous conditions.”

Methods: “All samples in the same species collected for our study were of the same phenotype, this is: pigmented and massive for *P. ficiformis* and *C. reniformis*, pigmented and lobular for *C. nucula* and encrusting for *C. crambe*, and all from illuminated habitats, with analogous conditions as specified in Fig. 1” (L 156 – 159),

Conclusions: “Our study performed on samples of a single phenotype appertaining to four different target species, allowed us to encounter trends in the microbiomes, metabolomes and/or their inter-omics interactions that seem to correlate with the potentiality of range traits for each sponge.” (L 797 – 800).

In any case, we could encounter certain trends in the microbiomes, metabolomes and/or their interactions discussed over the text that seem to correlate with the potentiality of range phenotypic traits for each sponge. For instance, *P. ficiformis* has the most diverse microbiome with the smallest core and largest variable microbiome, reporting also the largest ASV condision with the seawater as HMA sponge. This dynamic microbiome may allow this species to colonize a wide range of habitats (dark, illuminated) adopting different morphologies (massive and pigmented to reticular and bleached). This can likely occur along shifting and shuffling processes of symbiont members, driving small adjustments of the metabolic profile, as can be suggested by the highest canonical correlation found in this species between microbiome and metabolome. *Chondrosia reniformis* instead, can colonize also a wide range of habitats, from dark to illuminated, with changes on its pigmentation, but with no changes in its morphology. This species revealed a large core microbiome, and is described as having a diverse but very stable microbiome across habitats and locations (as opposite of *P. ficiformis*). In this case, *C. reniformis* probably relies on high metabolic redundancy of its diverse but stable microbiomes for its ability to colonize a range of conditions with no apparent major phenotypic changes, as suggested by its low canonical correlation between microbiome and metabolome, as compared with *Petrosia*. *Crambe crambe* shows dominance of a single symbiont probably non shared with the seawater and likely obtained via vertical transmission, this reflecting its brooding reproductive nature. This host shares many taxa with the sweater, including the majority of its core taxa, which include cyanobacteria that correlate with its preference for photic to shaded zones. It further reveals inter-omics canonical correlations similar to *P. ficiformis*, thus demonstrating the functionality of the dynamic rare biosphere, while its largest condision with the seawater, along with certain association patterns to groups of bacteria link with its LMA condition. Finally, *C. nucula* has predominance of Cyanobacterial symbionts that explain its requirement for photic habitats to get photosynthates from these photosymbionts, as reflected in the predicted functions outcomes. It has the largest core, and the lowest condision with the seawater as HMA sponge, which may reflect larger vertical transgenerational transmission of endosymbionts, and slower intragenerational acclimatization capability.

Another two general points I don't personally agree with are:

1. the description of rare taxa in LMA sponges as part of the microbiome diversity. I believe it can just be food/passing bacteria (see specific comments below).

Thank you for this observation. This is in fact a debating aspect in sponge microbiology. In our case, we have to say that all sponge samples were treated in the same way, rinsed thoroughly with

sterile SW prior extractions, sampled in the same way, extracted the same day with the same protocol, etc... And all and all our HMA sponges did not behave in the same way as *Crambe crambe*, as they shared very few ASVs with the ambient water. This suggests some sort of sponge preference in the case of *Crambe* to horizontally retain a larger amount of ASVs shared with the seawater. Why would HMA sponges not share as many microbes with the seawater under the same technical conditions? In addition, we still find a strong host specific signal in our sponge microbiomes, and in all cases significantly very different from the seawater. Beta diversity analyses yielded different microbial compositions between *C. crambe* and other sponges, and with the seawater. Hence, we believe this rare biosphere should play a role and be selectively retained in *Crambe crambe*. Consider that *Crambe crambe* was significantly different from the SW in microbial composition, even if its microbiome is strongly dominated by one single ASV. This difference with the SW is corroborated by several diversity methods, including methods based on presence absence (Jaccard), relative abundance based distances (Bray-Curtis) and compositionality based analyses (Aitchison). Hence, even if *C. crambe* uptakes a certain portion of microbes from the seawater, this process is not passive. This host selects in which proportion retaining these taxa, indicating some functional significance of these horizontally acquainted microbes. Note also that eight of the ASV shared with the seawater are part of the 13 core taxa in *C. crambe*, while none of the core taxa in any of the 3 HMA sponges in the study was shared with the seawater. Also, our correlation analyses demonstrated association of rare taxa with a few metabolite classes (see L 508 – 523).

Here below we add some bibliography that further supports our findings on the relevance of the rare biosphere as microbial endosymbionts, and with large condision with the seawater, hence likely horizontally transmitted into their host:

<https://doi.org/10.1038/ismej.2013.227>
<https://doi.org/10.21203/rs.3.rs-3365419/v1>

Finally, from personal observations, again, with another co-occurring LMA species in the same collecting areas, *Crambe crambe* and this other LMA sponges have very different microbiomes from each other, even when living on the same rock, and even if sharing a considerable amount of taxa with the same seawater! ... Meaning that both co-occurring LMA sponge holobionts take up different players from the seawater to join their symbiotic microbiome.

2. the presence of bacteriocytes in *C. reniformis*. In the microscopy images selected I definitely don't see that, and the references used to support that claim are in fact not correct (see comments below), so I would need a better image or proof to accept this finding.

We feel a bit debating here, actually, in the sense that the point raised by the reviewer is surely an interesting hot topic. Since we do not have at the moment solid imaging material to prove we decided to remove the statement of presence of bacteriocytes in *C. reniformis* still as an uncovered mystery. (L: 527 – 529).

However, we have to point out that the bibliography on this field is not clear at all. Indeed some authors (see below) assert the presence of bacteriocytes in the embryos of the sponge *C. reniformis*.

<https://doi.org/10.1007/s00227-004-1489-1>
<https://doi.org/10.1111/j.1095-8312.2009.01202.x>
https://doi.org/10.1007/978-1-4615-0747-5_32

In my opinion this work is interesting for the scientific community, but it could be greatly improved by checking and clarifying some of the issues I raised. Therefore I suggest acceptance after major changes are made.

We are very happy that the reviewer finds our study interesting for the scientific community, and we have made our best to address her/his inquiries and concerns to improve the contents and the message.

Specific comments

Line 41-42. Could be contaminants from water in channels, food or passing bacteria.

As exposed above, we do believe these microbes have relevance in LMA sponge microbiomes, as all sponges were equally treated (e.g., vigorously rinsed with SW prior to extraction, and followed the same procedures), and the other sponges did not show this behavior. Hence, we believe this rare biosphere should play a role and be selectively retained in *Crambe crambe*. Still, *Crambe crambe* was significantly different from the SW in microbial composition, even if its microbiome is dominated by one single ASV. This difference with the SW is corroborated by several diversity methods, including methods based on presence absence (Jaccard), relative abundance based distances (Bray-Curtis) and compositionality based analyses (Aitchison). Hence, even if *C. crambe* uptakes a certain portion of microbes from the seawater, this host selects in which proportion retaining these taxa, indicating some functional significance of these horizontally acquainted microbes. Another maybe interesting information to respond the reviewer, is that *Crambe crambe*'s core microbiome has 13 core taxa, of which eight are shared with the seawater, and likely representing key functional taxa. Instead, none of the core taxa in any of the 3 HMA sponges in our study was shared with the seawater. Also, our correlation analyses demonstrated association of rare taxa with a few metabolite classes (see L 508 – 523). We edited this part as follows: “*C. crambe* shared many rare amplicon sequence variants (ASV) with the surrounding seawater. This suggests important inputs of microbial diversity acquired by selective horizontal acquisition.” (L 40 – 42).

See also:

DOI: <https://doi.org/10.21203/rs.3.rs-3365419/v1>

Line 46. Change to: "According to putative functions, the microbiome of *P. ficiformis* and *C. reniformis* were mostly functionally heterotrophic."

We modified the text accordingly, thank you (L 43 – 45).

Fig. 1 Legend: Also add "(photic zone)" for the preferred habitat of *c. crambe*.

Thanks for the indication, we added "photic zone" for the preferred habitat of *C. crambe* as suggested (Fig. 1). (L 1363)

Table S10. It is called Table S9 in the download area. Correct.

According to our records (see below screenshot), Table S10 corresponding to the metabolomics dataset is correctly called Table S10 in the download area. Maybe this is a problem of the reviewer's personal area?

Manuscript Items

1. Author Cover Letter PDF (46KB)
2. Article File PDF (2027KB)
3. Supplementary Material PDF (766KB)
4. Table S8 PDF (124KB)
5. Table S10 PDF (73KB)
6. Reporting Summary PDF (1610KB)
7. Editorial Policy Checklist PDF (1555KB)
8. Reviewer Zip File "Zip of files for Reviewer"

There are action items pending. Please click on the links next to the arrows →.

Line 88. *C. reniformis* is used as example of both via horizontal acquisition and vertical transmission. I would leave *P. ficiformis* as horizontal acquisition as this species does not present any vertical transmission, and the other 3 as example of VT (possible combined with HA of course).

Yes, we appreciate this point. As suggested, we left *P. ficiformis* as a host obtaining symbionts mostly by horizontal acquisition, and left the other 3 spp. (including *C. reniformis*) as examples of holobionts acquiring their microbes majorly via vertical transmission, in agreement with the reviewer's suggestion. The text was modified as "...as majorly described for for *P. ficiformis*, or by vertical transmission from parents to offspring in diversified reproductive strategies and modes (e.g., through brooding larvae as in *C. crambe*, or within oocytes in *C. nucula*, *C. reniformis*) 20–25" (L 86-89).

Line 95. Why is "(endo)symbionts" specified here. Symbionts living outside host cells, do the same functions.

By (endo)symbiont we meant microbial symbionts living inside its host body, either inside or outside host cells, so indeed there is no need for specification really. To avoid any confusion, the text was edited as "symbionts" (L 96).

Line 108. This sentence should follow the previous with a comma (,). Eg. "While, others ..."

Edited as indicated, please see: L 106 – 109.

Line 144. It took me a while to understand the numbers in Line 159, since not all species were collected in all 3 locations. I think that should be clarified here.

Yes, we apologize, the reviewer is right that it was not clear as written previously. This has been now clarified in: L 160 – 164 >>> "Sponge pieces (~10 cm³) of the target species *Petrosia ficiformis* (n = 12), *Crambe crambe* (n = 14), *Chondrilla nucula* (n = 10), and *Chondrosia reniformis* (n = 15) were collected along with ambient seawater (n = 9) in spring 2018 by scuba diving at 2-5 m depth around the island of Ischia, (Southern Tyrrhenian, Mediterranean Sea, Italy), where all species are highly abundant^{61,62}. Sponges were recovered from three nearby sites with similar conditions: Castello Aragonese (CCO; 40°43'55.9"N - 13°57'52.9"E), Grotta Mago external (GF; 40°42'41.6"N - 13°57'51.4"E) and Sant'Anna (SA; 40°43'36.5"N - 13°57'43.4"E) (Fig. 2)."; also in L 177 – 179 >>> "DNA from all four species (n = 51 total sponge samples) and from seawater filters (n = 9) was extracted using the DNeasy PowerSoil Pro Kit (Qiagen)."; and further in the Supplementary file: section "1.3. Quality control and filtering of raw sequence data" L 57 – 59

“One sample belonging to the sponge species *C. crambe*, one belonging to the sponge *P. ficiformis* and one seawater sample were dropped after rarefaction, due to sequencing depth not passing the established threshold of minimal reads.”

Line 159. Why are not 15 samples for *P. ficiformis* and *C. Crambe*?? It should be explained whether the authors collected 5 individuals in each location or not. Or whether the extractions failed for some of them and why. But numbers are not clear at the moment (same for the water samples).

Yes, we agree, this part was not very clear with replicates and actual DNA samples used. We could not collect the same number of replicates for all species, plus three samples failed in the downstream analysis after rarefaction, so as pointed above, we clarified this observation in the main text in several parts as follows: in L 160 – 164 >>> “Sponge pieces (~10 cm³) of the target species *Petrosia ficiformis* (n = 12), *Crambe crambe* (n = 14), *Chondrilla nucula* (n = 10), and *Chondrosia reniformis* (n = 15) were collected along with ambient seawater (n = 9) in spring 2018 by scuba diving at 2-5 m depth around the island of Ischia, (Southern Tyrrhenian, Mediterranean Sea, Italy), where all species are highly abundant^{61,62}. Sponges were recovered from three nearby sites with similar conditions: Castello Aragonese (CCO; 40°43'55.9"N - 13°57'52.9"E), Grotta Mago external (GF; 40°42'41.6"N - 13°57'51.4"E) and Sant'Anna (SA; 40°43'36.5"N - 13°57'43.4"E) (Fig. 2).”; also in L 177 – 179 >>> “DNA from all four species (n = 51 total sponge samples) and from seawater filters (n = 9) was extracted using the DNeasy PowerSoil Pro Kit (Qiagen).”; and further in the Supplementary file: section “1.3. Quality control and filtering of raw sequence data” L 57 – 59

“One sample belonging to the sponge species *C. crambe*, one belonging to the sponge *P. ficiformis* and one seawater sample were dropped after rarefaction, due to sequencing depth not passing the established threshold of minimal reads.”

Fig. 3. I am not aware of *C. reniformis* presenting bacteriocytes, but definitely in those images I don't see any. I only see a host cell doing some phagocytosis, or cell extensions (same in Supp. Fig. S2)

As mentioned above, we eliminated this statement as our images are not clear at all. However, the literature on this field is controversial and not clear at all. Thus, even if we believe that *C. reniformis* has bacteriocytes (as previously reported, see below), we are not able to prove it at the moment and we removed these sentences from the main text as suggested. See for instance L 1370-1373.

Line 200. I would need a better prove of the bacteriocytes in *C. reniformis*.

Yes, as above, at the moment, and following reviewers' consideration, we agree to eliminate the statement of presence of bacteriocytes in *C. reniformis*.

See L 329 –325.

However, just as a curiosity during our re-reading to address this revision, the literature on this topic is not clear at all and several authors indicated the presence of bacteriocytes in the sponge *C. reniformis*.

Here are some examples:

<https://doi.org/10.1007/s00227-004-1489-1>

<https://doi.org/10.1111/j.1095-8312.2009.01202.x>

https://doi.org/10.1007/978-1-4615-0747-5_32

In any case, we thank the reviewer for her/his suggestion.

Line 203. I strongly believe both *C. crambe* and *C. reniformis* present absence of bacteriocytes.

We understand the reviewer views, but literature is not always congruent. For instance for *C. crambe* Maldonado 2007 (DOI: <https://doi.org/10.1017/S0025315407058080>) describes bacteriocytes, see picture of Bacteriocytes in *C. crambe* therein. For *C. reniformis*, we exposed our view above. In our case, we did not find bacteriocytes in *C. crambe* either, and this was stated in the main text. as “*C. crambe* and *C. reniformis* were as characterized by the absence of bacteriocytes and by the presence of prokaryotic cells occupying the mesohyl” in L 332 – 335.

Concerning *C. reniformis*, our responses were exposed responses several times above. Bacteriocytes within embryos of this sponge as reported by the following articles/chapters.

<https://doi.org/10.1007/s00227-004-1489-1>
<https://doi.org/10.1111/j.1095-8312.2009.01202.x>
https://doi.org/10.1007/978-1-4615-0747-5_32

Line 214. I wouldn't say the pattern of reads is the same as the Shannon. *C. nucula* had low number of reads, but its diversity is higher, as expected for a HMA.

Thank you for this observation. This was clarified as “and this pattern was also partially observed for the Shannon diversity index. Indeed *C. nucula* displayed in this case a higher alpha diversity” in L 343 – 344.

Line 223. I have not seen this deicode biplot before, and I don't understand which sponge species are defined by the arrows pointing to the left?? I also don't know where to find the most significant ASVs driving sample grouping, or the meaning of the Log-Ratio or numerator/denominator. Maybe the authors need to explain in more detail the meaning, procedure and outcome of this analysis.

We understand the reviewer's point of view and we agree that this analysis is a bit complicated to understand due to its novelty, as many previous work is based on classical relative abundance data. In this case, DEICODE is a compositional approach that accounts for data compositionality, as that of microbiome and metabolome data. In the precedent version of our MS, we left most of the method procedures and bioinformatics pipelines details in the Supplementary file, to lighten the main article text. Here we provide some explanation on DEICODE procedure: DEICODE (<https://library.qiime2.org/plugins/deicode/19>) is the name plugin integrated in QIIME2 or as a stand alone plugin, used to compare beta diversity compositions between sample groupings based on Aitchison distances. This approach is highly recommended for microbiome and metabolomics studies, as it takes into account the compositionality of data, and high levels of sparsity and zero values, common of these types of omics data. The Aitchison distance is calculated as an Euclidean distance on Centered Log Ratio (CLR) transformed data, and the result of this CLR transformation transforms the absolute counts or peak areas data into log ratios, which represent well how big differences between features (microbes/metabolites) are. The resulting Aitchison distance matrix is then visualized in ordination plots of the kind: robust compositional principal component analysis (RPCA) biplots, where features (microbes) loadings (log ratios) are stored and can be plotted as vectors. Feature loadings were visualized on the visualization tool plugin Qurro (Fedarko et al., 2019; <https://github.com/biocore/qurro>), where certain consortia

of features (microbial taxa) could be identified as major drivers (high ranked) of sample type grouping along the selected ordination axis of the RPCA biplot. The resulting selected numerator and denominator differential taxa were then used to generate/calculate new log-ratios of the most explanatory microbial balances separating sample types. These microbial balances' log ratios based were then tested statistically with Welch's *t*-tests for dissimilarity and significance between sample groupings ($p < 0.05$).

The most significant ASVs driving sample grouping are represented by vectors in the RPCA biplot, these are the colored arrows which taxonomy appears at the side, coded by colour (see Fig. 5).

This said, and in order to address the reviewer inquiries, together with that of Rev # 1 (see specific explanations above to Rev #1), we have edited the main text and included further details and explanations to the method itself and the terms used in either M&M section (please see M&M L 221 – 242)

Additionally, below these lines, we provide a three dimensional view separated in three panels of 2 by 2 dimensions of our RPCA biplot output. This is to demonstrate how the arrows on the left mentioned by the reviewer are pointing to the sponge *C. reniformis*. Note that this is not immediate or clearly realized due to the two dimensional constraints of the RPCA biplot, but as soon as the third major component is included, we can see clearly samples separation. We added a note in Figure 5 legend, L 1388, to address this argument: “* For other dimensional views, including the third ordination component PC3 of the RPCA plot (A) see Supplementary figure S5”, and added these 3-D plotting outcomes in the Supplementary S5.

For further information and examples applying these procedures see also:

<https://doi.org/10.1101/gr.265645.120>

<https://doi.org/10.1093/nargab/lqaa023>

<https://doi.org/10.1128/msystems.00016-19>

<https://doi.org/10.1016/j.scitotenv.2023.164040>

Fig. 5. In this figure, low abundant groups could be removed, so there are not so many colours together. It is hard to distinguish anything.

We did our best to improve the visualization and colour palettes used for this figure, and have also removed low abundance groups to this end, please see Fig. 6. Accordingly, we added a note in Figure 6 legend: L 1394 “* For the whole bacterial community including rare background taxa see Supplementary Figure S3”, and added the previous figure with all taxa in the Supplementary for the readers interested in the whole community composition.

Line 241. Archaea was definitely in low percentage since your primer choice does not amplify archaea. C. nucula, C. reniformis, and C. Crambe usually present low abundances (less than 4%) but P. ficiformis can harbour more than 20% relative abundance of archaea members. You could state this primer bias in methods, and don't worry about stressing here the low presence of archaea for each species.

The primers set used were Bakt_341F 5'-CCTACGGGNGGCWGCAG-3' and Bakt_805R 5'-GACTACHVGGGTATCTAATCC-3'XXX from Herlemann et al. XXX, which were designed for pyrosequencing. Indeed we did detect low prevalence of Archaea, which might be due to the reasons pointed by the reviewer. We appreciate this indication, and in agreement we included this limitation bias of the primers used in text as: "These primers are specific for bacterial communities and may reveal lower capture of Archaea communities." (L 182 – 183); and also "For prokaryotic bacterial diversity analysis, the V3-V4 hypervariable region of the 16S rRNA gene (Escherichia coli position: 341-805) was amplified..." (L 179 – 182)".

Line 273. I got lost with this numbers. If *C. nucula* had 121 ASVs (line 213), and 55 were in the core (line 270), doesn't it make a core of 45%? Where is the 95% coming from?

We think here the reviewer is confusing the number of diverse core taxa (without considering their contribution to relative abundance) with the core size. Core size is the % of relative abundance occupied by the core taxa from the total (so in the sum of core and non core). We have further clarified this concept in the text as "The size of the core microbiomes (percentage abundance occupied by the core microbial taxa with respect to the total microbiome)" (L 404 – 407).

Line 276. In the figure you can see 337 exclusive ASVs no 339, please check.

Thank you for pointing this put, we check and the correct number of exclusive ASVs was 337, this was corrected in the text "337", see L 409.

Line 344. References 10 and 52 does not mention any bacteriocytes, they are studies regarding HMA-LMA dichotomy. The 69 (Sara 1998) does mention "the follicular bacteriocytes enveloping the egg of *Chondrosia reniformis*" from the work of Levi and Levi 1976, but I think this is a bad translation of the original paper, which only mentions follicular cells around the oocytes. The follicular cells that form around the oocytes are not the same as the bacteriocytes containing bacteria in the mesohyl. Please, provide better references for this statement.

We provided references for this sentence L 529, while as exposed above, we cannot at the moment demonstrate the presence of bacteriocytes in *C. reniformis*, hence we remove any statement reporting the existence of bacteriocytes in *C. reniformis*.

However, as above mentioned we have found controversial opinions in literature and we have provided evidence that other authors mentioned the presence of bacteriocytes in *C. reniformis*. See:

<https://doi.org/10.1007/s00227-004-1489-1>
<https://doi.org/10.1111/j.1095-8312.2009.01202.x>
https://doi.org/10.1007/978-1-4615-0747-5_32

Line 352. Richness of *C. crambe* may have been similar, but diversity was clearly lower, which I believe is more important. *Crambe* probably have lots of the rare taxa contributing to the overall richness, but most likely, as LMA sponge, they are just passing bacteria from the seawater, not even symbionts.

We already responded to the reviewer on this argument: This is in fact a debating aspect in sponge microbiology. In our case, methodologically, all sponge samples, HMA and LMA, were treated in the same way, rinsed thoroughly with sterile SW prior extractions, sampled in the same way,

extracted the same day with the same protocol, etc... and after all the HMA sponges did not behave in the same way as *Crambe crambe*, as they shared very few ASVs with the ambient water. This suggests some sort of sponge selective filtering preference in the case of the LMA species of this study, *Crambe*, to horizontally retain a larger amount of ASVs shared with the seawater. In addition, we still find a strong host specific signal in our sponge microbiomes, and in all cases significantly very different from the seawater. Beta diversity analyses yielded different microbial compositions between *C. crambe* and other sponges, and with the seawater. Hence, we believe this rare biosphere should play a role and be selectively retained in *Crambe crambe*. Consider that *Crambe crambe* was significantly different from the SW in microbial composition, even if its microbiome is strongly dominated by one single ASV. This difference with the SW is corroborated by several diversity methods, including methods based on presence absence (Jaccard), relative abundance based distances (Bray-Curtis) and compositionality based analyses (Aitchison). Hence, even if *C. crambe* uptakes a certain portion of microbes from the seawater, this process is not passive. This host selects in which proportion retaining these taxa, which is in different proportion as in the ambient seawater, indicating some functional significance of these horizontally acquainted microbes. Also, our correlation analyses demonstrated association of rare taxa with a few metabolite classes, hence non a passive transient permanence in the sponge (see L 508 – 523).

Here below we add again some bibliography that supports our findings on the relevance of the rare biosphere as microbial endosymbionts, and with large condivision with the seawater, hence likely horizontally transmitted into their host:

<https://doi.org/10.1038/ismej.2013.227>
<https://doi.org/10.21203/rs.3.rs-3365419/v1>

Finally, from personal observations, again, with another co-occurring LMA species in the same collecting areas, *Crambe crambe* and this other LMA sponges have very different microbiomes from each other, even when living on the same rock, and even if sharing a considerable amount of taxa with the same seawater!. Meaning that both co-occurring LMA sponge holobionts take up different players from the seawater to join their symbiotic microbiome.

This said, we edited this part as follows, to address reviewer's concern, while keeping our point of view in this topic: "*C. crambe* specimens, despite hosting low microbial abundances, revealed values of ASVs richness comparable to those of HMA sponges, but with lower diversity. This outcome suggests that microbial densities and taxa richness are not always correlated⁷², and highlights the relevance of the rare taxa to the overall diversity, especially in microbiota dominated by one or few symbionts⁷³." (see L 538 – 543).

Line 359. I agree that a large part of the microbiome can be acquired from seawater (horizontal), but the simple presence of seawater bacteria in a LMA sponge is not sufficient for this affirmation, mostly if it is only part of the rare microbiome. I think this would need additional information on how abundant were those ASVs in the water and in *C. Crambe*, whether they are part of the species core, etc. There should be some kind of enrichment from the sponge side to consider them HA symbiont and not just passing/food bacteria. See DOI:10.1038/s41598-018-33545-1.

As the previous comment, we already responded above: This is in fact a debating aspect in sponge microbiology. In our case, we have to say that all sponge samples were treated in the same way, rinsed thoroughly with sterile SW prior extractions, sampled in the same way, extracted the same day with the same protocol, etc... And all and all our HMA sponges did not behave in the same way as *Crambe crambe*, as they shared very few ASVs with the ambient water. This suggests some sort of sponge preference in the case of *Crambe* to horizontally retain a larger amount of ASVs shared with the seawater. Why would HMA sponges not share as many microbes with the

seawater under the same technical conditions? In addition, we still find a strong host specific signal in our sponge microbiomes, and in all cases significantly very different from the seawater. Beta diversity analyses yielded different microbial compositions between *C. crambe* and other sponges, and with the seawater. Hence, we believe this rare biosphere should play a role and be selectively retained in *Crambe crambe*. Consider that *Crambe crambe* was significantly different from the SW in microbial composition, even if its microbiome is strongly dominated by one single ASV. This difference with the SW is corroborated by several diversity methods, including methods based on presence absence (Jaccard), relative abundance based distances (Bray-Curtis) and compositionality based analyses (Aitchison). Hence, even if *C. crambe* uptakes a certain portion of microbes from the seawater, this process is not passive. This host selects in which proportion retaining these taxa, indicating some functional significance of these horizontally acquainted microbes. Also, our correlation analyses demonstrated association of rare taxa with a few metabolite classes (see L 508 – 523).

The reviewer suggests adding further information on how abundant the shared ASVs between *Crambe crambe* and the seawater are in either compartments, the water and in *C. Crambe*. This type of information, can be extracted from the significant differences obtained in Beta diversity based Bray-Curtis (so those non binary distances, non-based on presence absence, but accounting also on the relative abundance of taxa). Regarding the presence of shared ASVs with the seawater as part of *C. crambe* core, we do not really believe this is a necessary condition for a microbe to be a functional symbiont, as this should be applied for the HMA sponges too. In any case, eight of the ASV shared with the seawater are part of the 13 core taxa in *C. crambe*, while none of the core taxa in any of the 3 HMA sponges was shared with the seawater. So, as the reviewer can see, in *C. crambe*, the microbes acquired by horizontal transmission are relevant, maintained across all conspecific hosts, and represent the majority of core the taxa.

As above, here below we again add some bibliography that further supports our findings on the relevance of the rare biosphere as microbial endosymbionts, and with large condision with the seawater, hence likely horizontally transmitted into their host:

<https://doi.org/10.1038/ismej.2013.227>
<https://doi.org/10.21203/rs.3.rs-3365419/v1>

From our personal observations, again, with another co-occurring LMA species in the same collecting areas, *Crambe crambe* and this other LMA sponges have very different microbiomes from each other, even when living on the same rock, and sharing a considerable amount of taxa with the same seawater! ... Meaning that both co-occurring LMA sponge holobionts take up different players from the seawater to join their symbiotic microbiome.

This said, we edited this part as follows, to address reviewer's concern, while keeping our point of view on this topic: "... Instead, background taxa are mainly acquired horizontally from the water column^{73,77}. These may include microbes retained with high fidelity, as the eight core taxa shared with the seawater recorded in our *C. crambe*, as well as other rare ASVs, accounting for up to ~175, in these hosts. The functional role of rare taxa is poorly understood, but they might be involved in host acclimatization processes. In this sense, background and/or newly introduced taxa may become operationally relevant, providing new metabolic pathways to face changing conditions, or replace lost functionalities during symbiont shuffling⁷⁸⁻⁸⁰..” (see L 547 – 553).

We also included the following edits to support the above observations: In the Results section: “*C. nucula* displayed the largest core microbiome in terms of ASV richness (55 ASVs), followed by *C. reniformis* (47), *P. ficiformis* (38), and *C. Crambe* (13; Fig. 7). *Crambe crambe* was the only host sharing core taxa with the seawater, sharing eight core ASVs” (see L 401 – 403).

Line 430. I definitely do not understand this analysis and plot. Rhodothermaceae and PAUC34f points to *C. nucula* in the plot. I don't understand how is it associated to *C. crambe*.

The reviewer here refers to two different results, which do not yield exact the same interpretation for being one the previous step of the other: 1) is the DEICODE beta diversity compositional analysis on unrarefied CLR transformed data and Aitchison distance represented by a RPCA and showing vectors at the ASV level, and 2) is the unsupervised differential abundance analysis based on the ordination output from the previous RPCA to detect differentially abundant taxa at higher taxonomical levels than the ASV (Genus, when possible or higher in case of poor annotation), which mostly contribute to separate samples by sponge species co-variate. This second higher taxonomical level analysis allows get a better overview and provide better interpretations, as many other papers, discussing the microbiome differences by Genus, Family, Order, Class or Phylum, depending on the scope and data.

1: In the DEICODE beta diversity compositional analysis, the plotting outputs of the beta diversity compositional RPCA shows the twenty most relevant ASVs (so at the ASV taxonomy level) driving grouping separation in the ordination space in the PCA, and we reported in the legend the best annotated taxonomy for each of them. This said, the two ASVs mentioned by the reviewer point to *C. nucula* in the RPCA. But many other ASVs with the same taxonomy (family Rhodothermaceae and phylum PAUC34f) are likely present in the rest of the sponge species, but may represent different ASVs, and are not represented in the plot as vectors for not being among the top most 20 descriptive ASVs.

2: Then from the ordination object from RPCA, we performed an unsupervised differential abundance analysis, based on CLR transformed data (log-ratios), using Qurro visualization tool (<https://github.com/biocore/qurro>). With Qurro and an ordination object as input data, we can visualize the loadings of all the ASVs in the dataset (at ASV level), and select the numerator and denominator ASVs (with diverse selection options, by ASV name, agglomerated by taxonomy name, by most correlated to sample groupings...) to generate a microbial balance that best separates our sponge host samples by species, along axis 1, 2 or 3 (PC1, PC2 or PC3). We created microbial balances based on axis 2 (PC2), for being more resolutive, and we selected numerator and denominator taxa for the microbial balance with the option that allows to group ASVs by taxonomy levels. As said, higher taxonomical level analysis (by Genus, when possible, or higher in case of poor annotation) allows us get a better overview of the whole microbial community structure, while providing room for better interpretations. With the guide of the vectors from the RPCA plot representing the major ASV drivers in the ordination space and in sponge species separation, we chose the taxonomy levels according to the taxonomy of those 20 vectors, in numerator or denominator positions, depending on the direction of the vector arrows. This created our microbial balance by taxonomy levels (and not at the ASV level as the RPCA vectors), and this output is the one used to discuss the section “*Interspecific variability of sponge microbiomes*”, so the one the reviewer refers to. From all this long explanation, we hope the reviewer can see why an AVS that points to a host in the RPCA (e.g., Rhodothermaceae and PAUC34f points to *C. nucula*) can have the same taxonomy level as a differentially abundant genus, or family, or order, etc... that may be correlated to a different host species (e.g.; Rhodothermaceae, ..., PAUC34f and ... being mostly associated to *C. crambe*).

We understand these procedures are not straightforward to explain in a response letter, but as any statistical approach involving ‘omics and differential abundance for compositional data, hence we invite the reviewer to read on these methods, in case we were not still clear enough: <https://github.com/biocore/qurro>; <https://academic.oup.com/nargab/article/2/2/lqaa023/5826153?login=true> .

For further clarification, we modified the text in the following parts: Figure 5, legend “Figure 5. A) Compositional RPCA biplot of beta diversity of bacterial communities associated with the four sponge species based on Aitchison distance. Sponge species are depicted by colors and points represent individuals. Twenty most relevant ASVs driving differences in the ordination space are

illustrated by coloured vectors labeled with the respective taxonomy, reported to the lowest best taxonomic annotation. B) Log-ratios of the microbial balance generated by the selected bacterial consortium at several taxonomy levels plotted across the four sponge species along axis 2 of the compositional RPCA. C) Feature rankings, with the ASVs selected as numerator and denominator taxa to generate the microbial balance that maximized group separations by sponge species, along axis 2 of the compositional RPCA. * For other dimensional views, including the third ordination component PC3 of the RPCA plot (A) see Supplementary figure S5.” L 1388 – 1389

In the Results section: Log ratios from DEICODE feature-loadings grouped by lowest best taxonomy annotation level, and which best explained the separation of sample groupings along axis 2 in the RPCA space, were composed by 81 ASVs in the numerator and 50 in the denominator. “(L 356 – 359).

As a final explanation tip of our unsupervised differential abundance analysis here we provided a table summarizing of all the taxa involved in the analysis to create the microbial balance (numerator *over* denominator taxa) to show how they were distributed in our sponge holobionts. Note that the same symbiont strains can be present in several hosts, and also see how different strains can be grouped in a certain taxonomy level, and this higher bacterial group can be present across diverse host species.

NUMERATOR							
C.crambe	Bacteria	Acidobact	Subgroup	Class_Sub	Class_Sub	Class_Sub	Class_Subgroup 6
C.nucula	Bacteria	Acidobact	Subgroup	Class_Sub	Class_Sub	Class_Sub	Class_Subgroup 6
C.reniformis	Bacteria	Acidobact	Subgroup	Class_Sub	Class_Sub	Class_Sub	Class_Subgroup 6
P.ficiformis	Bacteria	Acidobact	Subgroup	Class_Sub	Class_Sub	Class_Sub	Class_Subgroup 6
C.crambe	Bacteria	Chlorofle	Anaerolin	SBR1031	A4b	Family_A	Family_A4b
C.nucula	Bacteria	Chlorofle	Anaerolin	SBR1031	A4b	Family_A	Family_A4b
C.reniformis	Bacteria	Chlorofle	Anaerolin	SBR1031	A4b	Family_A	Family_A4b
P.ficiformis	Bacteria	Chlorofle	Anaerolin	SBR1031	A4b	Family_A	Family_A4b
C.crambe	Proteobac	Deltaprot	Bdellovibi	Bdellovibi	Bdellovibi	Genus_Bdellovibri	
C.nucula	Proteobac	Deltaprot	Bdellovibi	Bdellovibi	Bdellovibi	Genus_Bdellovibri	
C.reniformis	Proteobac	Deltaprot	Bdellovibi	Bdellovibi	Bdellovibi	Genus_Bdellovibri	
P.ficiformis	Proteobac	Deltaprot	Bdellovibi	Bdellovibi	Bdellovibi	Genus_Bdellovibri	
DENOMINATOR							
C.crambe	Bacteria	Cyanobac	Oxyphoto	Synechoc	Cyanobiaceae		
C.nucula	Bacteria	Cyanobac	Oxyphoto	Synechoc	Cyanobiaceae		
C.reniformis	Bacteria	Cyanobac	Oxyphoto	Synechoc	Cyanobiaceae		
P.ficiformis	Bacteria	Cyanobac	Oxyphoto	Synechoc	Cyanobiaceae		
C.crambe	Bacteria	Bacteroid	Rhodothe	Rhodothe	Rhodothe	Family_Rh	Family_Rhodotherm
C.nucula	Bacteria	Bacteroid	Rhodothe	Rhodothe	Rhodothe	Family_Rh	Family_Rhodotherm
C.reniformis	Bacteria	Bacteroid	Rhodothe	Rhodothe	Rhodothe	Family_Rh	Family_Rhodotherm
P.ficiformis	Bacteria	Bacteroid	Rhodothe	Rhodothe	Rhodothe	Family_Rh	Family_Rhodotherm
C.crambe	Bacteria	PAUC34f	Phylum_P	Phylum_P	Phylum_P	Phylum_P	Phylum_PAUC34f
C.nucula	Bacteria	PAUC34f	Phylum_P	Phylum_P	Phylum_P	Phylum_P	Phylum_PAUC34f
C.reniformis	Bacteria	PAUC34f	Phylum_P	Phylum_P	Phylum_P	Phylum_P	Phylum_PAUC34f
P.ficiformis	Bacteria	PAUC34f	Phylum_P	Phylum_P	Phylum_P	Phylum_P	Phylum_PAUC34f
C.crambe	Bacteria	Proteobac	Gammapr	Betaprote	EC94	Family_EC	Family_EC94
C.nucula	Bacteria	Proteobac	Gammapr	Betaprote	EC94	Family_EC	Family_EC94
C.reniformis	Bacteria	Proteobac	Gammapr	Betaprote	EC94	Family_EC	Family_EC94
P.ficiformis	Bacteria	Proteobac	Gammapr	Betaprote	EC94	Family_EC	Family_EC94

Line 443 and 448. Still extremely confused about all this. This two sentences: "Within Proteobacteria, a dominant phylum in marine sponges genus *Bdellovibrio* and order EC94 were correlated to HMA species." vs. "Among bacteria mostly associated with *C. crambe*, we found the EC94 group(Betaproteobacteriales)". Is the EC94 the same in both sentences? How is it correlated to HMA and associated with *C. Crambe* (LMA) at the same time? Which one is it?

The reviewer is right in this comment, and we thank her/him for pointing this out! Yes, we made a mistake in the first sentence "Within Proteobacteria, a dominant phylum in marine sponges, genus *Bdellovibrio* and order EC94 were correlated to HMA species.". We wrongly added order EC94 as associated with HMA sponges as well. We have now corrected this in the text removing order EC94 as: "Within Proteobacteria, a dominant phylum in marine sponges^{7,26}, genus *Bdellovibrio* was correlated to HMA species" Please see L 635 – 636.

supplementary

Line 13. Correct: Supplementary figure S1 and figure S2

Figures were corrected as indicated. Please see Figs. S1-S5.

Line 47. What library construction was used?

Our library construction was described in the Supplementary material as the main text is already very lengthy. Please, consult our Suppl. file (Section 1.2). DNA extraction, amplification and sequencing; L 38 - 51, where we explain these procedures in more detail: "Quantity and quality of the extracted DNA from sponges and seawater filters were determined using a Thermo Scientific Nanodrop™ 1000. DNA aliquots were sent to Personal Genomics laboratories (www.personalgenomics.it) for amplicon library construction preparations and sequencing. For bacterial and archaeal diversity analysis, the V3-V4 hypervariable region of the 16S rRNA gene (*Escherichia coli* position: 341-805) was amplified using the bacterial and archaea universal primers: Bakt_341F 5'-CCTACGGGNGGCWGCAG-3', Bakt_805R 5'-GACTACHVGGGTATCTAATCC-3' following the Illumina protocol. Amplification reactions were prepared using 2.5 µL DNA (5 ng/µL), 5µL forward primer (1 µM), 5 µL reverse primer (1 µM) and 12.5 µL KAPA HiFi HotStart ReadyMix, for a total volume of reaction of 25 µL. PCR were performed in a Veriti™ 96-Well Thermal Cycler using the following thermocycling conditions: 95°C for 3 minutes followed by 25 cycles of 95°C for 30 seconds, 55°C for 30 seconds and 72°C for 30 seconds and a final elongation performed at 72°C for 5 minutes. After purification with AMPure XP beads the amplicons were analysed by a MiSeq sequencer (Illumina platform)."

We hope this answers this reviewer's inquiry.

Table S1. I would add info of the phenotypes of each sample.

Thank you for the observation, information about the phenotypes of each sample was added in the supplementary as suggested adding a note.

We added a note in the legend of the table S1 (as well as in the legend of the Table 1) clarifying the phenotypes of study: (L 158 SM) "*All samples in the same species are of the same phenotype:

pigmented and massive for *P. ficiformis* and *C. reniformis*, pigmented and lobular for *C. nucula* and encrusting for *C. crambe*, and all from illuminated habitats and analogous conditions.”

Reviewer #3 (Remarks to the Author):

This study by Mazzella et al. investigates the microbiomes and metabolomes of four ecologically and biologically variable co-occurring sponge species, with the aim of uncovering functional relationships to help to explain niche/ phenotypic diversification among hosts.

Sponges are a valuable and intriguing model system for host-microbiome studies and while these species are well-studied in terms of their microbiomes using 16S methods, this study constitutes a novel experimental design with the additional use of metabolomics and the ecological and biological diversity of the species from the same geographic locales. The methods employed are technically sound and reproducible from the microbiome perspective (I do not have expertise in metabolomics so cannot comment to this) and the manuscript is well written and well presented. I enjoyed the discussion and the suggestions made to link microbiome/ metabolome features to the biology and ecology of the sponge holobionts – however I also have some reservations (comment 2, below) about the limitations of the evidence for these links and the conclusions drawn. Overall this study is very interesting and adds to the current knowledge of the functional role of sponge microbiomes within an ecological context.

We appreciate the kind compliments from the reviewer, in particular his/her recognition about the novelty introduced by the experimental design with the use of metabolomics, and the good evaluation received for our applied methodologies in general, as well as for the discussion and the interpretations made linking microbiome/metabolome features to the biology and ecology of sponge holobionts. Additionally, it is delightful to know that the reviewer has enjoyed the reading, and has found our MS very interesting and timely to the current knowledge of the functional role of sponge microbiomes within an ecological context.

Specific comments:

1) Intraspecific variation in growth forms, and bleached/ colored forms correlating to microhabitat are described in the introduction for some of the species. However, the authors have not described which of these were sampled or if this was consistent within species. This is vital to know as these characters would be expected to influence the variables measured in this study.

Thank you for the observation, we apologize if this was not clear before. All the samples from the same species were collected with the same morphology, and sampling sites shared analogous conditions (light, temp, etc...). In the introduction we described the range of potential intraspecific variation in growth forms, color forms, habitat preferences, microbial densities, reproduction typology, predominant symbiont transmission modes, colouration, feeding strategies, habitat preferences, allelochemistry known. for the four target species, with the aim to correlate some aspects of microbiome structure and metabolome with these range of phenotypic potentialities. Our intention here was to introduce all those possible phenotypes described in nature, and try to find linking patterns potentialities with the microbiome compositions and the metabolome profiling across the four species, and further make correlations with these multi-omics datasets. Therefore, our scope in this paper is not to seek for intraspecific phenotypic changes, either in morphology or habitat in correlation to microbial and metabolic dynamics, as this is being object of upcoming contributions. Therefore, in our experimental design, we only include a single type of phenotype per species, this means that all samples in the same species are of the same phenotype and location: pigmented and massive for *P. ficiformis* and *C. reniformis*, and similarly pigmented and lobular for *C. nucula* and encrusting for *C. crambe*, and all from

illuminated habitats, with similar conditions in general. We clarified this in the text as follows: In the **Introduction**: “These species were selected on the basis of distinctive bio-ecological potential range traits, involving reproductive strategies, microbial abundance, preferred habitat, general morphology, coloration, growth-type and allelochemistry (summarised in Table 1 and Fig. 1)” (L 65 – 68). Also in (L 139 – 142). “to investigate the relationships between sponge microbiomes and metabolic signatures, in the context of host specificity, and considering several distinctive bio-ecological trait ranges, characterizing the four holobiont systems selected.”

” (L 147), Table 1 legend “**Table 1.** Summary of the main range of bio-ecological characteristics of the selected sponge species shown in Fig. 1 as living morphotypes.”. We added a note under the table clarifying the phenotypes of study: (L 149) “*All samples in the same species are of the same phenotype: pigmented and massive for *P. ficiformis* and *C. reniformis*, pigmented and lobular for *C. nucula* and encrusting for *C. crambe*, and all from illuminated habitats and analogous conditions.” **Methods** “All samples in the same species collected for our study were of the same phenotype, this is: pigmented and massive for *P. ficiformis* and *C. reniformis*, pigmented and lobular for *C. nucula* and encrusting for *C. crambe*, and all from illuminated habitats, with analogous conditions as specified in Fig. 1. ” (L 156 – 159),

” **Conclusions** “Our study performed on samples of a single phenotype appertaining to four different target species, allowed us to encounter trends in the microbiomes, metabolomes and/or their inter-omics interactions that seem to correlate with the potentiality of range traits for each sponge. “ (L 797 – 800),

In any case, we could encounter certain trends in the microbiomes, metabolomes and/or their interactions discussed over the text that seem to correlate with the potentiality of range phenotypic traits for each sponge. For instance, *P. ficiformis* has the most diverse microbiome with the smallest core and largest variable microbiome, reporting also the largest ASV condision with the seawater as HMA sponge. This dynamic microbiome may allow this species to colonize a wide range of habitats (dark, illuminated) adopting different morphologies (massive and pigmented to reticular and bleached). This can likely occur along shifting and shuffling processes of symbiont members, driving small adjustments of the metabolic profile, as can be suggested by the highest canonical correlation found in this species between microbiome and metabolome. *Chondrosia reniformis* instead, can colonize also a wide range of habitats, from dark to illuminated, with changes on its pigmentation, but with no changes in its morphology. This species revealed a large core microbiome, and is described as having a diverse but very stable microbiome across habitats and locations (as opposite of *P. ficiformis*). In this case, *C. reniformis* probably relies on high metabolic redundancy of its diverse but stable microbiomes for its ability to colonize a range of conditions with no apparent major phenotypic changes, as suggested by its low canonical correlation between microbiome and metabolome, as compared with *Petrosia*. *Crambe crambe* shows dominance of a single symbiont probably non shared with the seawater and likely obtained via vertical transmission, this reflecting its brooding reproductive nature. This host shares many taxa with the sweater, including the majority of its core taxa, which include cyanobacteria that correlate with its preference for photic to shaded zones. It further reveals inter-omics canonical correlations similar to *P. ficiformis*, thus demonstrating the functionality of the dynamic rare biosphere, while its largest condision with the seawater, along with certain association patterns to groups of bacteria link with its LMA condition. Finally, *C. nucula* has predominance of Cyanobacterial symbionts that explain its requirement for photic habitats to get photosynthates from these photosymbionts, as reflected in the predicted functions outcomes. It has the largest core, and the lowest condision with the seawater as HMA sponge, which may reflect larger vertical transgenerational transmission of endosymbionts, and slower intragenerational acclimatization capability.

2) Related to above - a central premise and novelty for the study is around the bio-ecological differences between and within the species. However the samples appear to have only been collected from shallow (i.e. light) habitats, which limits exploration of the intraspecific differences described. Despite this, the authors do discuss their results in the context of these intraspecific differences – I feel that without having sampled multiple habitats and morphologies, and with only correlative evidence, some of the conclusions overreach in this regard and some

further exploration of potential mechanisms in the discussion would be useful. E.g. the authors state regarding *P. ficiformis* that its microbiome variability may enable colonisation of different habitats. But different habitats and depths were not sampled here. Elsewhere it is suggested that *P. ficiformis* maintains heterotrophic bacteria related to its ability to colonise dark/ shaded areas – how and why would this symbiont relationship be maintained in light-dwelling individuals sampled here, especially if horizontal transmission is thought to be the main mode of symbiont acquisition for *P. ficiformis*, and therefore symbiont switching could allow phenotypic plasticity in this regard?

Thanks for this comment. We hope that what we wrote in the points above would be enough to clarify the reviewer's concerns.

3) The metabolome results offers interesting findings, but the authors seem to attribute this only to the microbiome in some areas of the text, but the sponge itself could potentially be producing some of these compounds?

The reviewer is perfectly correct here, yes, the sponge host could potentially be producing some of these compounds, although the lack of metabolomics published data for sponge holobionts is largely limiting. Fact that seriously hampers the capability to clearly attribute the production of many metabolites to either the host or its associated microbes. This is particularly true in compounds that have been reported from either holobiont compartments in literature (this is, compounds with no clear microbial or metazoan nature). Nonetheless, thanks to additional correlation analyses performed between microbiome and metabolome datasets, we were able to draw some trends of interconnection patterns between the microbial associates and the metabolites. These new approaches afforded indirect information about microbial vs host influence in metabolite fingerprinting. For instance, see in the results: L 480 – 523.

4) Predicted function for microbiome – this analysis is based on 16S taxonomy at often quite broad taxonomic levels (e.g. order). Taxonomy does not correlate well to function in many cases and so many sponge symbionts have not been cultured or had their genomes fully sequenced - this needs to be acknowledged in the text as a caveat.

To perform the predicted functions analyses we used the full taxonomic annotation, while to visualize the taxa barplot we used the order as the the best taxonomic rank shared among the four sponge species.

We are aware of this, and have acknowledged this method limitation in the text as suggested: “keeping in mind tough its limitations related to the taxonomy resolution”.

Furthermore we added a reference DOI: 10.34133/ehs.007 which explains FAPROTAX limitations. please see L 257.

5) The introduction gives detailed background information on the study system and context, but does not highlight clearly the knowledge gap addressed and the novelty of the study.

Thank you for pointing us need to highlight the novelty of the study and the knowledge gap on the topic, as suggested, we acknowledged these aspects in the introduction as “To the best of our knowledge ours is one of the few studies on this field analyzing microbiome-metabolomics...”. Please, see L 144 – 146.

6) In the methods, I would like to see more information about relationships that were investigated in the final section on statistics, even if the statistical tests themselves cannot be described due to

space limitations. I understand the journal style is to keep the methods section short, so perhaps it's not possible though.

We did our best to address the requests of the reviewer and provide more information about relationships that were investigated in the statistics section in the main text in the Methods section LL 288-325; 261-270; 243-259 as well as in the Supplementary Tables.

7) Results -I prefer to see the exact p values in the results (except when $p < 0.001$) – currently they are just given as < 0.05 when significant.

As the majority of the p-values were $p < 0.001$ we modified the statement “ $p < 0.05$ ” by replacing $p < 0.001$ in the result sections. All the exact values are reported in the supplementary stats tables.

Line by line comments:

Line 67 – Consider substituting ‘resumed’ for ‘summarised’

Edited as suggested.

Line 77 – Consider substituting ‘either group’ for ‘both groups’

Corrected as suggested.

Line 89 – Please specify in what way they are more variable (intraspecifically, spatially, temporally?)

As pointed out, we specified in which way microbial symbiotic assemblages are likely more variable when horizontally transmitted as “Assemblages acquired horizontally are in general more spatially and temporally variable (depending on time, location and environmental conditions), and can occur simultaneously involving similar taxa in diverse host species” (L 89 – 90).

Line 98 – Many readers may be unfamiliar with the sponge loop so this should be defined/described briefly here

We briefly described the principles of the Sponge Loop, as “DOM is filtered by sponges and subsequently released as POM in the form cellular detritus, thus becoming available for the neighboring trophic web levels”. Please, see L 100 – 101.

Line 99 – ‘others’ instead of ‘other’

Changed as indicated.

Line 102-110 – It was initially difficult to see where this was going until the final sentence. (“All these microhabitat acclimating aspects could be related to host- microbiome-metabolome interactions.”). I feel introducing this idea prior to describing the study species microhabitats would be better for reader navigation.

Thank you for this indication, we can be more focused yes... we re-phrased that part of the text for better flow, as follows: “Microhabitat acclimating aspects could be related to host-

microbiome-metabolome interactions, and their joint potential. *Petrosia ficiformis* and *C. reniformis*, for instance, flourish in large populations, either in illuminated areas as pigmented forms, or in shaded or even dark caves as bleached morphotypes often in reticulated growth (e.g., *P. ficiformis*)^{33,34}. While *C. crambe* seem more abundant in the presence of some light or at most shade but not dark, and are colourful and charged of secondary metabolites to deter putative predators and competitors^{35,36}. *Chondrilla nucula* still, are clearly photophile and intolerant to shade-to-dark conditions^{37,38}. There are still many gaps of knowledge on the metabolic functional role of diversified microbiomes in how these and other holobionts are adapted to similar or distinct environments²⁸⁻³².”. Please, see **L 103 – 112**.

Line 127 – Should be ‘supplying’?

Edited as suggested, **L 114**.

Table 1 – It is not clear why there is both ‘Morphology’ and ‘Growth type’ columns – some information is shared between these.

We agree that morphology and Growth type can be unified in the same column. We modified the table accordingly, see **L 147, Table 1**.

Line 164-165 – Please give references for primer sequences

References were added as indicated, thank you for the observation, **L 182**.

Line 177 – Species names are not in italics

Species names were changed to italics.

Line 209 – This part is a little unclear – does it mean 351900 reads across the sponge samples and 56304 reads across the seawater samples? I don’t think it’s necessary to include this as we already have the number of reads per sample given which is more informative than the cumulative total across sample types.

Thank you for pointing this out, we modified for clarification this part of the text as follows “Sequence data were rarefied to 7038 reads (Fig. S1) per sample resulting in 408204 reads distributed over 58 samples (samples details in Table S1).”, please, see **L 338 – 339**

Line 211 – Not clear what ‘feature’ means here

In this context we meant by feature XXX ASV, we specified better in the text “Sequence reads ranged between a minimum of 7 and a maximum of 64215, distributed over 1181 ASVs.”. Please, see **L 339 – 340**.

Line 212-213 – Standard error would be nice to see here for the mean ASV numbers

Standard errors were shown in a supplementary figure, see **Supplementary Fig. S4**.

Line 360-361 – “But they might be involved in host acclimatization processes, via symbiont shuffling –becoming functionally relevant”. It would be good to add a little more explanation for clarity.

Further explanatory details were added to this idea, as indicated: “... The functional role of rare taxa is poorly understood, but they might be involved in host acclimatization processes. In this sense, background and/or newly introduced taxa may become operationally relevant, providing new metabolic pathways to face changing conditions, or replace lost functionalities during symbiont shuffling⁷⁸⁻⁸⁰.”. Please, see L 549 – 553.

Line 382, line 420 – Suggest to replace “researches” with “studies”

We replaced “researches” with “studies” as suggested.

Line 542 - “may allow it to colonize habitats with certain light exposure”- this is a little vague, please be specific on light exposure.

As requested, we modified this statement in the text as : “may allow it to colonize habitats with wide-range of light exposure.”. Please, see L 803 – 804.

Figure 2 – Scale bars needed for two left side maps. The lower-left map does not seem to correlate to the box highlighted in the upper left map? Labelling A B and C might help with orienting the reader

All corrections above were included in new Figure 2 as indicated, Fig. 2.

Figure 4 - It would be good to see error bars for the per sample mean ASVs. I am unsure why inertia ellipses have been used for the Jaccard NMDS but not the Bray-Curtis NMDS. The inertia ellipses and what they show needs mentioning in the legend. The figures themselves are not labelled A B C D.

We provided error bars for sample mean ASVs in a supplementary figure. Please See SM Figure S4. Instead ellipses were non provided in the bray-curtis plot as their visualization was tricky due to the points overlapping in the plot. Furthermore, we provided the following plot showing the Bray-Curtis PCoA with the ellipses overplotted on the sample clouds.

Ellipses show 95% confidence interval. We stated this in the legend of the plot as requested. L 1379 – 1380.

Figures 4, 5, 6, 7, 8, 9- Species names need to be italicised, and underscores removed from variable names.

As suggested, species names were italicised and underscores from variable names removed in new Figures 4, 5, 6, 7, 8, 9 as indicated, **Figs. 4, 5, 6, 7, 8, 9.**

The authors very much appreciate the constructive comments raised by all the reviewers, which have notoriously helped to improve the quality and presentation of our manuscript. We hope this new version meets all the requirements for acceptance.

Looking forward to hearing from you soon,

Yours sincerely,

Laura Núñez-Pons and Co-workers

Reviewers' comments:

Reviewer #1 (Remarks to the Author):

In the revised manuscript, the authors have addressed the majority of my concerns. They have made a commendable effort to incorporate additional correlation methods to integrate the microbiome and metabolome datasets, presenting a few specific and intriguing examples. Furthermore, the authors have provided detailed information in the analysis and statistical methods section and have introduced a graphic abstract to illustrate the central concept.

Nevertheless, the revised version contains several instances of filler words like [REF] where new text has been inserted, noticeable at Line 340, 342, 549, and 782.

Additionally, at Line 779, the abbreviations for "Cer" or "DG" have not been defined.

Reviewer 3 has expressed concern that the current sampling collection does not capture the impact of intra-species variation on microbiome diversity and function across diverse habitats. In response, the authors have adjusted their focus, placing less emphasis on intra-species variation and instead concentrating on a more targeted comparison and discussion of species-level variation and evolutionary strategies. Consequently, I think the response should adequately address the reviewer's major concern. Additionally, the authors have addressed other minor concerns raised by Reviewer 3

Reviewer #2 (Remarks to the Author):

Authors have solved some of the main issues I presented, mostly the confusion with the type of phenotype included in the sampling. They have improved considerably the methods section, but I still have few comments.

First, I need to point out that the version of the manuscript with track changes (19139_1_revised_manuscript_marked_up_590624_s4ml5l.docx) and the clean version (19139_1_art_file_590617_s4ml5k.docx) are not the same, which has been quite confusing. For instance, in the version with track changes (the one I chose to read to see more easily the modifications), authors left unfinished information like "(REF)" many times in the text, together with typos, different fonts and even contrasting word use such as vanishment vs. banishment in line 738 of the clean version. The authors should be careful uploading the same exact version of the manuscript with and without tracks.

My general comments on the rebuttal and/or new text are:

- Regarding the presence of bacteriocytes in *C. reniformis*. I see that authors decided to remove that claim, which I believe is the most cautious solution considering the lack of more visual evidence to prove it. However they still provide references that dispute this both in the rebuttal and in the manuscript. I will recommend the authors to double check the references they are providing. As I commented before, cells charged with bacteria during the reproduction of the sponge are in fact nurse cells that will nourish the eggs with bacteria and yolk during development, and can be found in the adult tissue, around the oocytes/embryos or even inside the reproductive forms for a while. That does not mean that the species present bacteriocytes in the mesohyl during the none reproductive periods.

Therefore, double check these references you provided:

Maldonado, M. Embryonic development of verongid demosponges supports the independent acquisition of spongin skeletons as an alternative to the siliceous skeleton of sponges. *Biol J Linn Soc* 97, 427–447 (2009).

This is most likely describing nurse cells charged with symbiotic bacteria.

Ereskovsky, A. V., Gonobobleva, E. & Vishnyakov, A. Morphological evidence for vertical transmission of symbiotic bacteria in the viviparous sponge *Halisarca dujardini* Johnston (Porifera, Demospongiae, Halisarcida). *Marine Biology* 146, 869–875 (2005).

This paper describes *H. dujardini*, it only refers to *C. reniformis* through the Levi and Levi 1976 work (same as the reference you had from Sara 1998 in the first submission). As I said before, Levi and Levi 1975 only mentions follicular cells around the oocytes, which are in fact the nurse cells only.

Boury-Esnault, N. Order Chondrosida Boury-Esnault & Lopès, 1985. Family Chondrillidae Gray, 1872. in *Systema Porifera* 291–297 (Springer, Boston, MA, 2002). doi:10.1007/978-1-4615-0747-5_32. I don't even see any mention of bacteriocytes in this work.

- Regarding the idea of contaminant bacteria in *C. crambe*.

When I was reading the text I found this idea in many sentences throughout the text, so I reported all of them to be sure that authors would not miss any mention if they had to modify that. You replied to my comment in a very general manner on the first time I pointed out the issue, therefore there is no need to copy/paste the same text for every single response, you can simply say "this has already been addressed above" and avoid repeating the same lines and making the rebuttal unnecessarily long.

This said, I was not challenging the idea that *C. crambe* takes its symbionts from the water, in fact, except for its dominant symbiont that seems to be vertically transmitted, the rest is probably acquired from the water and finds an optimal niche inside the sponge to grow. It is also common that LMA species share more taxa (including their core members) with the water, while HMA usually harbour a more sponge-specific community.

This subject is really difficult to resolve and I believe it may depend on the abundances of the ASVs in both environments (water and sponge). If the shared ASVs is more abundant (relative to the whole community) in the sponge, I believe they can be symbionts, that are positively selected from the water or overgrow inside the sponge. But if the shared ASV is very abundant in the water but only represents a "rare" fraction in the sponge, then it is likely to be passing bacteria.

I asked before what were those percentages for the core shared ASVs in the sponge and water, but I still don't have an answer.

(See previous comment: Line 359. I agree that a large part of the microbiome can be acquired from seawater (horizontal), but the simple presence of seawater bacteria in a LMA sponge is not sufficient for this affirmation, mostly if it is only part of the rare microbiome. I think this would need additional information on how abundant were those ASVs in the water and in *C. crambe*, whether they are part of the species core, etc. There should be some kind of enrichment from the sponge side to consider them HA symbiont and not just passing/food bacteria. See DOI:10.1038/s41598-018-33545-1.). I still would like to see those numbers.

The argument that all sponges were treated in the same way and only *Crambe* shared ASVs with the water is not fully convincing. First, the rinsing step has not been proved to efficiently remove passing bacteria. As soon as the sponge is disturbed during the sampling, they stop pumping, and therefore sea water most likely stays in the channels and filtering chambers until the tissue is fixed, and ultimately sequenced. The fact that LMA sponges present a smaller symbiotic community and filter larger volumes of water, makes them to have a larger proportion of passing bacteria vs. symbiotic

community. This could be ultimately identified in the sequencing as rare taxa, while in HMA sponges with large numbers of associated bacteria, and lower numbers of passing bacteria are not so easily detected.

As none of this has been proven unambiguously, I am happy for the authors to share their idea, I just wanted to advise that your results and arguments are not a reliable proof of that.

- I think I overlooked before the classification of the dominant symbiont in *Crambe*. It has been lately identified as Gammaproteobacteria, order Ca. Tethybacterales. Double check that.

See: Taylor JA, Palladino G, Wemheuer B, Steinert G, Sipkema D, Williams TJ, et al. Phylogeny resolved, metabolism revealed: functional radiation within a widespread and divergent clade of sponge symbionts. *ISME J.* 2021;15:503–19.

- One last thing that does not affect this work directly but relates with your future studies on the phenotypes as you commented. You have noticed that *Petrosia* species present the most diverse microbiome, be aware that this species has been described to present low gene flow, somehow close to a speciation event. Maybe your differently adapted phenotypes belong to isolated populations that do harbour different microbiomes.

See: 10.1111/mec.15635

Napoli, 28th January 2024

Dear Reviewers,

Our manuscript entitled: “**High microbiome and metabolome diversification in coexisting sponges with different bio-ecological traits**” submitted to the journal Communications Biology (Ms. Ref. No.: COMMSBIO-23-2829A) has been revised attending to the raised inquiries by the reviewer 2. We hope to have addressed satisfactorily all the comments and discussed all the interesting questions raised.

Please, find our detailed responses (in black), underneath each reviewer’s suggestions (in blue). You may locate the edits in the CLEAN MS “Article File” by searching the indicated line numbers as “L XXX – XXX”. Reviewers could optionally compare the edited changes in a REVISED MARKED MS (“Revised Manuscript with Track Changes”), sent separately as a supplementary file.

Reviewers' comments:

Reviewer #1 (Remarks to the Author):

In the revised manuscript, the authors have addressed the majority of my concerns. They have made a commendable effort to incorporate additional correlation methods to integrate the microbiome and metabolome datasets, presenting a few specific and intriguing examples. Furthermore, the authors have provided detailed information in the analysis and statistical methods section and have introduced a graphic abstract to illustrate the central concept.

Thank you very much for these words, and for understanding all the worth efforts made in our new analyses. We have to acknowledge the previous suggestions instigating us to pursue an upgraded way of presenting our data. We are really delighted of the result and the nice comments received.

Nevertheless, the revised version contains several instances of filler words like [REF] where new text has been inserted, noticeable at Line 340, 342, 549, and 782.

These [REF] inserts were particles that we used to localize where to include new entries in the reference list, and that remained in the text of the MARKED revised manuscript unintentionally. They should have been removed in the previous revised CLEAN MS version. In any case, we have revised the new version to avoid the occurrence of any of these typos. Thank you!

Additionally, at Line 779, the abbreviations for "Cer" or "DG" have not been defined.

All the definitions to the abbreviated metabolite classes were defined in the previous section, in their first instance appearance, this is Section “Metabolomics profiles of sponge holobionts”.

Again, we sincerely appreciate the invaluable input provided by Reviewer 1, whose constructive comments have significantly enhanced the quality of our manuscript. Her/his thoughtful feedback has been crucial in refining and strengthening our work. We are grateful for considering our reviewing effort “commendable” and for the time invested in reviewing our manuscript.

Reviewer #3 has expressed concern that the current sampling collection does not capture the impact of intra-species variation on microbiome diversity and function across diverse habitats. In response, the authors have adjusted their focus, placing less emphasis on intra-species variation

and instead concentrating on a more targeted comparison and discussion of species-level variation and evolutionary strategies. Consequently, I think the response should adequately address the reviewer's major concern. Additionally, the authors have addressed other minor concerns raised by Reviewer 3

We express our gratitude to reviewer 3 for helping us improve the overall focus of our manuscript. Furthermore, other minor comments were highly valuable to ameliorate the presentation of our work and interpret the outcomes.

Reviewer #2 (Remarks to the Author):

Authors have solved some of the main issues I presented, mostly the confusion with the type of phenotype included in the sampling. They have improved considerably the methods section, but I still have few comments.

Authors acknowledge the wise suggestions provided regarding the phenotype focus, and also the recommendations to improve the Methods presentation. Our MS has gained solidness with most of the points raised by all three reviewers. We also appreciate the interesting discussions taken over in the rebuttal conversations with the reviewer. We believe that the collaborative efforts of the Reviewer 2 have greatly contributed to the overall improvement of our present work and ongoing and upcoming research, thanks to the constructive debates that we continue in the lines below.

First, I need to point out that the version of the manuscript with track changes (19139_1_revised_manuscript_marked_up_590624_s4ml5l.docx) and the clean version (19139_1_art_file_590617_s4ml5k.docx) are not the same, which has been quite confusing. For instance, in the version with track changes (the one I chose to read to see more easily the modifications), authors left unfinished information like "(REF)" many times in the text, together with typos, different fonts and even contrasting word use such as vanishment vs. banishment in line 738 of the clean version. The authors should be careful uploading the same exact version of the manuscript with and without tracks.

We apologize, the mentioned [REF] inserts were particles that we used to localize where to include new entries in the reference list, and that remained in the text of the MARKED revised manuscript unintentionally, as probably other typos pointed out. They should have been removed in the previous revised CLEAN MS version. In any case, we have revised the new version to avoid the occurrence of any of these artifacts. Thank you!

My general comments on the rebuttal and/or new text are:

- Regarding the presence of bacteriocytes in *C. reniformis*. I see that authors decided to remove that claim, which I believe is the most cautious solution considering the lack of more visual evidence to prove it. However they still provide references that dispute this both in the rebuttal and in the manuscript. I will recommend the authors to double check the references they are providing.

Yes, we agreed to be cautious, following the reviewers' points. Further observations may provide accurate evidence on this argument.

As I commented before, cells charged with bacteria during the reproduction of the sponge are in fact nurse cells that will nourish the eggs with bacteria and yolk during development, and can be found in the adult tissue, around the oocytes/embryos or even inside the reproductive forms for a

while. That does not mean that the species present bacteriocytes in the mesohyl during the none reproductive periods.

Therefore, double check these references you provided: Maldonado, M. Embryonic development of verongid demosponges supports the independent acquisition of spongin skeletons as an alternative to the siliceous skeleton of sponges. *Biol J Linn Soc* 97, 427–447 (2009). This is most likely describing nurse cells charged with symbiotic bacteria.

Ereskovsky, A. V., Gonobobleva, E. & Vishnyakov, A. Morphological evidence for vertical transmission of symbiotic bacteria in the viviparous sponge *Halisarca dujardini* Johnston (Porifera, Demospongiae, Halisarcida). *Marine Biology* 146, 869–875 (2005). This paper describes *H. dujardini*, it only refers to *C. reniformis* through the Levi and Levi 1976 work (same as the reference you had from Sara 1998 in the first submission). As I said before, Levi and Levi 1975 only mentions follicular cells around the oocytes, which are in fact the nurse cells only.

Boury-Esnault, N. Order Chondrosida Boury-Esnault & Lopès, 1985. Family Chondrillidae Gray, 1872. in *Systema Porifera* 291–297 (Springer, Boston, MA, 2002). doi:10.1007/978-1-4615-0747-5_32.

I don't even see any mention of bacteriocytes in this work.

We are aware of the reviewer's experience in this field. For this reason we decided to remove all the ambiguous sentences and citations about bacteriocytes in *C. reniformis*. Please see (L531 - 533): "Concerning the sponge *C. reniformis*, our observations did not reveal the presence bacteriocytes. Previous studies instead, reported nurse cells charged with symbiotic bacteria during the development 20." Furthermore, we double checked the references provided. This last reference (doi:10.1007/978-1-4615-0747-5_32) in particular states about bacteriocytes, (right column; pp-293; see snapshot below).

Fig. 1. *Chondrosia*. A, in situ photograph of a Mediterranean specimen of *Chondrosia reniformis* (J. Vacelet) (scale 0.5 cm). B, microphotograph of histological slide from the Lendenfeld collection BMNH 96.11.5.109 (abbreviations: CO, cortex; CH, choanosome; sp, foreign spicule; arrow, cribriporal chone) (scale 400 μ m).

Description of type species

Chondrosia reniformis Nardo, 1847b (Fig. 1).

Synonymy. *Chondrosia reniformis* Nardo, 1847b: 267; Schmidt, 1862: 40; *Gummina ecaudata* Schmidt, 1862: 38; *Gummina gilricauda* Schmidt, 1862: 38; Schulze, 1877: 13; Topsent, 1895: 568; Topsent, 1925c.

Material examined. Holotype: Unknown – North Adriatic Sea. Neotype: MNHN DNBE 2000; BMNH: 96.11.5.112 (slides) – coll. February in Trieste, Lendenfeld Adriatic sponges; BMNH: 96.11.5.109 (radial serial section, thick) – Trieste, Collection Rao; BMNH: 33.3.1.45 – Napoli. Other material. Personal collection – specimens from Western Mediterranean (Provence Coast). *Gummina ecaudata* Schmidt – evidently from Schmidt species, Adriatic. Slides from BMNH (labelled *Chondrosia reniformis*) – doubtful or evidently misinterpretations: BMNH 158.12.29.127 – Hebrew University collection, Eylath Red Sea; BMNH 1954.2.23.37 – Herdman's Ceylon sponges; BMNH 1936.11.2b.8a – Reef bay, Pt Elizabeth South Africa, 5.7.36, Presd. Prof. JA Stephenson; BMNH 32.4.5.16a – Dry Tortugas, Dendy collection; BMNH 1939.5.8.56 – off Tampa Bay, USA, Belgian Museum Coll. 'Mercator'.

Description. Specimens are generally lobate and can reach 30 cm in greatest dimension and 3 cm thick. The colour in life is from black on parts exposed to the light to white on parts unexposed to light. The consistency in life is cartilaginous, firm, and tough. The surface is smooth, shiny (Fig. 1A) (features taken from living Mediterranean specimens). The upper cortex is composed of two layers: one superficial layer with numerous spherulous cells, one internal layer with few spherulous cells. The basal cortex shows a lower density of spherulous cells and the external zone is devoid of any spherulous cells. Spherulous cells are abundant also around canals. The density of the spherulous cells in the choanosome is similar to that of the internal layer of the upper cortex. The spherulous cells contain about 20 spherules of about 3 μ m in diameter. Cortex is also characterised by a net of large fascicles of collagen fibrils. Foreign

spicules are present in the choanosome and in the internal layer of the cortex. The border cortex/choanosome is underlined by a line of spherulous cells (Fig. 1B). Choanocyte chambers ovoid to spherical are about 40 μ m in diameter with a small aphodus at the apople of the chamber. Extracellular symbiotic bacteria and bacteriocytes are present in the mesohyl (anatomical description taken from neotype and other slides made by Lendenfeld in 1896 from Adriatic specimens).

Reproduction. The type species is oviparous (Scalera-Liaci *et al.*, 1971). Oogenesis occurs from May to August and spermatogenesis from July to August. The emission of the spermatozooids and oocytes occurs between the first quarter and the full moon of August (Lévi & Lévi, 1976). The oocytes remain trapped in follicular cells near the basis of the sponge. The fecundation is external. The larva is a blastula. During embryogenesis, there is a transmission of bacteria and spherulous cells from the mother-sponge to the larva (Lévi & Lévi, 1976).

Remarks and distribution. We consider that only populations of *Chondrosia reniformis* from the Mediterranean and the nearest Atlantic (Coasts of Spain, Portugal and Morocco) (Lazoski *et al.*, 2001) belong to this species. Specimens from localities outside the Mediterranean vary principally in the localisation and abundance of foreign materials and spherulous cells and are certainly different species (Lazoski *et al.*, 2001).

CHONDRILLA SCHMIDT, 1862

Synonymy

Chondrilla Schmidt, 1862: 38. *Magog* Sollas, 1888: 442 (for *Chondrilla sacciformis* Carter, 1879b: 299); de Laubenfels, 1936a: 182; Topsent, 1895: 512; Topsent, 1918: 601. *Chondrillastra* Topsent, 1918: 603 (for *C. australiensis* Carter, 1873b); de Laubenfels, 1936a: 182.

As far as we understand this sentence has been taken from a very old description (Lendenfeld, 1896). We tried to translate the original statement (found at: <https://www.marinespecies.org/charms/aphia.php?p=sourcedetails&id=7827>) which seems to be: I found parasitic algae in several specimens [R. von Lendenfeld (1889, p. 461 [sep. p. 56], plate XXVIII, fig. 89, 90)], which occur in large hydatid-like bubbles.

“In mehreren Exemplaren habe ich parasitische Algen gefunden [R. von Lendenfeld (1889, p. 461 [sep. p. 56], Taf. XXVIII, Fig. 89, 90)], welche in grossen hydatidenartigen Blasen vorkommen.” As none of us is a German speaker and since these investigations are very ancient we rely on the reviewer's experience. Nonetheless, we strongly believe that this has been an intriguing discussion, and we are delighted with the Reviewer for sharing his knowledge and guiding us in the writing of this part of the manuscript.

- Regarding the idea of contaminant bacteria in *C. crambe*. When I was reading the text I found this idea in many sentences throughout the text, so I reported all of them to be sure that authors would not miss any mention if they had to modify that. You replied to my comment in a very general manner on the first time I pointed out the issue, therefore there is no need to copy/paste the same text for every single response, you can simply say "this has already been addressed above" and avoid repeating the same lines and making the rebuttal unnecessarily long.

Yes, we might have taken the “answers point by point” too seriously, and followed to the letter, we apologize for the length, it was a long response letter.

This said, I was not challenging the idea that *C. crambe* takes its symbionts from the water, in fact, except for its dominant symbiont that seems to be vertically transmitted, the rest is probably acquired from the water and finds an optimal niche inside the sponge to grow. It is also common that LMA species share more taxa (including their core members) with the water, while HMA usually harbour a more sponge-specific community. This subject is really difficult to resolve and I believe it may depend on the abundances of the ASVs in both environments (water and sponge). If the shared ASVs are more abundant (relative to the whole community) in the sponge, I believe they can be symbionts, that are positively selected from the water or overgrow inside the sponge. But if the shared ASV is very abundant in the water but only represents a “rare” fraction in the sponge, then it is likely to be passing bacteria. I asked before what were those percentages for the core shared ASVs in the sponge and water, but I still don't have an answer.

We provide numbers and answers below.

(See previous comment: Line 359. I agree that a large part of the microbiome can be acquired from seawater (horizontal), but the simple presence of seawater bacteria in a LMA sponge is not sufficient for this affirmation, mostly if it is only part of the rare microbiome. I think this would need additional information on how abundant were those ASVs in the water and in *C. crambe*, whether they are part of the species core, etc. There should be some kind of enrichment from the sponge side to consider them HA symbionts and not just passing/food bacteria. See DOI:10.1038/s41598-018-33545-1.).

I still would like to see those numbers.

Thank you for these close observations. Learning from the past, we will answer both comments in a single paragraph, as they are related to the same argument. We provide a screenshot of a table with the percentage relative abundances of the eight core ASVs shared in *Crambe crambe* sponge hosts and the seawater:

	A	B	C	D	E	F	G	H	I	J	K	L	M	N	O	P	Q	R	
1	Sample	ASV	DNA.CCO	DNA.CCO	DNA.GF.S	DNA.GF.S	DNA.GF.S	DNA.SA.S	DNA.SA.S	DNA.SA.S	DNA.SA.S	DNA.SA.S.SW3.165							Average
2	Seawater	3af6c38e	6,820119	4,887752	1,974992	2,046035	2,017618	11,75049	11,53736	13,17135								6,775717	
3	Seawater	55a4efd0	0,156294	0,056834	0,056834	0,127877	0,071042	0,142085	0,099460	0,142085								0,106564	
4	Seawater	742ced2a	0,341005	0,369423	0,397840	0,298380	0,497300	0,156294	0,255754	0,269963								0,323245	
5	Seawater	7d2abbec	0,710429	0,568343	0,696220	0,568343	0,539926	0,568343	0,653594	0,682011								0,623401	
6	Seawater	a087e285	2,429667	2,017618	5,967604	5,626598	4,731457	2,017618	2,429667	3,168513								3,548593	
7	Seawater	e8479614	0,682011	0,966183	1,832907	2,202330	1,861324	0,781472	0,852514	1,023017								1,275220	
8	Seawater	e8d30aee	4,987212	5,015629	1,861324	1,591361	1,520318	3,339016	3,040636	4,006820								3,170289	
9	Seawater	eddec924	1,946575	1,960784	0,767263	0,554134	0,568343	1,662404	1,520318	1,790281								1,346263	
10																			
11			DNA.CCO	DNA.CCO	DNA.CCO	DNA.CCO	DNA.GFC	DNA.GFC	DNA.GFC	DNA.GFC	DNA.GFC	DNA.SA.C	DNA.SA.C	DNA.SA.C	DNA.SA.C	DNA.SA.C	DNA.SA.C	DNA.SA.CCS.165	
12	C.crambe	3af6c38e	0,809889	0,525717	0,369423	0,667803	0,184711	0,056834	0,099460	0,099460	0,156294	0,326797	0,866723	0,937766	0,468883			0,428443	
13	C.crambe	55a4efd0	0,269963	0,156294	0,170502	0,099460	0,127877	0,042625	0,071042	0,156294	0,071042	0,255754	0,014208	0,113668	0,213128			0,135528	
14	C.crambe	742ced2a	0,880932	0,724637	0,738846	0,653594	0,412048	0,539926	0,667803	0,781472	0,596760	0,696220	0,539926	0,525717	0,241545			0,615340	
15	C.crambe	7d2abbec	2,216538	1,463483	0,937766	1,136686	1,023017	1,548735	1,520318	1,264563	1,079852	1,477692	0,951974	1,548735	0,568343			1,287516	
16	C.crambe	a087e285	3,253765	1,889741	1,065643	1,420858	2,600170	2,827507	3,211139	3,708439	1,804489	0,809889	1,008809	1,918158	0,753054			2,020897	
17	C.crambe	e8479614	2,315998	1,150895	1,008809	1,278772	1,236146	2,102870	2,287581	2,216538	1,491901	1,136686	0,468883	1,292980	0,568343			1,427416	
18	C.crambe	e8d30aee	4,949985	2,728047	1,477692	1,463483	0,525717	0,582551	0,738846	0,582551	0,454674	1,335606	0,795680	2,529127	0,696220			1,373860	
19	C.crambe	eddec924	2,813299	2,330207	1,094060	1,491901	0,554134	0,539926	0,539926	0,710429	0,525717	1,051435	0,653594	1,690821	0,653594			1,126849	

As we can see, the shared core ASVs between *C. crambe* and the seawater are not in any case dominant in the seawater, being the largest average percentage 7 % (this was also true in *C. crambe*, as this sponge has its dominant taxon quite faithful and predominant > 70 %). In most cases, the ratios of relative abundance were maintained between seawater and *C. crambe* sponge samples, and this could be considered as an enrichment, since the potential amount of seawater left in the sponge samples (in the filtering system) upon DNA extraction is negligible with respect to the sponge tissue. But further, the amount of microbes present, diluted naturally in the seawater and potentially remaining within the sponge channels is not comparable with the seawater microbes present in the filters, which were concentrated for the sequencing. As a conclusion, we consider the core microbes 100% shared between seawater and *Crambe crambe* as likely symbionts, while the other rare microbes shared with the seawater, we do not have evidence, but we believe some could also be assigned to a symbiotic classification. All this said, we provided clarification according to the reviewer concerns as follows: “This outcome suggests that microbial densities and taxa richness are not always correlated 111, and highlights the relevance of the rare taxa to the overall diversity, especially in microbiota dominated by one or few symbionts (but see below) 112. In LMA microbiomes dominated by one or few ASVs (as in *C. crambe*) dominant microbes are transmitted from adults to embryos with high fidelity thanks to brooding reproduction, as these symbionts are essential for fitness 25,113–115. Instead, background taxa are mainly acquired horizontally from the water column 112,116. These may include symbiont microbes retained with high fidelity, as the eight core taxa shared with the seawater recorded in our *C. crambe*, as well as other rare ASVs, some potentially representing transient microbes or food, accounting for up to ~175, in these hosts.” (L 544 – 553)

The argument that all sponges were treated in the same way and only *Crambe* shared ASVs with the water is not fully convincing. First, the rinsing step has not been proved to efficiently remove passing bacteria. As soon as the sponge is disturbed during the sampling, they stop pumping, and therefore sea water most likely stays in the channels and filtering chambers until the tissue is fixed, and ultimately sequenced. The fact that LMA sponges present a smaller symbiotic community and filter larger volumes of water, makes them to have a larger proportion of passing bacteria vs. symbiotic community. This could be ultimately identified in the sequencing as rare taxa, while in HMA sponges with large numbers of associated bacteria, and lower numbers of passing bacteria are not so easily detected.

As none of this has been proven unambiguously, I am happy for the authors to share their idea, I just wanted to advise that your results and arguments are not a reliable proof of that.

Thank you for opening this interesting debate, as these aspects are highly interesting, and unfortunately still not fully proven. And since the rare biosphere is gaining more and more interest in associated microbiomes in general, due to the possibility of becoming relevant in abundance through symbiont shuffling, and afford mechanisms of acclimatization under changing conditions, or favour opportunist colonization and dysbiosis. The arguments raised by the reviewer are reasonable and we partially agree, but there are many factors that do not make

metabarcoding a mathematically trustful approach that reflects microbial communities. Some taxa seem to be favoured in the sequencing process with respect to other groups due to primers selection, or from the extraction procedures, among other factors, and this affects the number of reads attributed to one taxon or another respect to the rest of the community. For these reasons compositional approaches are being more accepted in microbiome studies lately (with respect to relative abundance). But despite these technical challenges, in either LMA as in HMA sponges, the amount of seawater left in the sample (in the sponge channels) after collection, drying and rinsing (let's say 3–5 drops) represents a negligible small proportion with respect to any of both tissues. Considering also that the seawater is not concentrated in these 3–5 drops, as it is in the filter used to characterize seawater communities for the sequencing comparison, the microbes recovered by sequencing in the sponges of seawater origin must be, by probability, somewhat enriched or technically favoured with respect to the microbes present in the actual sponge organic material. All this said, at the end, and as the reviewer says: “It is also common that LMA species share more taxa (including their core members) with the water, while HMA usually harbour a more sponge-specific community”, we find these same trends in our data, thus there is no contradiction to what it is defined in the sponge microbiome knowledge. At the moment, we just need to improve our approaches to be able to prove with evidence which microbes acquired from the seawater do perform a symbiotic activity in the host, and even this would be an intricate aspect to describe: what is a functional role?; the exchange of nutrients?; the elimination of waste or oxidative stress radicals?; providing protection with metabolites?; staying in dormant low abundances but becoming active and abundant under challenging conditions?; or just being there occupying a niche that otherwise would be occupied by a “worse” opportunistic strain?... To reach a consensus, we added a clarifying sentence in the text, as reported in the previous response, to address this debate on the symbiotic nature of the rare taxa found in *Crambe crambe*.

- I think I overlooked before the classification of the dominant symbiont in *Crambe*. It has been lately identified as Gammaproteobacteria, order Ca. Tethybacterales. Double check that. See: Taylor JA, Palladino G, Wemheuer B, Steinert G, Sipkema D, Williams TJ, et al. Phylogeny resolved, metabolism revealed: functional radiation within a widespread and divergent clade of sponge symbionts. *ISME J.* 2021;15:503–19.

Thank you for the heads up. Seems like we also overlooked this new classification based on MAGs. We have accordingly updated the main text as: “Our results based on SILVA v.128 release 69, expand such taxonomic resolution, by identifying a dominant Betaproteobacteriales ASV in Family EC94 (~70–80 %). Recently, predominant Gammaproteobacteria associating with several sponges including *C. crambe*, were proposed to take part of a new order named Ca. Tethybacterales, and in particular to the species Ca. *Beroebacter blanensis*, in *C. crambe* hosts from Spain ¹¹⁰. Along this dominant symbiont, we found a moderate representation of cyanobacteria, and a high number of minor bacterial taxa.” (L 535 – 542). Also in “Among bacteria mostly associated with *C. crambe*, we found the EC94 group (Betaproteobacteriales), which has been previously reported as an important taxon in other LMA sponges from a variety of habitats of Antarctic ecosystems ¹⁴¹, and in corals ¹⁴². The maintenance of EC94 (Ca. Tethybacterales ¹¹⁰) group over evolutionary time, ...” (L 645 – 648).

- One last thing that does not affect this work directly but relates with your future studies on the phenotypes as you commented. You have noticed that *Petrosia* species present the most diverse microbiome, be aware that this species has been described to present low gene flow, somehow close to a speciation event. Maybe your differently adapted phenotypes belong to isolated populations that do harbour different microbiomes. See: [10.1111/mec.15635](https://doi.org/10.1111/mec.15635)

We greatly appreciate these tips and recommendations, some colleagues abroad are working on these exciting topics, and we are attending to contribute with our expertise along.

As said, reviewer #2 has been constructive and suggestive, and we cannot do anything other than thank the scrupulous and careful revisions made.

Laura Núñez-Pons and Co-workers

REVIEWERS' COMMENTS:

Reviewer #1 (Remarks to the Author):

The authors have made revisions that address my concerns.

Reviewer #2 (Remarks to the Author):

This last version of Mazzella et al. has addressed all my previous concerns. For the issues that included open debates, I think the authors have reached an adequate consensus that will satisfy all readers.

I believe this is now a high-quality report and is acceptable for publication.